# History-Guided Video Diffusion

**Kiwhan Song\*** [1]  **Boyuan Chen\*** [1]  **Max Simchowitz** [2]  **Yilun Du** [3]  **Russ Tedrake** [1]  **Vincent Sitzmann** [1]

## Abstract

Classifier-free guidance (CFG) is a key technique for improving conditional generation in diffusion models, enabling more accurate control while enhancing sample quality. It is natural to extend this technique to video diffusion, which generates video conditioned on a variable number of context frames, collectively referred to as history. However, we find two key challenges to guiding with variable-length history: architectures that only support fixed-size conditioning, and the empirical observation that CFG-style history dropout performs poorly. To address this, we propose the Diffusion Forcing Transformer (DFoT), a video diffusion architecture and theoretically grounded training objective that jointly enable conditioning on a flexible number of history frames. We then introduce *History Guidance*, a family of guidance methods uniquely enabled by DFoT. We show that its simplest form, *vanilla history guidance*, already significantly improves video generation quality and temporal consistency. A more advanced method, *history guidance across time and frequency* further enhances motion dynamics, enables compositional generalization to out-of-distribution history, and can stably roll out extremely long videos. Project website: https://boyuan.space/history-guidance

## 1 Introduction

Diffusion models are effective generative models in domains such as image, sound, and video. Critical to their success is classifier-free guidance (CFG) (Ho & Salimans, 2022), which trades off between sample quality and diversity by jointly training a conditional and an unconditional diffusion model and combining their score estimates when sampling.

In the realm of video generative models, CFG commonly relies on either text or image prompts as conditioning vari-

---

\*Equal contribution   [1]MIT [2]Carnegie Mellon University [3]Harvard University. Correspondence to: Kiwhan Song <kiwhan@mit.edu>, Boyuan Chen <boyuanc@mit.edu>.

*Proceedings of the 42$^{nd}$ International Conference on Machine Learning*, Vancouver, Canada. PMLR 267, 2025. Copyright 2025 by the author(s).

ables. Yet, another conditioning variable, namely the entire collection of previous video frames, or *history*, deserves further exploration. In this paper, we investigate the following question: Can we use different portions of history - variable lengths, subsets of frames, and even different image-domain frequencies - as a form of guidance for video generation? Importantly, CFG with flexible history is incompatible with existing diffusion model architectures and the most obvious fix significantly degrades sample quality (see Section 3).

To address these limitations, we propose the Diffusion Forcing Transformer (DFoT), a video diffusion framework that enables flexible conditioning on any portion of the input history. Extending the "noising-as-masking" paradigm in Diffusion Forcing (Chen et al., 2024) to non-causal transformers, DFoT trains video diffusion models by applying independent noise levels to each frame. During sampling, portions of the history can be selectively masked with noise, enabling flexible conditioning and guidance. For instance, in CFG, the unconditional score corresponds to our model with the entire history masked out. Notably, DFoT is compatible with existing architectures such as DiT (Peebles & Xie, 2023) and U-ViT (Hoogeboom et al., 2023; 2024) and can be efficiently implemented through fine-tuning of pre-trained video diffusion models.

At sampling time, the DFoT facilitates a family of history-conditioned guidance methods, collectively referred to as *History Guidance* (HG). The simplest of these, *Vanilla History Guidance* (HG-v), uses an arbitrary length of history as the conditioning variable for CFG. Notably, even this simple method significantly enhances video quality. We further introduce two advanced methods enabled by the DFoT: *Temporal History Guidance* (HG-t) and *Fractional History Guidance* (HG-f) . These extend history guidance beyond a special case of CFG. Temporal History Guidance combines scores from different history windows. Fractional History Guidance conditions on history windows corrupted by varying levels of noise, effectively acting as a "low-pass filter" on historical frames. With minor modifications, it can also target specific *frequency bandwidths* to enhance the dynamic degree of generated videos (hence the frequency-based terminology). Together, we compose HG-t and HG-f to create a comprehensive history guidance paradigm, which we term *history guidance across time and frequency* (HG-tf).

The Diffusion Forcing Transformer and associated History

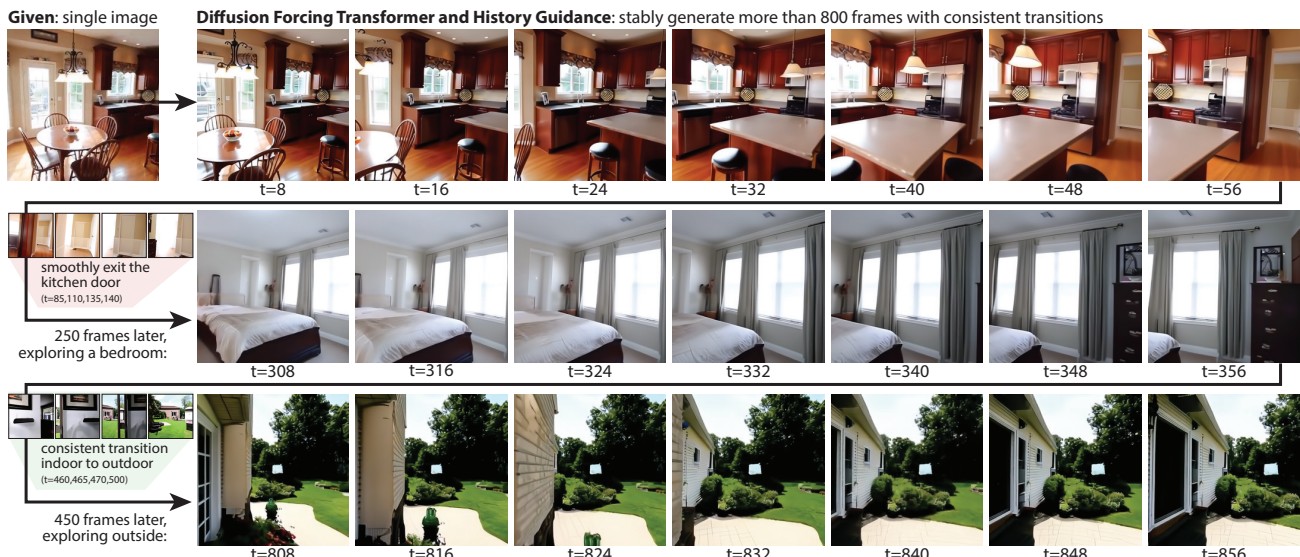

**Given**: single image — **Diffusion Forcing Transformer and History Guidance**: stably generate more than 800 frames with consistent transitions

t=8   t=16   t=24   t=32   t=40   t=48   t=56

smoothly exit the kitchen door (t=85,110,135,140)

250 frames later, exploring a bedroom:

t=308   t=316   t=324   t=332   t=340   t=348   t=356

consistent transition indoor to outdoor (t=460,465,470,500)

450 frames later, exploring outside:

t=808   t=816   t=824   t=832   t=840   t=848   t=856

*Figure 1.* **Diffusion Forcing Transformer with history guidance enables stable rollout of extremely long videos**. We visualize 21 frames from an 862-frame long navigation video generated by our DFoT model from a *single image* in a test set video that the model has never seen before. **Best viewed as videos on our project website.**

Guidance methods dramatically improve the quality and consistency of video generation, enabling the creation of exceptionally long videos through autoregressive extension, outperforming the de facto standard DiT diffusion and performing on par with industry models trained with an order of magnitude more compute. In Fig. 1, we showcase our method by using history guidance across time and frequency with DFoT to generate an 862-frame navigation video from a single image—many times longer than prior results and the maximum video length in the training set.

Our contributions can be summarized as follows: **1.** We propose the *Diffusion Forcing Transformer* (DFoT), a competitive video diffusion framework that enables sampling-time conditioning using *any portion* of history, a capability that is difficult to achieve with existing models. **2.** We introduce *History Guidance* (HG), a family of history-conditioned guidance methods enabled by DFoT that significantly enhance sample consistency, motion dynamics, and visual quality in video diffusion. **3.** We empirically demonstrate the state-of-the-art performance and new capabilities enabled by our method, especially in long video generation. Additionally, we provide a theoretical justification of the training objective through a variational lower bound.

## 2 Preliminaries and Related Work

**Diffusion Models.** Diffusion models (Sohl-Dickstein et al., 2015; Ho et al., 2020; Song et al., 2021) define a forward process that transforms a data distribution into white noise via a stochastic process over increasing *noise levels* $k \in [0,1]$: $\mathbf{x}^k = \alpha_k \mathbf{x}^0 + \sigma_k \boldsymbol{\epsilon}$, where $\boldsymbol{\epsilon} \sim \mathcal{N}(0, I)$. The goal of the model is to reverse this process by learning to estimate the *score function* $s_{\boldsymbol{\theta}}(\mathbf{x}^k, k) \approx \nabla \log p_k(\mathbf{x}^k)$ (Vincent, 2011),

which enables iterative denoising of a data point, gradually transforming it from white noise back to a sample from the original distribution. In practice, the score function is often parameterized as an affine function of alternative objectives such as the noise prediction $\boldsymbol{\epsilon}_{\boldsymbol{\theta}}(\mathbf{x}^k, k) \approx \boldsymbol{\epsilon}$.

**Video Diffusion Models (VDMs).** VDMs have enabled the generation of realistic, high-resolution videos (Brooks et al., 2024; Yang et al., 2024; Zheng et al., 2024; Kong et al., 2024). Their success is largely attributed to advancements such as transferring successful image diffusion models (Singer et al., 2022; Guo et al., 2023), scaling data and model (Blattmann et al., 2023a), improving transformer-based architectures (Peebles & Xie, 2023; Gupta et al., 2024; Jin et al., 2024), and enhancing computational efficiency through multi-stage approaches like latent VDMs (He et al., 2022; Blattmann et al., 2023b; Ma et al., 2024; Yin et al., 2024). Many of these models (Blattmann et al., 2023a; Yang et al., 2024) focus on generating videos from a single first image. In contrast, our model is designed to condition on arbitrary length histories, a crucial capability for autoregressively extending newly generated videos.

**Conditional Diffusion Sampling with Guidance.** *Classifier-free guidance* (CFG) (Ho & Salimans, 2022) is a crucial technique for improving sample quality in diffusion models. CFG jointly trains conditional and unconditional models $s_{\boldsymbol{\theta}}(\mathbf{x}, \mathbf{c}, k) \approx \nabla \log p_k(\mathbf{x}^k|\mathbf{c})$ and $s_{\boldsymbol{\theta}}(\mathbf{x}, \varnothing, k) \approx \nabla \log p_k(\mathbf{x}^k)$ by randomly dropping out the conditioning $\mathbf{c}$. During sampling, the true conditional score $\nabla \log p_k(\mathbf{x}^k|\mathbf{c})$ is replaced with the weighted score

$$\nabla \log p_k(\mathbf{x}^k) + \omega \left[ \nabla \log p_k(\mathbf{x}^k|\mathbf{c}) - \nabla \log p_k(\mathbf{x}^k) \right], \quad (1)$$

where $\omega \geq 1$ is the *guidance scale* that pushes the sample towards the conditioning. In VDMs, CFG is predominantly

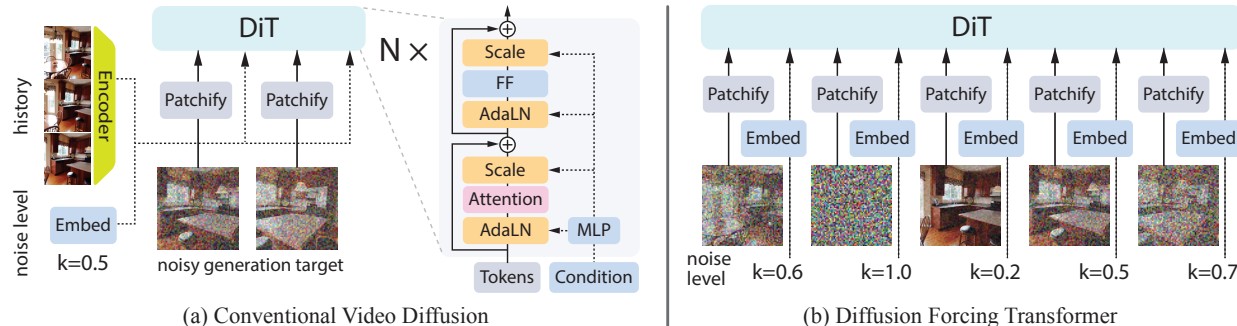

(a) Conventional Video Diffusion | (b) Diffusion Forcing Transformer

*Figure 2.* **Comparison of the conventional conditional video diffusion models and Diffusion Forcing Transformer.** At training time, conventional (a) approaches treat history as part of the conditioning input, first encoded by an *separate* encoder and then injected to the DiT via Adaptive Layer Norm and scaling. The Diffusion Forcing Transformer (b) instead does not distinguish between history and generation target frames. It trains a DiT to denoise *all* frames of a sequence, where frames have independently varying noise levels.

used for text guidance (Ho et al., 2022b; Wang et al., 2023). For frame conditioning, "first frame" guidance is commonplace in image-to-video models (Blattmann et al., 2023a; Yang et al., 2024), or "fixed set of few frames" (Blattmann et al., 2023b; Gupta et al., 2024; Watson et al., 2025), likewise in multi-view diffusion models (Gao et al., 2024).

Our work generalizes CFG by enabling guidance with a variable number of conditioning frames and later extends beyond the conventional approach of subtracting an unconditioned score - similar to prior works in compositional generative models (Du & Kaelbling, 2024; Liu et al., 2022; Du et al., 2023), we compose score from multiple conditioning to combine their behaviors. Additionally, we eliminate the reliance on binary-dropout training, the default mechanism for enabling CFG, which we empirically show performs sub-optimally when extended to history guidance.

**Diffusion Forcing.** Traditionally, diffusion models are trained using uniform noise levels across all tokens. Diffusion Forcing (DF) (Chen et al., 2024) proposes training sequence diffusion models with independently varied noise levels per frame. Although DF provides theoretical and empirical support for this approach, their work focuses on causal, state-space models. CausVid (Yin et al., 2024) builds on DF by scaling it to a causal transformer, creating an autoregressive video foundation model. Our work extends the flexibility of DF by developing both the theory and architecture for non-causal, state-free models, enabling new, unexplored capabilities in video generation.

## 3 Challenges when Guiding with History

Video diffusion models are conditional diffusion models $p(\mathbf{x}|\mathbf{c})$, where $\mathbf{x}$ denotes frames to be generated, and $\mathbf{c}$ represents the conditioning (e.g. text prompt, or a few observed prior frames). For simplicity, we refer to the latter as *history*, even when the observed images could be e.g. a subset of keyframes that are spaced across time. Our discussion of $\mathbf{c}$ will focus exclusively on history conditioning and exclude text or other forms of conditioning in notation.

Formally, let $\mathbf{x}_\mathcal{T}$ denote a $T$-frame video clips with indices $\mathcal{T} = \{1, 2, \ldots, T\}$. Define $\mathcal{H} \subset \mathcal{T}$ as the indices of history frames used for conditioning, and $\mathcal{G} = \mathcal{T} \setminus \mathcal{H}$ as the indices of the frames to be generated. Our objective is to model the conditional distribution $p(\mathbf{x}_\mathcal{G}|\mathbf{x}_\mathcal{H})$ with a diffusion model.

We aim to extend classifier-free guidance (CFG) to this setting. Since the history $\mathbf{x}_\mathcal{H}$ serves as conditioning, sampling can be performed by estimating the following score:

$$\nabla \log p_k(\mathbf{x}_\mathcal{G}^k) + \omega \left[ \nabla \log p_k(\mathbf{x}_\mathcal{G}^k|\mathbf{x}_\mathcal{H}) - \nabla \log p_k(\mathbf{x}_\mathcal{G}^k) \right]. \quad (2)$$

This approach differs from conventional CFG in two ways: 1) The generation $\mathbf{x}_\mathcal{G}$ and conditioning history $\mathbf{x}_\mathcal{H}$ belong to the same signal $\mathbf{x}_\mathcal{T}$, differing only in their indices $\mathcal{G}, \mathcal{H} \subset \mathcal{T}$; thus, the generated $\mathbf{x}_\mathcal{G}$ can be reused as conditioning $\mathbf{x}_\mathcal{H}$ for generating subsequent frames. 2) The history $\mathbf{x}_\mathcal{H}$ can be any subset of $\mathcal{T}$, allowing its length to vary. Guiding with history, therefore, requires a model that can estimate both conditional and unconditional scores given arbitrary subsets of video frames. Below, we analyze how these differences present challenges for implementation within the current paradigm of video diffusion models (VDMs).

**Architectures with fixed-length conditioning.** As shown in Figure 2a, DiT (Peebles & Xie, 2023) or U-Net-based diffusion models (Bao et al., 2023; Rombach et al., 2022) typically inject conditioning using AdaLN (Peebles & Xie, 2023; Perez et al., 2018) layers or by concatenating the conditioning with noisy input frames along the channel dimension. This design constrains conditioning to a fixed-size vector. While some models adopt sequence encoders for variable-length conditioning (e.g., for text inputs), these encoders are often pre-trained (Yang et al., 2024) and cannot share parameters with the diffusion model to encode history frames. Consequently, guidance has been limited to fixed-length and generally short history (Blattmann et al., 2023a; Xing et al., 2023; Yang et al., 2024; Watson et al., 2025).

**Framewise Binary Dropout performs poorly.** Classifier-free guidance is typically implemented using a single network that jointly represents the conditional and uncondi-

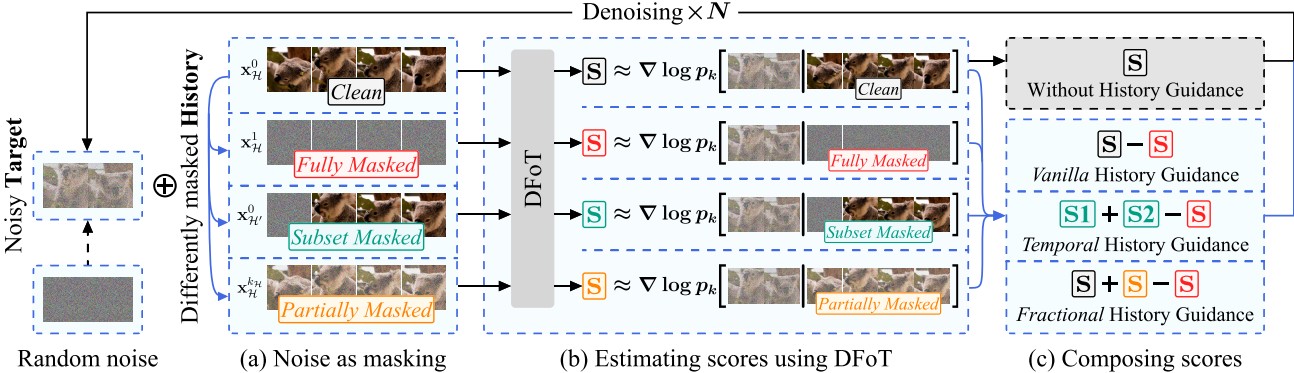

*Figure 3.* **Sampling with DFoT and History Guidance.** A DFoT can be used to estimate scores conditioned on differently masked histories using *noise as masking*. This includes clean (full history), fully masked (unconditional), subset masked (shorter history), or partially masked (low-frequency history). These scores can be composed when sampling to obtain a family of *History Guidance* methods.

tional models. These are trained via *binary dropout*, where the conditioning variable $\mathbf{c}$ is randomly masked during training with a certain probability. History guidance can, in principle, be achieved by randomly dropping out subsets of history frames during training. However, our ablations (Sec. 6.2) reveal that this approach performs poorly. We hypothesize that this is due to inefficient token utilization: although the model processes all $|\mathcal{T}|$ frames via attention, only a random subset of $|\mathcal{G}|$ frames contribute to the loss. This becomes more pronounced as videos grow longer, making framewise binary dropout a suboptimal choice.

## 4 The Diffusion Forcing Transformer

In this section, we introduce the Diffusion Forcing Transformer (DFoT), a simple yet powerful video diffusion framework designed to model score functions associated with *different portions of history*. This includes variable-length histories, arbitrary subsets of frames, and even history processed at different image-domain frequencies. DFoT improves video generation performance as a base model even without guidance. By addressing the challenges outlined in Section 3, DFoT further enables guidance with flexible history and a more advanced family of history guidance methods described in Section 5.

**Noise as Masking.** The forward diffusion process turns the $t$-th frame $\mathbf{x}_t$ of a video sequence into a noisy frame $\mathbf{x}_t^{k_t}$ at noise levels $k_t \in [0, 1]$. One can interpret this as progressively masking $\mathbf{x}_t$ with noise (Chen et al., 2024) - $\mathbf{x}_t^0$ is clean and hence unmasked, $\mathbf{x}_t^1$ is *fully masked* and contains no information about the original $\mathbf{x}_t$. Intermediate noise levels $(0 < k_t < 1)$ yield a *partially masked* frame $\mathbf{x}_t^{k_t}$, retaining a noisy snapshot of the original frame's information.

**History as noise-free frames.** Denoising generated frames $\mathbf{x}_\mathcal{G}^k$ conditioned on history $\mathbf{x}_\mathcal{H}$ can be unified under the noise-as-masking framework. Specifically, this involves denoising the entire sequence of frames $\mathbf{x}_\mathcal{H} \cup \mathbf{x}_\mathcal{G}^k$ with noise levels $k_\mathcal{T} = [k_1, k_2, \cdots, k_T]$ defined as:

$$k_t = \begin{cases} 0 & \text{if } t \in \mathcal{H} \\ k & \text{if } t \in \mathcal{G}. \end{cases} \tag{3}$$

This formulation treats history and generated frames as parts of the same input to the transformer, rather than separating history as a distinct "conditioning" input (see Figure 2 and Section 3). This unification allows any full-sequence transformer to be fine-tuned into a history-conditional model with variable-length history, simply by varying the noise levels within each sequence.

**Training: Per-frame Independent Noise Levels.** As illustrated in Figure 2b, instead of setting noise levels to zero for all history frames, we adopt *per-frame independent noise levels* introduced in Diffusion Forcing (Chen et al., 2024). Each frame $\mathbf{x}_t \in \mathbf{x}_\mathcal{T}$ is assigned an independent noise level $k_t \in [0, 1]$, resulting in random sequences of noise levels $k_\mathcal{T}$ in contrast with Equation 3. The DFoT model is then trained to minimize the following noise prediction loss, where $\boldsymbol{\epsilon}_\mathcal{T}$ denotes noise added to all frames:

$$\mathbb{E}_{k_\mathcal{T}, \mathbf{x}_\mathcal{T}, \boldsymbol{\epsilon}_\mathcal{T}} \left[ \| \boldsymbol{\epsilon}_\mathcal{T} - \boldsymbol{\epsilon}_\theta(\mathbf{x}_\mathcal{T}^{k_\mathcal{T}}, k_\mathcal{T}) \|^2 \right], \tag{4}$$

Crucially, noise levels are selected independently for all frames without distinguishing the past and the future. This enables parallel training while also allowing *non-causal* conditioning on partially masked future frames. In Appendix A.5, we further discuss a simplified objective when $\max(|\mathcal{H}|) \ll T$ and a causal adaptation of our model. In Appendix A.1, we justify this training objective as optimizing a (reweighted) valid Evidence Lower Bound (ELBO) on the expected log-likelihoods:

**Theorem 4.1** (Informal). *The DFoT training objective (Equation (4)) optimizes a reweighting of an Evidence Lower Bound (ELBO) on the expected log-likelihoods.*

Compared to conventional video diffusion methods, where a single noise level $k \in [0, 1]$ is uniformly applied to all generation frames $\mathbf{x}_\mathcal{G}$, our approach provides two key benefits: (1) token utilization is improved by computing a loss

*Table 1.* **Comparison with generic diffusion models on Kinetics-600.** "✗", "▲", and "✔" indicate whether a model can condition on a "single predefined," "arbitrary under approximation," or "arbitrary" history. DFoT, both trained from *scratch* and *fine-tuned*, outperforms all generic diffusion baselines under the same architecture and is on par with industry models trained with more compute resources (see Appendix C.4).

| | Flexible? | Method | FVD ↓ |
|---|---|---|---|
| Industry size and compute | ✗ | MAGVIT-v2 (Yu et al., 2023b) | $4.3_{\pm 0.1}$ |
| | | W.A.L.T (Gupta et al., 2024) | $\mathbf{3.3}_{\pm 0.1}$ |
| | | Rolling Diffusion (Ruhe et al., 2024) | 5.2 |
| | ▲ | Video Diffusion (Ho et al., 2022b) | $16.2_{\pm 0.3}$ |
| | ✔ | MAGVIT (Yu et al., 2023a) | $9.9_{\pm 0.3}$ |
| Same Architecture | ✗ | SD | $4.8_{\pm 0.0}$ |
| | ▲ | FS | $95.5_{\pm 0.4}$ |
| | ✔ | BD | $6.4_{\pm 0.1}$ |
| | | **DFoT** (*scratch*) | $\mathbf{4.3}_{\pm 0.1}$ |
| | | **DFoT** (*fine-tuned from FS*) | $\underline{4.7}_{\pm 0.0}$ |

conditioned on all frames $\mathbf{x}_{\mathcal{T}}$ instead of a smaller subset; second, (2) this objective places variable history lengths "in-distribution" of the training objective, leading to more flexible use of history lengths as detailed below.

**Sampling: Conditioning on Arbitrary History.** Unlike standard VDMs that require fixed-length history during sampling, DFoT allows conditioning on arbitrary history. To generate $\mathbf{x}_{\mathcal{G}}$ conditioned on $\mathbf{x}_{\mathcal{H}}$ at each sampling step with noise level $k$, we estimate the conditional score $\nabla \log p_k(\mathbf{x}_{\mathcal{G}}^k|\mathbf{x}_{\mathcal{H}})$ by feeding the model noisy $\mathbf{x}_{\mathcal{G}}^k$ and clean history frames $\mathbf{x}_{\mathcal{H}}^0$. Sampling is then performed using standard score-based sampling schemes such as DDPM (Ho et al., 2020) or DDIM (Song et al., 2020). This flexibility in conditioning enables history guidance and its more advanced variants, as described in the next section.

## 5 History Guidance

Leveraging the flexibility of Diffusion Forcing Transformer (DFoT), we introduce *History Guidance* (HG), a family of techniques for history-conditioned video generation. These methods enhance generation quality, improve motion dynamics, enable robustness to out-of-distribution (OOD) histories, and unlock novel capabilities such as compositional video generation. Please refer to Figure 3 for an overview.

**Simplest HG: Vanilla History Guidance.** The simplest form of HG, referred to as *Vanilla History Guidance* (HG-v), directly performs classifier-free guidance (CFG) with a chosen history length, following Equation 2. The conditional score for any history $\mathcal{H}$ can be computed as described in the previous section. To perform CFG, we need to estimate the *unconditional* score $\nabla \log p_k(\mathbf{x}_{\mathcal{G}}^k)$. Notably, the unconditional score is a special case of the conditional score with $\mathcal{H} = \varnothing$ and can be estimated by masking history frames $\mathbf{x}_{\mathcal{H}}$ with *complete* noise. Even this simple form of HG

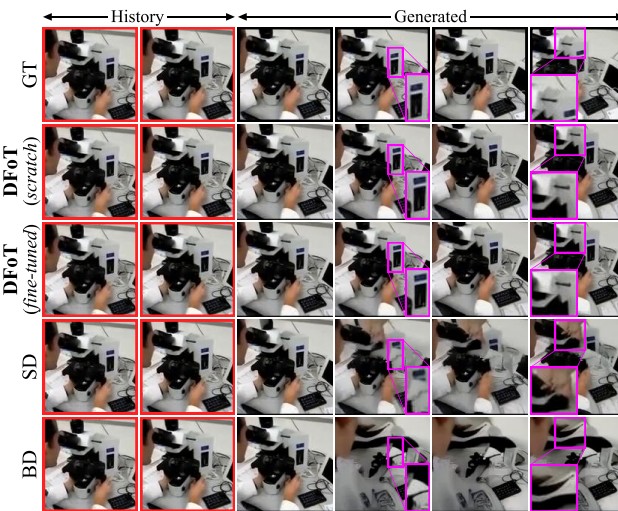

*Figure 4.* **Qualitative comparison on Kinetics-600.** DFoT (both *scratch* and *fine-tuned*) generates higher-quality samples consistent with the history than baselines. FS omitted for poor quality. We show 6 of 16 frames; see Figure 14 for more comparisons.

significantly improves generation quality and consistency.

**History Guidance Across Time and Frequency.** While history guidance has been presented as a special case of CFG so far, its full potential extends far beyond CFG. Consider the following generalization of Equation 2:

$$\nabla \log p_k(\mathbf{x}_{\mathcal{G}}^k) + \sum_i \omega_i \left[ \nabla \log p_k(\mathbf{x}_{\mathcal{G}}^k|\mathbf{x}_{\mathcal{H}_i}^{k_{\mathcal{H}_i}}) - \nabla \log p_k(\mathbf{x}_{\mathcal{G}}^k) \right], \quad (5)$$

where the total score is a weighted sum of conditional scores, each conditioned on possibly *different segments of history* $\{\mathcal{H}_i\}$, and each masked with a possibly *different noise level* $k_{\mathcal{H}_i}$. This formulation enables better generalization than a single score function conditioned on a full long history. By composing scores, each individual score component operates on a restricted conditional context, reducing the likelihood of being out-of-distribution (Du & Kaelbling, 2024). Appendix A.3 provides informal mathematical intuition on why summing conditional scores is permissible.

Equation 5 effectively allows us to compose the scores conditioned on 1) different history subsequences, and 2) history frames that are partially noisy. We refer to these two principal axes as *time* and *frequency*, which together form a 2D plane of options that we refer to as *History Guidance across Time and Frequency*. For simplicity, we introduce composition along these two axes separately.

**Time Axis: Temporal History Guidance.** Due to the curse of dimensionality, the amount of data that we require to guarantee constant data support grows exponentially with the length of history we wish to condition on. As a result, history conditioned models are particularly prone to out-of-distribution (OOD) history without an inductive bias of sparse dependency. Common symptoms include blowing up or overfitting to irrelevant features. To address this, we propose *Temporal History Guidance* (HG-t), which composes

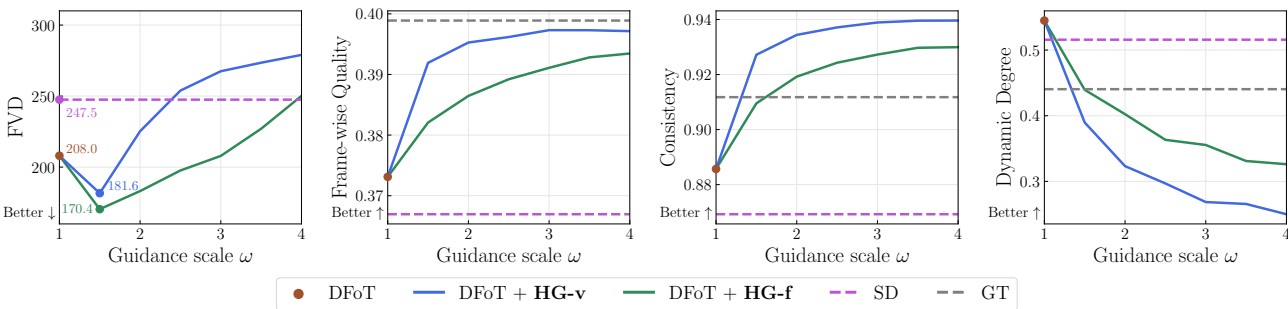

Figure 5. **Various metrics as a function of guidance scale $\omega$ for vanilla and fractional history guidance on Kinetics-600**, comparing against $\omega = 1$ (●, *w/o HG*), SD, and ground truth (GT). FS is omitted for poor performance (FVD = 1040). Vanilla history guidance trades off dynamics · diversity for quality · consistency. Fractional history guidance better balances these trade-offs, achieving the best FVD.

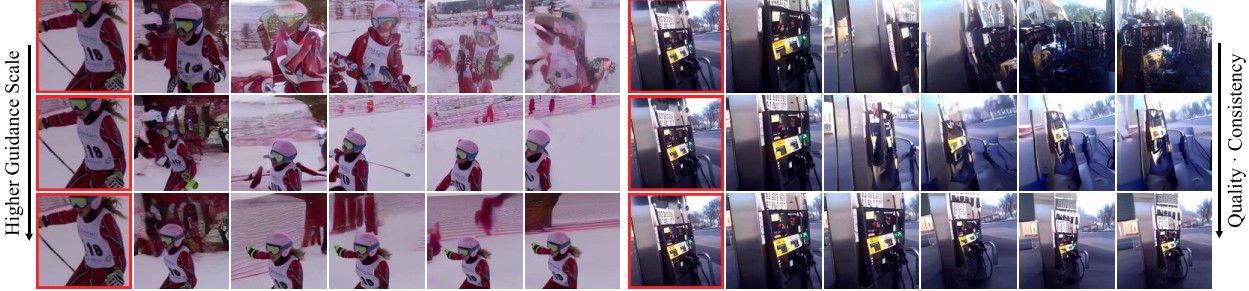

(a) **Vanilla history guidance significantly improves frame quality and consistency with an increasing guidance scale.** We sample with varying guidance scales $\omega = 1$ (top, *without history guidance*), 1.5 (middle), and 3 (bottom).

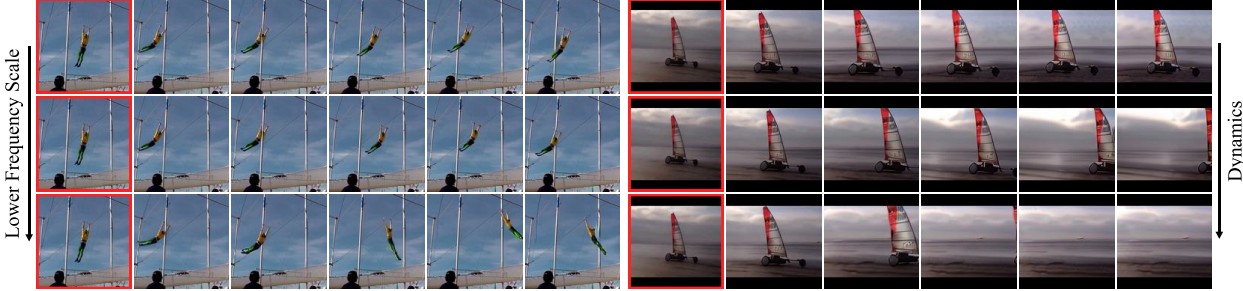

(b) **Fractional history guidance resolves the issue of static videos, improving dynamics by guiding with lower frequencies.** We sample with varying frequency scales, with $k_{\mathcal{H}} = 0$ (top, *vanilla guidance leading to static videos*), 0.3 (middle), and 0.6 (bottom).

Figure 6. Qualitative results for vanilla · fractional history guidance on Kinetics-600. *Best viewed zoomed in.* Red box = history frames.

scores conditioned on different subsequences of history by setting $k_{\mathcal{H}_i} = 0$ in Equation 5. This composition can be performed with either: 1) long and short history $\{\mathcal{H}_{\text{long}}, \mathcal{H}_{\text{short}}\}$, aiming to trade-off between the two imperfect predictive models, reducing the likelihood of OOD while preserving both long and short-term dependencies, or 2) multiple short, overlapping in-distribution histories $\{\mathcal{H}_{\text{short}_1}, \mathcal{H}_{\text{short}_2}, \cdots\}$, to simulate the conditional distribution of the full history.

**Frequency Axis: Fractional History Guidance.** We observe that a major failure mode of HG-v under high guidance scales is the generation of overly static videos with minimal motion. This occurs because HG-v encourages consistency with history, leading to a trivial solution of simply copying the most recent history frame. To address this, we propose *Fractional History Guidance* (HG-f), which guides the sampling process using *fractionally masked history*. Frac-

tionally masking history retains only low-frequency information (Dieleman (2024), Appendix A.2), allowing high-frequency details (e.g., fine textures and fast motions) to remain unconstrained by guidance. This approach makes videos more dynamic while maintaining consistency, which is mainly associated with low-frequency details. Specifically, the HG-f score is given by:

$$\nabla \log p_k(\mathbf{x}_{\mathcal{G}}^k | \mathbf{x}_{\mathcal{H}}) + \omega \left[ \nabla \log p_k\left(\mathbf{x}_{\mathcal{G}}^k | \mathbf{x}_{\mathcal{H}}^{k_{\mathcal{H}}}\right) - \nabla \log p_k(\mathbf{x}_{\mathcal{G}}^k) \right], \quad (6)$$

where $k_{\mathcal{H}} \in (0, 1)$ controls the degree of masking to focus on lower-frequency details, and $\omega$ is the guidance scale for the partially masked history $\mathbf{x}_{\mathcal{H}}^{k_{\mathcal{H}}}$. In principle, different history frames could contribute information at different frequency bands, such as high-frequency details from recent frames and low-frequency motion from earlier frames. While a detailed exploration of sophisticated sampling strategies is left to future work, our experiments show that even

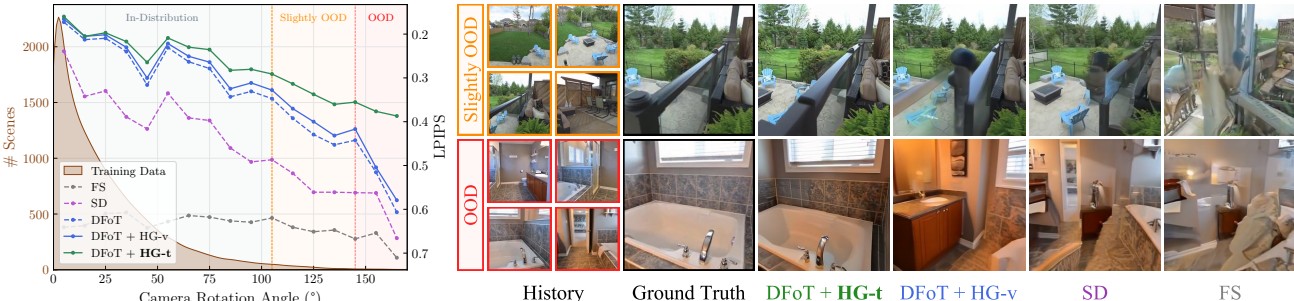

*Figure 7.* **Robust performance of temporal history guidance given OOD history unseen in the training data. Left:** Baselines sharply lose performance transitioning from in-distribution, slightly OOD, to OOD tasks, while DFoT with HG-t shows minimal drop **Right:** Baselines produce blurry, inconsistent frames with artifacts on slightly OOD history and unrecognizable frames on OOD history, whereas DFoT with HG-t generates high-quality, accurate samples. Each frame shown is one of four generated; see Figure 10 for full results.

simple implementations of HG-f significantly improve motion dynamics without sacrificing consistency.

## 6 Experiments

We empirically evaluate the performance of the Diffusion Forcing Transformer and history guidance. We first validate the DFoT as a generic video model without history guidance (Sec. 6.2), demonstrating the effectiveness of the modified training objective. Next, we examine the effectiveness and additional capabilities of history guidance (Secs. 6.3 and 6.4). Finally, we showcase very long videos generated by DFoT with history guidance (Sec. 6.5).

### 6.1 Experimental Setup

**Datasets.** Throughout our experiments, we train and evaluate a separate DFoT model for each dataset as follows: Kinetics-600 (Kay et al. (2017), 128×128), a standard video prediction benchmark, RealEstate10K or RE10K (Zhou et al. (2018), 256×256), a dataset of real-world indoor scenes with camera pose annotations, and Minecraft (Yan et al. (2023), 256×256), a dataset of long-context Minecraft navigation videos with discrete actions. We employ Fruit Swapping, an imitation learning task adapted from Diffusion Forcing (Chen et al., 2024) to test the combined ability to handle long-term memory and reactive behavior with a physical robot. Details are in Appendix C.1. We use Kinetics-600 for benchmarking and quantitative comparisons, and the other three for studying new applications.

**Baselines.** 1) Standard Diffusion (SD): A single-task model trained for specific test history lengths following the standard conditional diffusion setup (Gupta et al., 2024; Watson et al., 2025). 2) Binary-Dropout Diffusion (BD): An ablative baseline trained with framewise binary dropout for history guidance instead of independent per-frame noise levels. Note that BD requires DFoT's architecture as opposed to conditioning via adaptive LayerNorm to support flexible history lengths, effectively making it an ablation. 3) Full-Sequence Diffusion with Reconstruction Guidance (FS): An unconditional video diffusion model trained with

maximum sequence length. Flexible-length conditioning is achieved during sampling via history replacement and reconstruction guidance (Ho et al., 2022b).

**Evaluation.** To evaluate the overall video generation performance encompassing quality and diversity, we use Fréchet Video Distance (FVD, Unterthiner et al. (2018)). For a more detailed analysis of video quality, we use VBench (Huang et al., 2024), which provides separate scores for different aspects such as frame quality, consistency, and dynamics. For highly deterministic tasks, we evaluate according to Learned Perceptual Image Patch Similarity (LPIPS, Zhang et al. (2018)), computed frame-wise against the ground truth. Additional experimental details are provided in Appendix C.

### 6.2 Evaluating the Diffusion Forcing Transformer

We validate DFoT as a competitive video generative model *without* history guidance by answering the questions:

- **Q1:** How does DFoT compare to the conventional video diffusion approach in standard video benchmarks?
- **Q2:** Does binary dropout diffusion (BD) perform competitively as an alternative training approach that also supports flexible history?
- **Q3:** Is DFoT empirically flexible enough to handle arbitrary sets of history frames?
- **Q4:** Can we fine-tine an existing model into DFoT?

We summarize quantitative and qualitative results in Table 1 and Figure 4 respectively.

**Competitive Performance of DFoT (Q1) without Guidance.** DFoT outperforms all baselines, including single-task standard diffusion (SD), despite SD being optimized for the eval's specific history length. This demonstrates DFoT's flexibility without sacrificing task-specific performance, aligning with observations from (Chen et al., 2024).

**Limited Performance of Binary Dropout (Q2).** While BD enables flexible history conditioning, it suffers a significant performance drop compared to SD. Notably, BD produces artifacts and inconsistent generations (Figure 4), highlighting its inefficiency as an alternative to DFoT 's

training objective.

**Flexibility of DFoT (Q3).** We demonstrate DFoT's flexibility by tasking it with various video generation tasks on RE10K, such as future prediction, frame interpolation, and mixed history setups. As shown in Figure 11, DFoT generates consistent, high-quality samples across all tasks.

**Fine-tune existing models into DFoT (Q4).** As discussed in Sec. 4, an DFoT can be obtained by fine-tuning an existing video diffusion model. We fine-tune the full-sequence model on Kinetics-600 into a DFoT using only 12.5% of the training cost. The fine-tuned model surpasses all baselines and performs comparably to the DFoT trained from scratch (see Appendix D.1 for detailed analysis). This confirms the feasibility of fine-tuning large foundation models into DFoT to support history guidance.

### 6.3 Improving Video Generation via History Guidance

We examine the effect of history guidance on video quality in terms of frame-wise quality, frame-to-frame consistency and dynamic degree of generated video. We benchmark 64-frame video generation using sliding window rollout on Kinetics-600, a challenging setup that requires outstanding consistency to avoid blowing up. Note that this is a setup where conventional image-to-video models struggle since they can only condition on the final generated frame to extend the video. We present quantitative and quantitative results in Figures 5 and Figure 6 respectively.

**Vanilla History Guidance.** We visualize samples generated with vanilla history guidance with increasing guidance scale in Figure 6a. *Stronger history guidance consistently improves frame quality and consistency*, which is also reflected in their corresponding VBench scores in Figures 5b and 5c. In Figure 5a, we obtain the best FVD result with a small guidance scale of $\omega$=1.5. Beyond that, FVD increases sharply, indicating a loss of diversity with higher guidance scales, similar to the quality-diversity trade-off of CFG.

**Fractional History Guidance.** Despite notable quality improvements, we observe that vanilla history guidance tends to generate *static videos* at high guidance scales ($\omega \geq 3$), as illustrated in the top rows of Figure 6b, with significantly less motion than ground truth in Figure 5d. Fractional history guidance resolves this in the side-by-side visualization. We find that *guiding with lower frequencies (higher $k_{\mathcal{H}}$) consistently increases dynamics while maintaining quality*, as shown in Figure 6b. This further lowers the best FVD of vanilla history guidance (181.6) to 170.4, surpassing FS (1040), SD (247.5), and DFoT without guidance (208.0).

### 6.4 New Abilities via Temporal History Guidance

Temporal history guidance brings new capabilities to DFoT, allowing it to solve tasks impossible for previous models. We discuss three representative tasks.

**Task 1. Robust to Out-of-Distribution (OOD) History.** We evaluate robustness to OOD histories on RealEstate10K by creating scenarios with extreme camera rotations between history frames and ask the model to interpolate. Baselines fail to generalize, producing incoherent generations. In contrast, DFoT with temporal history guidance splits OOD histories into shorter, in-distribution subsequences, composing their scores to maintain both local and global dependencies. This enables DFoT to handle OOD histories effectively, as shown in Figure 7.

**Task 2. Long Context Generation.** Minecraft is a video dataset that requires long context to achieve good FVD scores. We found generating coherent videos with long contexts often leads to OOD histories. Baselines prioritize consistency with the context at the expense of quality. Our hypothesis is that temporal guidance blends scores from long-context and short-context models, balancing memory retention with robustness to OOD. This strategy improves FVD scores from 97.63 to 79.19, achieving long-term coherent high-quality generations. See Appendix D.3 for details.

**Task 3. Long-horizon yet Reactive Imitation Learning.** We test on a robotic manipulation task requiring both long-term memory for object rearrangement and short-term reactivity for disturbances. Each data point in the dataset contains either of these two behaviors but never both. Baselines fail to integrate the two behaviors, while DFoT combines full-history scores (for memory) with single-frame scores (for reactivity) using temporal history guidance. This allows the robot to recover from disturbances and complete tasks, achieving a success rate of 83% while baselines fail to perform the task completely. See Appendix D.4 for details.

### 6.5 Ultra Long Video Generation

In Figure 1, we present a showcase that utilizes all of this paper's contributions - we extend a single image to an 862-frame video in RE10K. Even the most high-performing prior methods can only roll out for dozens of frames under the same setup. This is made possible by enhanced quality, consistency, and rollout stability through history guidance, plus DFoT's flexibility that enables this. See Appendix C.9 for more details and Appendix D.6 for more samples (Figures 8a to 8d), including notable failures of other models.

## 7 Conclusion

Enabling flexible conditioning on different portions of history, the Diffusion Forcing Transformer not only establishes itself as a competitive video diffusion framework but also gives rise to History Guidance, a family of powerful history-conditioned guidance methods that significantly enhances video quality, consistency, and motion degree. Additionally, we demonstrate that DFoT can be efficiently fine-tuned from existing models, suggesting future potentials of integrating History Guidance with current foundation models.

## Acknowledgements

This work was supported by the National Science Foundation under Grant No. 2211259, by the Singapore DSTA under DST00OECI20300823 (New Representations for Vision, 3D Self-Supervised Learning for Label-Efficient Vision), by the Intelligence Advanced Research Projects Activity (IARPA) via Department of Interior/ Interior Business Center (DOI/IBC) under 140D0423C0075, by the Amazon Science Hub, and by the MIT-Google Program for Computing Innovation.

## Impact Statement

This paper aims to advance the field of video generative modeling. As a video generative model, our approach may enable the creation of longer, higher-quality videos, with potential applications in robotics and other fields. However, we acknowledge the potential risks associated with misuse, such as the generation of harmful or unethical content. We emphasize the importance of ethical considerations and responsible use of this work.

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

# A Proofs, Explanations, and Extensions

## A.1 Derivation of an ELBO

This section includes a derivation of an ELBO corresponding to the DFoT training objective. By taking a sequence modeling perspective, the derivation below streamlines that of the Diffusion Forcing ELBO in (Chen et al., 2024).

Let $\mathcal{T}$ denote the index set associated with a sequence $\mathbf{x}$, so that $\mathbf{x}_{\mathcal{T}} = (\mathbf{x}_t)_{t \in \mathcal{T}}$ is the whole sequence. We use the notation $\mathbf{k} = (k_t)_{t \in \mathcal{T}}$ for the sequence of noise levels. A *path* $\rho$ is a sequence of noising steps that transition from an unnoised sequence to a noised one. Specifically,

**Definition A.1** (Path). We define a **path** $\rho$ as a sequence $(\mathbf{k}^j)_{0 \le j \le N}$ that begins at zero noise $\mathbf{k}^0 = (0, 0, \ldots, 0)$, and terminates at full noise $\mathbf{k}^N = (K, K, \ldots, K)$.

Given a path $\rho$, we let $\mathbf{x}^\rho = \mathbf{x}^{\mathbf{k}^{0:N}}$ denote the sequence with $(\mathbf{x}_t^{\mathbf{k}_t})_{t \in \mathcal{T}}$. Note that there is nothing intrinsically causal or temporal about the indices $t$; indeed, we can define noising paths on other objects like trees or graphs. Examples of paths include:

- Autoregressive diffusion, where $k_t^j$ is equal to $K$ if $t \le \lfloor j/K \rfloor$, equal to $0$ if $t > \lfloor j/K \rfloor + 1$, and equal to $j - K \lfloor j/K \rfloor$ otherwise. This path looks like $(0, \ldots, 0)$, $(1, 0, \ldots, 0), \ldots, (K, 0, \ldots, 0)$, $(K, 1, 0, \ldots, 0)$, increasing lexicographically.

- Full-sequence diffusion, where $k_t^j = j$ and $N = K$; i.e. all points are denoised together.

- We can accomodate skips in noiseless, e.g. DDIM, or paths with linearly increasing noise, such as those considered in (Chen et al., 2024).

Typically, we assume that $k_t^j$ is non-decreasing in $j$ (the noise level is monotonic up to $\mathbf{k}^N = (K, \ldots, K)$).

The essential property that we require is that our learned model and forward process factor nicely along such paths. It is straightforward to check that this is indeed the case for Diffusion Forcing Transformer with these monotonic paths:

**Definition A.2** (Factoring Property). We say that a model $p_{\boldsymbol{\theta}}$ and forward process $q$ factor along a path $\rho$ if for any path, $\rho = (\mathbf{k}^1, \ldots, \mathbf{k}^N)$ be a path, $q(\mathbf{x}^{\mathbf{k}^{1:N}} \mid \mathbf{x}^{\mathbf{k}^0})$ factors as $q(\mathbf{x}^{\mathbf{k}^{1:N}} \mid \mathbf{x}^{\mathbf{k}^0}) = \prod_{j=1}^{n} q(\mathbf{x}^{\mathbf{k}^j} \mid \mathbf{x}^{\mathbf{k}^{j-1}})$, and $p_{\boldsymbol{\theta}}$ factors as $p_{\boldsymbol{\theta}}(\mathbf{x}^{\mathbf{k}^{0:N}}) = \prod_{j=1}^{N} p_{\boldsymbol{\theta}}(\mathbf{x}^{\mathbf{k}^{j-1}} \mid \mathbf{x}^{\mathbf{k}^j}) p_{\boldsymbol{\theta}}(\mathbf{x}^{\mathbf{k}^N})$, with $p_{\boldsymbol{\theta}}(\mathbf{x}^{\mathbf{k}^N})$ not depending on $\theta$.

When the model factors along paths, a general ELBO holds. We first state the general form, then specialize to Diffusion via Gaussian forward processes, and conclude with the proof of the general result.

**Theorem A.3.** *Suppose that $(p_{\boldsymbol{\theta}}, q)$ factor along a path $\rho = (\mathbf{k}^1, \ldots, \mathbf{k}^N)$. Then, for some constant $C$ not depending on $\theta$, we have*

$$\ln p(\mathbf{x}^{\mathbf{k}^0}) \ge C + \mathbb{E}_{\mathbf{x}^{\mathbf{k}^{1:N}} \sim q(\mathbf{x}^{\mathbf{k}^{1:N}} | \mathbf{x}^{\mathbf{k}_0})} \left[ \ln p_{\boldsymbol{\theta}}(\mathbf{x}^{\mathbf{k}^0} \mid \mathbf{x}^{\mathbf{k}^1}) + \sum_{j=1}^{N-1} \mathrm{D}_{\mathbb{KL}}\big(p_{\boldsymbol{\theta}}(\mathbf{x}^{\mathbf{k}^j} \mid \mathbf{x}^{\mathbf{k}^{j+1}}) \parallel q(\mathbf{x}^{\mathbf{k}^j} \mid \mathbf{x}^{\mathbf{k}^{j+1}}, \mathbf{x}^{\mathbf{k}^0}) \big) \right]. \quad (7)$$

*Consequently, if $\mathbb{E}_{\mathbf{k}^{1:N} \sim \mathcal{D}_p}$ denotes an expectation over paths $\rho = (\mathbf{k}^1, \ldots, \mathbf{k}^K)$ along which $(p_{\boldsymbol{\theta}}, q)$ factor, then*

$$\ln p(\mathbf{x}^{\mathbf{k}^0}) \ge C + \mathbb{E}_{\mathbf{k}^{1:N} \sim \mathcal{D}_p} \mathbb{E}_{\mathbf{x}^{\mathbf{k}^{1:N}} \sim q(\mathbf{x}^{\mathbf{k}^{1:N}} | \mathbf{x}^{\mathbf{k}_0})} \left[ \ln p_{\boldsymbol{\theta}}(\mathbf{x}^{\mathbf{k}^0} \mid \mathbf{x}^{\mathbf{k}^1}) + \sum_{j=1}^{N-1} \mathrm{D}_{\mathbb{KL}}\big(p_{\boldsymbol{\theta}}(\mathbf{x}^{\mathbf{k}^j} \mid \mathbf{x}^{\mathbf{k}^{j+1}}) \parallel q(\mathbf{x}^{\mathbf{k}^j} \mid \mathbf{x}^{\mathbf{k}^{j+1}}, \mathbf{x}^{\mathbf{k}^0}) \big) \right].$$

We now specialize Theorem A.3 to Gaussian diffusion. For now, we focus on the "$\mathbf{x}$-prediction" formulation of diffusion. The "$\boldsymbol{\epsilon}$-prediction", used throughout the main body of the text and the "$\mathbf{v}$-prediction formalism, which is the one used in our implementation, can be derived similarly (see Section 2 of (Chan et al., 2024) for a clean exposition). The following theorem is derived directly by applying standard likelihood and KL-divergence computations for the DDPM (Ho et al., 2020; Chan et al., 2024) to Theorem A.3.

For simplicity, we focus on paths with a single increment (e.g. DDPM), but extending to jumps (e.g. DDIM) is straightforward (albeit more notationally burdensome).

**Corollary A.4.** *Consider only paths $\rho$ for which $\mathbf{k}^j \geq \mathbf{k}^{j-1}$ entrywise, and for any $t$ and $j$ for which $k_t^j > k_t^{j-1}$, $k_t^j = k_t^{j-1} + 1$ increments by one.*

$$q(\mathbf{x}^{\mathbf{k}^{j+1}} \mid \mathbf{x}_t^{\mathbf{k}^j}) = \prod_{t:k_t^j < k_t^{j+1}} \mathcal{N}(\mathbf{x}_t^{k_t^j}; \sqrt{1 - \beta_{k_t^j}} \mathbf{x}_t^{k_t^{j-1}}, \beta_{k_t^j} \mathbf{I}), \tag{8}$$

*and define $\alpha_k = (1 - \beta_k)$, $\bar{\alpha}_k = \prod_{j=1}^k \alpha_j$. Suppose that we parameterize $p_{\boldsymbol{\theta}}(\mathbf{x}^{\mathbf{k}^j}; \mathbf{x}^{\mathbf{k}^{j+1}}, \mathbf{k}^j) = \mathcal{N}(\mu_{\boldsymbol{\theta}}(\mathbf{x}^{\mathbf{k}^j}; \mathbf{x}^{\mathbf{k}^{j+1}}, \mathbf{k}^j), \sigma_j^2)$, where further,*

$$\mu_{\boldsymbol{\theta}}(\mathbf{x}^{\mathbf{k}^j}; \mathbf{x}^{\mathbf{k}^{j+1}}, \mathbf{k}^j) = \frac{(1 - \bar{\alpha}_{j-1})\sqrt{\alpha_j}}{1 - \bar{\alpha}_j} \mathbf{x}^{\mathbf{k}_j} + \frac{(1 - \alpha_j)\sqrt{\bar{\alpha}_{j-1}}}{1 - \bar{\alpha}_j} \hat{\mathbf{x}}_{\boldsymbol{\theta}}(\mathbf{x}^{\mathbf{k}^j}; \mathbf{x}^{\mathbf{k}^{j+1}}, \mathbf{k}^j), \quad \sigma_j^2 := \frac{(1 - \alpha_j)(1 - \sqrt{\bar{\alpha}_{j-1}})}{1 - \bar{\alpha}_j}.$$

*Further, let $\hat{\mathbf{x}}_{\boldsymbol{\theta}}^0(\mathbf{x}_t^{k_t^j}; \mathbf{x}^{\mathbf{k}^{j+1}}, \mathbf{k}^j) = \hat{\mathbf{x}}_{\boldsymbol{\theta}}^0(\mathbf{x}^{\mathbf{k}^j}; \mathbf{x}^{\mathbf{k}^{j+1}}, \mathbf{k}^j)_t$ denote the $t$-block component of $\hat{\mathbf{x}}_{\boldsymbol{\theta}}^0(\mathbf{x}^{\mathbf{k}^j}; \mathbf{x}^{\mathbf{k}^{j+1}}, \mathbf{k}^j)$, and suppose that if $k_t^j = k_t^{j+1}$, then $\hat{\mathbf{x}}_{\boldsymbol{\theta}}^0(\mathbf{x}_t^{k_t^j}; \mathbf{x}^{\mathbf{k}^{j+1}}, \mathbf{k}^j) = \mathbf{x}^{\mathbf{k}^{j+1}}$ (i.e., if no denoising occurs, we do not re-predict the denoising). Then, for some distribution $\mathcal{D}_\rho$ over paths $\rho$ along which $(p_{\boldsymbol{\theta}}, q)$ satisfy the requisite factoring property, and for some constant $C$ independent of $p_{\boldsymbol{\theta}}$,*

$$\ln p_{\boldsymbol{\theta}}((\mathbf{x}^{\mathbf{k}^0})] \geq C + \mathbb{E}_{\rho = \mathbf{k}^{0:N} \sim \mathcal{D}_\rho} \mathbb{E}_{p, \mathbf{z}_{1:T}} \left[ \sum_{j=1}^N \sum_{t \in \mathcal{T}: k_t^j < k_t^{j+1}} c_{k_t^j} \| \hat{\mathbf{x}}_{\boldsymbol{\theta}}^0(\mathbf{x}_t^{k_t^j}; \mathbf{x}^{\mathbf{k}^{j+1}}, \mathbf{k}^j) - \mathbf{x}_t^{k_t^0} \|^2 \right],$$

*where above, we define $c_i = \frac{(1-\alpha_j)^2 \bar{\alpha}_{i-1}}{2\sigma^2(1 - \bar{\alpha}_i)^2}$.*

*Proof of Corollary A.4.* The first inequality follows from the standard computations for the "$\mathbf{x}$-prediction" formulation of Diffusion (see Section 2.7 of (Chan et al., 2024) and references therein). $\square$

*Remark* A.5 (Factoring). Observe that forward process in Equation (8) naturally factorizes across all the paths $\rho$ considered in Corollary A.4. While $p_{\boldsymbol{\theta}}$ (by definition) factors across any single path $\rho$, these factorizations may be inconsistent across paths. Enforcing some explicit consistency remains open for future work.

*Proof of Theorem A.3.* The first step is the standard ELBO trick:

$$\ln p(\mathbf{x}^{\mathbf{k}^0}) = \ln \int_{\mathbf{x}^{\mathbf{k}^{1:N}}} p(\mathbf{x}^{\mathbf{k}^{0:N}}) \mathrm{d}\mathbf{x}^{\mathbf{k}^{1:N}}$$

$$= \ln \mathbb{E}_{\mathbf{x}^{\mathbf{k}^{1:N}} \sim q(\mathbf{x}^{\mathbf{k}^{1:N}} | \mathbf{x}^{\mathbf{k}^0})} \frac{p(\mathbf{x}^{\mathbf{k}^{0:N}})}{q(\mathbf{x}^{\mathbf{k}^{1:N}} \mid \mathbf{x}^{\mathbf{k}^0})}$$

$$\geq \mathbb{E}_{\mathbf{x}^{\mathbf{k}^{1:N}} \sim q(\mathbf{x}^{\mathbf{k}^{1:N}} | \mathbf{x}^{\mathbf{k}^0})} \ln \frac{p(\mathbf{x}^{\mathbf{k}^{0:N}})}{q(\mathbf{x}^{\mathbf{k}^{1:N}} \mid \mathbf{x}^{\mathbf{k}^0})}.$$

where the last step follows from Jensen's inequality.

We now expand

$$\ln \frac{p_{\boldsymbol{\theta}}(\mathbf{x}^{\mathbf{k}^{0:N}})}{q(\mathbf{x}^{\mathbf{k}^{1:N}} \mid \mathbf{x}^{\mathbf{k}^0})}$$

$$= \ln p_{\boldsymbol{\theta}}(\mathbf{x}^{\mathbf{k}^N}) + \ln \frac{p_{\boldsymbol{\theta}}(\mathbf{x}^{\mathbf{k}^0} \mid \mathbf{x}^{\mathbf{k}^1})}{q(\mathbf{x}^{\mathbf{k}^1} \mid \mathbf{x}^{\mathbf{k}^0})} + \sum_{j=1}^{N-1} \ln \frac{p_{\boldsymbol{\theta}}(\mathbf{x}^{\mathbf{k}^j} \mid \mathbf{x}^{\mathbf{k}^{j+1}})}{q(\mathbf{x}^{\mathbf{k}^{j+1}} \mid \mathbf{x}^{\mathbf{k}^j}, \mathbf{x}^{\mathbf{k}^0})} \quad \text{(Factoring, Definition A.2)}$$

$$= \ln p_{\boldsymbol{\theta}}(\mathbf{x}^{\mathbf{k}^N}) + \ln \frac{p_{\boldsymbol{\theta}}(\mathbf{x}^{\mathbf{k}^0} \mid \mathbf{x}^{\mathbf{k}^1})}{q(\mathbf{x}^{\mathbf{k}^1} \mid \mathbf{x}^{\mathbf{k}^0})} + \sum_{j=1}^{N-1} \ln \frac{p_{\boldsymbol{\theta}}(\mathbf{x}^{\mathbf{k}^j} \mid \mathbf{x}^{\mathbf{k}^{j+1}})}{q(\mathbf{x}^{\mathbf{k}^j} \mid \mathbf{x}^{\mathbf{k}^{j+1}}, \mathbf{x}^{\mathbf{k}^0})} + \ln \frac{q(\mathbf{x}^{\mathbf{k}^j} \mid \mathbf{x}^{\mathbf{k}^0})}{q(\mathbf{x}^{\mathbf{k}^{j+1}} \mid \mathbf{x}^{\mathbf{k}^0})} \quad \text{(Bayes' Rule on } q)$$

$$= \ln p_{\boldsymbol{\theta}}(\mathbf{x}^{\mathbf{k}^N}) + \ln \frac{p_{\boldsymbol{\theta}}(\mathbf{x}^{\mathbf{k}^0} \mid \mathbf{x}^{\mathbf{k}^1})}{q(\mathbf{x}^{\mathbf{k}^1} \mid \mathbf{x}^{\mathbf{k}^0})} + \ln \frac{q(\mathbf{x}^{\mathbf{k}^1} \mid \mathbf{x}^{\mathbf{k}^0})}{q(\mathbf{x}^{\mathbf{k}^N} \mid \mathbf{x}^{\mathbf{k}^0})} + \sum_{j=1}^{N-1} \ln \frac{p_{\boldsymbol{\theta}}(\mathbf{x}^{\mathbf{k}^j} \mid \mathbf{x}^{\mathbf{k}^{j+1}})}{q(\mathbf{x}^{\mathbf{k}^j} \mid \mathbf{x}^{\mathbf{k}^{j+1}}, \mathbf{x}^{\mathbf{k}^0})} \quad \text{(Telescoping)}$$

$$= \ln \frac{p_{\boldsymbol{\theta}}(\mathbf{x}^{\mathbf{k}^N})}{q(\mathbf{x}^{\mathbf{k}^N} \mid \mathbf{x}^{\mathbf{k}^0})} + \ln p_{\boldsymbol{\theta}}(\mathbf{x}^{\mathbf{k}^0} \mid \mathbf{x}^{\mathbf{k}^1}) + \sum_{j=1}^{N-1} \ln \frac{p_{\boldsymbol{\theta}}(\mathbf{x}^{\mathbf{k}^j} \mid \mathbf{x}^{\mathbf{k}^{j+1}})}{q(\mathbf{x}^{\mathbf{k}^j} \mid \mathbf{x}^{\mathbf{k}^{j+1}}, \mathbf{x}^{\mathbf{k}^0})}. \quad \text{(Canceling)}$$

We observe that $\ln p_{\boldsymbol{\theta}}(\mathbf{x}^{\mathbf{k}^N})$ and $\ln \frac{1}{q(\mathbf{x}^{\mathbf{k}^N}|\mathbf{x}^{\mathbf{k}^0})}$ do not depend on $\boldsymbol{\theta}$ (recall $p_{\boldsymbol{\theta}}(\mathbf{x}^{\mathbf{k}^N})$ is the distribution over noise), so taking an expectation over the $q(\cdot)$, we can regard these as a constant $C$. This yields

$$
\begin{aligned}
\ln p(\mathbf{x}^{\mathbf{k}^0}) &\geq C + \mathbb{E}_{\mathbf{x}^{\mathbf{k}^{1:N}} \sim q(\mathbf{x}^{\mathbf{k}^{1:N}}|\mathbf{x}^{\mathbf{k}^0})} \left[ \ln p_{\boldsymbol{\theta}}(\mathbf{x}^{\mathbf{k}^0} \mid \mathbf{x}^{\mathbf{k}^1}) + \sum_{j=1}^{N-1} \ln \frac{p_{\boldsymbol{\theta}}(\mathbf{x}^{\mathbf{k}^j} \mid \mathbf{x}^{\mathbf{k}^{j+1}})}{q(\mathbf{x}^{\mathbf{k}^j} \mid \mathbf{x}^{\mathbf{k}^{j+1}}, \mathbf{x}^{\mathbf{k}^0})} \right] \\
&= C + \mathbb{E}_{\mathbf{x}^{\mathbf{k}^{1:N}} \sim q(\mathbf{x}^{\mathbf{k}^{1:N}}|\mathbf{x}^{\mathbf{k}_0})} \left[ \ln p_{\boldsymbol{\theta}}(\mathbf{x}^{\mathbf{k}^0} \mid \mathbf{x}^{\mathbf{k}^1}) + \sum_{j=1}^{N-1} \mathrm{D}_{\mathbb{KL}}(p_{\boldsymbol{\theta}}(\mathbf{x}^{\mathbf{k}^j} \mid \mathbf{x}^{\mathbf{k}^{j+1}}) \parallel q(\mathbf{x}^{\mathbf{k}^j} \mid \mathbf{x}^{\mathbf{k}^{j+1}}, \mathbf{x}^{\mathbf{k}^0})) \right].
\end{aligned}
$$

$\square$

## A.2   Understanding Frequency Guidance

For simplicity, we focus on 1-dimensional discrete signals with even dimension $d$, but extending to 2-dimensions is straightforward. We provide a simple mathematical explanation that "noising" a feature corresponds to a form of low-pass filtering.

Specifically, we consider a regression setting with features $\mathbf{x} \in \mathbb{R}^d$ and targets $\mathbf{y} \in \mathbb{R}^m$. We now study the conditional distribution of $\mathbf{y} \mid \mathbf{x}_\sigma$, where $\mathbf{x}_\sigma = \mathbf{x} + \sigma \mathbf{z}$ is a noisy measurement of $\mathbf{x}$. To understand effects in the frequency domain, we study the conditional distribution of the Fourier transform of $\mathbf{y}$ given a measurement of $\mathbf{x}_\sigma$. We assume that the entries of $\mathbf{x}$ can be interpreted as entries in a sequence and we interpret this conditional distribution as a function of the Fourier transformation, $\mathcal{F}_d(\mathbf{x})$, of $\mathbf{x}$. Similarly, we define $\mathcal{F}_m(\mathbf{y})$. For simplicity, we focus on a 1-d Fourier transform, but analogous statements hold for 2-d features $\mathbf{x}$ (e.g. 2-d frames in a video).

We begin by recalling the Fourier transform of a vector.

**Definition A.6.** Let $\mathcal{F}_d : \mathbb{R}^d \to \mathbb{R}^d$ denote the (real) discrete Fourier transform, specified by

$$
\mathcal{F}_d(\mathbf{x})(k) = \begin{cases} \sum_{i=1}^d \mathbf{x}[i] \sin(ik/2\pi) & 1 \leq k \leq d/2 \\ \sum_{i=1}^d \mathbf{x}[i] \cos(ik/2\pi) & d/2 < k \leq 1 \end{cases} \tag{9}
$$

We note that, by Parseval's theorem, $\mathcal{F}_d$ is an isometry:

$$
\frac{1}{d}\|\mathcal{F}_d(\mathbf{x})\|_{\ell_2}^2 = \|\mathbf{x}\|_{\ell_2}^2 \tag{10}
$$

Because $\mathcal{F}_d$ is a bijective linear mapping, we identify it with an invertible matrix in $\mathbb{R}^{d \times d}$.

We now characterize the conditional of $\mathcal{F}_m(\mathbf{y}) \mid \mathcal{F}_d(\mathbf{x})$.

**Proposition A.7.** *Let $\mathbf{x} \sim \mathcal{N}(0, \Sigma_x^2)$, and $\mathbf{y} \mid \mathbf{x} \sim \mathcal{N}(\mathbf{Ax}, \Sigma_y^2)$. Define $\mathbf{x}_\sigma = \mathbf{x} + \sigma \mathbf{z}$, where $\mathbf{z} \sim \mathcal{N}(0, \mathbf{I})$ is independent of $\mathbf{x}, \mathbf{y}$. Define $\hat{\mathbf{A}} := \mathcal{F}_m \mathbf{A} \mathcal{F}_d^{-1}$, $\hat{\boldsymbol{\Sigma}}_x := \mathcal{F}_d \Sigma_x \mathcal{F}_d^\top$ and $\hat{\mathbf{S}}(\sigma) := \hat{\boldsymbol{\Sigma}}_x (\hat{\boldsymbol{\Sigma}}_x + d\sigma^2 \mathbf{I})^{-1}$, and $\hat{\boldsymbol{\Sigma}}_y := \mathcal{F}_m \Sigma_y \mathcal{F}_m^\top$. Then,*

- $\mathcal{F}_d(\mathbf{x}) \sim \mathcal{N}(0, \hat{\boldsymbol{\Sigma}}_x)$

- $\mathcal{F}_m(\mathbf{y}) \mid \mathcal{F}_d(\mathbf{x}) \sim \mathcal{N}(\hat{\mathbf{A}}\mathcal{F}_d(\mathbf{x}), \hat{\boldsymbol{\Sigma}}_y)$

- *The distribution of $\mathcal{F}_m(\mathbf{y}) \mid \mathbf{x}_\sigma$ (or $\mathcal{F}_m(\mathbf{y}) \mid \mathcal{F}_d(\mathbf{x}_\sigma)$) is*

$$
\mathcal{N}(\hat{\mathbf{A}}\hat{\mathbf{S}}(\sigma)\mathcal{F}_d(\mathbf{x}_\sigma), \hat{\boldsymbol{\Sigma}}_y + d\sigma^2 \hat{\mathbf{A}}\hat{\mathbf{S}}(\sigma)\hat{\mathbf{A}}^\top) \tag{11}
$$

*Proof.* Set $\hat{\mathbf{x}} = \mathcal{F}_d(\mathbf{x})$ and $\hat{\mathbf{y}} = \mathcal{F}_d(\mathbf{y})$. As $\mathcal{F}_d, \mathcal{F}_m$ are linear, we see that $\hat{\mathbf{x}} \sim \mathcal{N}(0, \hat{\boldsymbol{\Sigma}}_x^2)$ and $\hat{\mathbf{y}} \sim \mathcal{N}(\mathcal{F}_m \mathbf{A}(\mathbf{x}), \mathcal{F}_m \Sigma_y \mathcal{F}_m^\top) = \mathcal{N}(\hat{\mathbf{A}}\mathcal{F}_d(\mathbf{x}), \hat{\boldsymbol{\Sigma}}_y)$.

For the last statement, we have that $\mathcal{F}_d(\mathbf{x}_\sigma) = \hat{\mathbf{x}} + \sigma \mathcal{F}_d(\mathbf{z})$. As $\frac{1}{\sqrt{d}}\mathcal{F}_d$ is an isometry (i.e orthogonal), we have $\frac{1}{d}\mathbb{E}[\mathcal{F}_d(\mathbf{z})\mathcal{F}_d(\mathbf{z})^\top] = \mathbf{I}$. Thus, $\sigma \mathcal{F}_d(\mathbf{z}) = \sigma\sqrt{d}\hat{\mathbf{z}}$, where $\hat{\mathbf{z}} \sim \mathcal{N}(0, \mathbf{I}_d)$ is independent of $\hat{\mathbf{x}}, \hat{\mathbf{y}}$. We may now invoke Lemma A.8 to show that Equation (11) describes the distribution of $\mathcal{F}_m(\mathbf{y}) \mid \mathcal{F}_d(\mathbf{x}_\sigma)$. As $\mathcal{F}_d$ is a bijection, conditioning on $\mathcal{F}_d(\mathbf{x}_\sigma)$ and $\mathbf{x}_\sigma$ is equivalent. $\square$

**Interpretation in Terms for Frequency Attenuation:** It is common that natural signals exhibit power-law decay in the frequency domain. As an illustration, consider $\hat{\Sigma}_x = C\text{Diag}(\{i^{-\alpha}\}_{1 \leq i \leq d})$; that is, in the Fourier domain, $\mathbf{x}$ is independent across frequencies and exhibits a power-law decay with exponent $\alpha$. Then, $\hat{\mathbf{S}}(\sigma)$ is diagonal, and

$$\hat{\mathbf{S}}(\sigma)_{ii} = \frac{1}{1 + d\sigma^2 i^\alpha/C} \sim \begin{cases} 1 & i \leq (\frac{C}{d\sigma^2})^{1/\alpha} \text{ or } \sigma^2 \leq Ci^\alpha/d \\ i^{-\alpha} & i \geq (\frac{C}{d\sigma^2})^{1/\alpha} \text{ or } \sigma^2 \geq Ci^\alpha/d \end{cases}$$

also exhibits power law decay. Hence, when conditioning on $\mathbf{x}_\sigma$, the shrinkage operator $\hat{\Sigma}(\sigma)$ attenuates the contribution of the $i$-th frequency of $\mathbf{x}_\sigma$ in proportion to $i^{-\alpha}$ for $i$-large. Moreover, as $\sigma$ becomes larger, more frequencies are attenuated. In other words, conditioning on noisier examples leads to more aggressive attenuation.

Importantly, **there is no intrinsic bias of Gaussian noising towards preferring lower frequencies. Rather, noising serves to regularize away weaker frequencies. For natural images, this corresponds to high frequencies, but may not in other application domains.**

**Lemma A.8** (Gaussian Conditional Computation). *Let* $\mathbf{x} \sim \mathcal{N}(0, \Sigma_x^2)$, *and* $\mathbf{y} \mid \mathbf{x} \sim \mathcal{N}(\mathbf{A}\mathbf{x}, \Sigma_y^2)$. *Define* $\mathbf{x}_\sigma = \mathbf{x} + \sigma\mathbf{z}$, *where* $\mathbf{z} \sim \mathcal{N}(0, \mathbf{I})$ *is independent of* $\mathbf{x}, \mathbf{y}$. *Set* $\mathbf{S}(\sigma) := \Sigma_x(\Sigma_x + \sigma^2\mathbf{I})^{-1}$. *Then, the distribution of* $\mathbf{y} \mid \mathbf{x}_\sigma$ *is* $\mathcal{N}(\mathbf{A}\mathbf{S}(\sigma)\mathbf{x}_\sigma, \Sigma_y + \sigma^2\mathbf{A}\mathbf{S}(\sigma)\mathbf{A}^\top)$.

*Proof.* First, we observe that $(\mathbf{x}_\sigma, \mathbf{y})$ are jointly Gaussian random variables with mean zero. We set $\Sigma_{22} = \mathbb{E}[\mathbf{x}_\sigma^2] = \sigma^2\mathbf{I} + \Sigma_x$, and $\Sigma_{11} = \mathbb{E}[\mathbf{y}^2] = \Sigma_y + \mathbf{A}\Sigma_x\mathbf{A}^\top$. Moreover, $\Sigma_{12} := \mathbb{E}[\mathbf{y}\mathbf{x}_\sigma^\top] = \mathbf{A}\Sigma_x$. Hence, from the standard formula for Gaussian conditional distributions, we have

$$\mathbf{y} \mid \mathbf{x}_\sigma \sim \mathcal{N}\left(\Sigma_{12}\Sigma_{22}^{-1}\mathbf{x}_\sigma, \Sigma_{11} - \Sigma_{12}\Sigma_{22}^{-1}\Sigma_{12}\right)$$
$$= \mathcal{N}\left(\mathbf{A}\Sigma_x(\Sigma_x + \sigma^2\mathbf{I})^{-1}\mathbf{x}_\sigma, \Sigma_y + \mathbf{A}\Sigma_x\mathbf{A}^\top - \mathbf{A}\Sigma_x(\Sigma_x + \sigma^2\mathbf{I})^{-1}\Sigma_x\mathbf{A}^\top\right).$$

We may then simplify $\mathbf{A}\Sigma_x\mathbf{A}^\top - \mathbf{A}\Sigma_x(\Sigma_x + \sigma\mathbf{I})^{-1}\Sigma_x\mathbf{A}^\top = \mathbf{A}(\Sigma_x - \Sigma_x(\Sigma_x + \sigma\mathbf{I})^{-1}\Sigma_x)\mathbf{A}^\top$. Note that $(\Sigma_x - \Sigma_x(\Sigma_x + \sigma^2\mathbf{I})^{-1}\Sigma_x) = (\Sigma_x - \Sigma_x(\Sigma_x + \sigma\mathbf{I})^{-1}(\Sigma_x + \sigma^2\mathbf{I}) - \Sigma_x(\Sigma_x + \sigma^2\mathbf{I})^{-1}\sigma^2\mathbf{I}) = \sigma^2\Sigma_x(\Sigma_x + \sigma^2\mathbf{I})^{-1}$. Define $\mathbf{S}(\sigma) := \Sigma_x(\Sigma_x + \sigma^2\mathbf{I})^{-1}$. We conclude that

$$\mathbf{y} \mid \mathbf{x}_\sigma \sim \mathcal{N}(\mathbf{A}\mathbf{S}(\sigma)\mathbf{x}_\sigma, \Sigma_y + \sigma^2\mathbf{A}\mathbf{S}(\sigma)\mathbf{A}^\top), \tag{12}$$

$\square$

## A.3 A Maximum Likelihood Interpretation for Score Addition.

The Diffusion Forcing Transformer achieves history guidance across time and frequency by sampling with linearly weighted diffusion scores conditioned on different history lengths. Though this appears to be purely heuristic, as in classifier-free guidance, we provide a meaningful probabilistic interpretation of the algorithm.

**Intuition for guidance via Gaussian MLE.** We begin by justifying linearly combining scores in simple Gaussian models. For now, let us assume that the goal is to sample $\mathbf{x} \sim q^\star(\mathbf{x})$, and the aim is to estimate the score $s^\star(\mathbf{x}) = \nabla_\mathbf{x} \ln q(\mathbf{x})$.

We make a strong assumption that we have $N$ estimators for the score functions, $(\hat{s}_i(\mathbf{x}))_{1 \leq i \leq n}$, and that errors are Gaussian.

**Assumption A.9** (Gaussian Errors). We assume that, conditioned on $\mathbf{x}$, the errors $\vec{\epsilon} := (\hat{s}_1(\mathbf{x}) - s^\star(\mathbf{x}), \hat{s}_2(\mathbf{x}) - s^\star(\mathbf{x}), \ldots, \hat{s}_n(\mathbf{x}) - s^\star(\mathbf{x}))$ form a Gaussian vector with mean zero and covariance $\Sigma(\mathbf{x}) \in \mathbb{R}^{dn \times dn}$.

Though the assumption is clearly not true in practice, it helps build intuition for the idea. Moreover, given that the reverse process of an SDE essentially involves Gaussian predictions, it is plausible to expect that the individual steps of the denoising process model Gaussian distributions, and consequently, errors are "Gaussian-like" (Huang et al., 2023) .

Let us now consider the maximum likelihood score estimator in this model. We introduce the notation

$$\mathbb{I}^\top = [\mathbf{I}_{d \times d}^\top, \mathbf{I}_{d \times d}^\top \ldots \mathbf{I}_{d \times d}^\top]^\top. \tag{13}$$

In this case, we have

$$\hat{\mathbf{s}}_{1:n}(\mathbf{x}) = (\hat{s}_1(\mathbf{x}), \hat{s}_2(\mathbf{x}), \ldots, s_n(\mathbf{x})) \mid \mathbf{x} \sim \mathcal{N}(\mathbb{I}s^\star(\mathbf{x}), \Sigma(\mathbf{x})). \tag{14}$$

Let us now characterize the maximum likelihood estimator, $\hat{s}^{\mathrm{MLE}}$. This solves

$$
\begin{aligned}
\hat{s}^{\mathrm{MLE}}(\mathbf{x}) &= \mathrm{argmax}_{s(\mathbf{x})} p(\hat{s}_{1:n}(\mathbf{x}); s(\mathbf{x})) \\
&= \arg\max_{s(\mathbf{x})} \frac{1}{\sqrt{(2\pi)^k |\boldsymbol{\Sigma}|}} \exp\left(-\frac{1}{2}\vec{\epsilon}^{\top}\boldsymbol{\Sigma}(\mathbf{x})^{-1}\vec{\epsilon}(\mathbf{x})\right) && (\vec{\epsilon} = \hat{\mathbf{s}}_{1:n} - \mathbb{I}s(\mathbf{x})) \\
&= \min_{s(\mathbf{x})} \frac{1}{2}\vec{\epsilon}^{\top}\boldsymbol{\Sigma}(\mathbf{x})^{-1}\vec{\epsilon}(\mathbf{x}) && (\vec{\epsilon} = \hat{\mathbf{s}}_{1:n} - \mathbb{I}s^{\star}(\mathbf{x})) \\
&= \arg\min_{s(\mathbf{x})}(\hat{\mathbf{s}}_N(\mathbf{x}) - \mathbb{I}s^{\star}(\mathbf{x}))^{\top}\boldsymbol{\Sigma}(\mathbf{x})^{-1}(\hat{\mathbf{s}}_N(\mathbf{x}) - \mathbb{I}s^{\star}(\mathbf{x})).
\end{aligned}
$$

An exercise in Calculus reveals that

$$
\hat{s}^{\mathrm{MLE}}(\mathbf{x}) = \left(\mathbb{I}^{\top}\Sigma(\mathbf{x})^{-1}\mathbb{I}\right)^{-1}\left(\mathbb{I}^{\top}\Sigma(\mathbf{x})^{-1}\right)\hat{\mathbf{s}}_{1:n}(\mathbf{x}). \tag{15}
$$

In other words, $\hat{s}^{\mathrm{MLE}}$ is some ($\mathbf{x}$-dependent) linear function of $\hat{\mathbf{s}}_{1:n}$.

We now describe a couple of special cases:

**Case 1: $d = 1$ ($\mathbf{x}$ is scalar) scores are independent.** In this case, $\boldsymbol{\Sigma}(\mathbf{x})$ has a diagonal inverse, and by positive definiteness, its entries are strictly positive. Thus, letting $\alpha_i$ denote the diagonal entries of $\boldsymbol{\Sigma}(\mathbf{x})^{-1}$, we have $\mathbb{I}^{\top}\boldsymbol{\Sigma}(\mathbf{x})^{-1}$ is a vector with strictly positive entries $(\alpha_1(\mathbf{x}), \ldots, \alpha_n(\mathbf{x}))$, and $\mathbb{I}^{\top}\boldsymbol{\Sigma}(\mathbf{x})^{-1}\mathbb{I} = \sum_{i=1}^{n}\alpha_i(\mathbf{x})$ is their sum.

In this case,

$$
\hat{s}^{\mathrm{MLE}}(\mathbf{x}) = \sum_{i=1}^{n} \frac{\alpha_i}{\left(\sum_j \alpha_j(\mathbf{x})\right)}\hat{s}_i(\mathbf{x}) \tag{16}
$$

is a convex combination of the various scores.

**Case 2: general $d$ ($\mathbf{x}$ is scalar) scores are independent, and the errors $\hat{s}_i - s^{\star}$ have scaled identity covariance.** In this case, $\boldsymbol{\Sigma}(\mathbf{x})$ is block diagonal with scaled-indenity blocks, so we can also show

$$
\hat{s}^{\mathrm{MLE}}(\mathbf{x}) = \sum_{i=1}^{n} \frac{\alpha_i(\mathbf{x})}{\left(\sum_j \alpha_j(\mathbf{x})\right)}\hat{s}_i(\mathbf{x}), \tag{17}
$$

where $\alpha_i^{-1}$ are the scalings of the identity blocks.

Now we can examine the specific case of history guidance. Let the $n$ pieces of evidences be the $n$ different history segments of different lengths that our model condition on. Diffusion Forcing Transformer is essentially trying to combine these evidences with Maximum A Posteriori (MAP) to get an overall estimation of the score of future tokens.

**Why MLE / Averaging Works in General?** Though the averages derived above hold for Gaussian case, there is a very general theory for combining multiple estimators into one called *Optimal Aggregation of Estimators* (see, e.g. (Rigollet & Tsybakov, 2007)). In this case, even beyond Gaussian settings, there are known benefits to optimizing over the convex hull of a family of estimators rather than choosing the best single one (see, e.g. (Bellec, 2017)). Another rational for combining estimators is that an average of $n$ estimators can do better than the best single estimator. Indeed, suppose that you have $n$ maps $\hat{s}_i : \mathbf{x} \in \mathcal{X} \to [0,1]$, and assume that the optimal value (for simplicity) is $s_i^{\star}(\mathbf{x}) = 0$ (also, scalar for simplicity). Suppose you partition the $\mathbf{x}$ space into $n$ components $\mathcal{X}_1, \ldots, \mathcal{X}_n$ such that

$$
\Pr[\mathbf{x} \in \mathcal{X}_i] = \frac{1}{n}, \quad \hat{s}_i(\mathbf{x}) = \begin{cases} 1 & \mathbf{x} \in \mathcal{X}_i \\ 0 & \text{otherwise} \end{cases} \tag{18}
$$

For any estimator, the expected square error is then

$$
\mathbb{E}[(\hat{s}_i)^2] = \mathbb{P}[\mathbf{x} \in \mathcal{X}_i] = \frac{1}{n}. \tag{19}
$$

---

**Algorithm 1** Flexible Sampling with DFoT and (optionally) History Guidance

---

**Task:** specified by indices $\mathcal{H}$, $\mathcal{G} = \mathcal{T} \setminus \mathcal{H}$, and history frames $\mathbf{x}_{\mathcal{H}}$.
**Input:** diffusion process defined by $\alpha_k, \sigma_k$, diffusion sampler $\mathcal{S}$ with sampling steps $N$,
**DFoT** model $\mathbf{s}_{\boldsymbol{\theta}}(\cdot, \cdot)$, and **History Guidance** scheme specified by $\{(\mathcal{H}_i, k_{\mathcal{H}_i}, \omega_i)\}_{i=1}^I$.
Sample $\mathbf{x}_{\mathcal{G}} \sim \mathcal{N}(0, I)$, then $\mathbf{x}_{\mathcal{T}} \leftarrow \mathbf{x}_{\mathcal{H}} \oplus \mathbf{x}_{\mathcal{G}}$      ▷ Sample random noise for generation frames
**for** $n = N, N-1, \ldots, 1$ **do**

$\quad k_{\mathcal{T}} \leftarrow (k_t)_{t=1}^T$ where $\begin{cases} k_t = \frac{n}{N} & \text{if } t \in \mathcal{G} \\ k_t = 1 & \text{if } t \in \mathcal{H} \end{cases}$

$\quad \hat{\mathbf{x}}_{\mathcal{T}} \leftarrow \mathbf{x}_{\mathcal{T}}$, then *replace* $\hat{\mathbf{x}}_{\mathcal{H}} \leftarrow \boldsymbol{\epsilon}$ where $\boldsymbol{\epsilon} \sim \mathcal{N}(0, I)$      ▷ Fully mask history
$\quad \hat{\mathbf{s}}^{\varnothing} \leftarrow \mathbf{s}_{\boldsymbol{\theta}}(\hat{\mathbf{x}}_{\mathcal{T}}, k_{\mathcal{T}})$      ▷ Estimate unconditional score
$\quad$ **for** $i = 1, \ldots, I$ **do**

$\quad\quad k_{\mathcal{T}} \leftarrow (k_t)_{t=1}^T$ where $\begin{cases} k_t = \frac{n}{N} & \text{if } t \in \mathcal{G} \\ k_t = k_{\mathcal{H}_i} & \text{if } t \in \mathcal{H}_i \\ k_t = 1 & \text{if } t \in \mathcal{H} \setminus \mathcal{H}_i \end{cases}$

$\quad\quad \hat{\mathbf{x}}_{\mathcal{T}} \leftarrow \mathbf{x}_{\mathcal{T}}$, then *replace* $\begin{cases} \hat{\mathbf{x}}_{\mathcal{H}_i} \leftarrow \alpha_{k_{\mathcal{H}_i}} \hat{\mathbf{x}}_{\mathcal{H}_i} + \sigma_{k_{\mathcal{H}_i}} \boldsymbol{\epsilon} \text{ where } \boldsymbol{\epsilon} \sim \mathcal{N}(0, I) \\ \hat{\mathbf{x}}_{\mathcal{H} \setminus \mathcal{H}_i} \leftarrow \boldsymbol{\epsilon} \text{ where } \boldsymbol{\epsilon} \sim \mathcal{N}(0, I) \end{cases}$    ▷ Mask history based on $\mathcal{H}_i$ and $k_{\mathcal{H}_i}$

$\quad\quad \hat{\mathbf{s}}^i \leftarrow \mathbf{s}_{\boldsymbol{\theta}}(\hat{\mathbf{x}}_{\mathcal{T}}, k_{\mathcal{T}})$      ▷ Estimate $i$-th conditional score
$\quad$ **end for**
$\quad \hat{\mathbf{s}} \leftarrow \hat{\mathbf{s}}^{\varnothing} + \sum_{i=1}^I \omega_i \cdot (\hat{\mathbf{s}}^i - \hat{\mathbf{s}}^{\varnothing})$      ▷ Compose scores
$\quad \mathbf{x}_{\mathcal{G}} \leftarrow \mathcal{S}(\mathbf{x}_{\mathcal{G}}, \hat{\mathbf{s}}_{\mathcal{G}}; \frac{n}{N}, \frac{n-1}{N})$      ▷ Denoise $k = \frac{n}{N} \rightarrow \frac{n-1}{N}$
**end for**
**Output:** $\mathbf{x}_{\mathcal{G}}$

---

However, for any $\mathbf{x}$, $\frac{1}{n} \sum_{i=1}^n \hat{s}_i(\mathbf{x}) = \frac{1}{n} \sum_i^n \mathbb{I}(x \in \mathcal{X}_i) = \frac{1}{n}$. Thus,

$$\mathbb{E}\left[\left(\frac{1}{n} \sum_{i=1}^n \hat{s}_i\right)^2\right] = \mathbb{P}[\mathbf{x} \in \mathcal{X}_i] = \frac{1}{n^2}. \tag{20}$$

Because estimators make errors on complementary regions of state space, they work in concert to cancel out errors to reduce overall error.

We suspect history guidance functions in a similar fashion: though attending to different history contexts may result in errors for different realizations of past frames, but by averaging all these effects out, we ameliorate total error.

### A.4 Sampling with DFoT and History Guidance

DFoT is capable of flexible sampling conditioning on *arbitrary history*, and is further capable of performing *history guidance*, a family of guidance methods we propose. In Algorithm 1, we provide a detailed sampling procedure for DFoT and history guidance, where any score-based sampler such as DDPM (Ho et al., 2020) or DDIM (Song et al., 2020) can be used for $\mathcal{S}$. Importantly, when estimating a score conditioned on a masked history, it is crucial to pass the corresponding noise levels $k_{\mathcal{T}}$ and to *replace* the clean history frames with noisy frames, which are created by diffusing the clean history to the noise levels. This ensures that the model input is consistent with what it encounters during training time. Note that Algorithm 1 can be applied given arbitrary history frames. For instance, to *extrapolate* the history of length $\tau$ to $T$ frames, set $\mathcal{H} = \{1, \ldots, \tau\}$ and $\mathcal{G} = \{\tau + 1, \ldots, T\}$; to *interpolate* between two frames, set $\mathcal{H} = \{1, T\}$ and $\mathcal{G} = \{2, \ldots, T - 1\}$. Below we provide several representative examples of how the algorithm is applied:

- **Conditional Sampling without History Guidance**: $\{(\mathcal{H}_i, k_{\mathcal{H}_i}, \omega_i)\}_{i=1}^I = \{(\mathcal{H}, 0, 1)\}$

- **Vanilla History Guidance** with a guidance scale $\omega > 1$: $\{(\mathcal{H}_i, k_{\mathcal{H}_i}, \omega_i)\}_{i=1}^I = \{(\mathcal{H}, 0, \omega)\}$

- **Temporal History Guidance** with $I$ subsequences $\{\mathcal{H}_i\}_{i=1}^I$ and guidance scales $\{\omega_i\}_{i=1}^I$: $\{(\mathcal{H}_i, k_{\mathcal{H}_i}, \omega_i)\}_{i=1}^I = \{(\mathcal{H}_i, 0, \omega_i)\}_{i=1}^I$

- **Fractional History Guidance** with a guidance scale $\omega$ and fractional masking level $k_{\mathcal{H}}$: $\{(\mathcal{H}_i, k_{\mathcal{H}_i}, \omega_i)\}_{i=1}^I = \{(\mathcal{H}, 0, 1), (\mathcal{H}, k_{\mathcal{H}}, \omega - 1)\}$

## A.5 Simplifying Training Objective

Diffusion Forcing (Chen et al., 2024) proposes to train the entire sequence with independent noise per frame. A natural question to ask is whether this mixed objective includes too many tasks compared to what one actually needs. Here we provide some insights from our experiments throughout the project: When the number of frames is small e.g. 10 latent frames, there is no noticeable decrease in training efficiency - Diffusion Forcing seems to converge as fast as standard diffusion from both training and validation curves. However, when we grow the number of latent frames to 50, we start to witness decreased performance at sampling time. While we firmly believe that binary dropout is not the ideal way to achieve objective reduction from our experiments, we believe that one can easily reduce our training objective by only applying independent noise up to the maximum training length one wants to support. In particular, if one wants to generate the next 10 frames from previous $1 - 10$ frames, it doesn't seem necessary for frame 11 to be independently masked as noise from time to time, since we will never need to mask it out for flexible conditioning. In addition, one may want to consider treating the number of history frames as a random variable at training time, sampling a length first and then applying uniform levels of masking to the history, though independent from the noise level of the generation target. We didn't investigate these simplifications in detail because we simply find Diffusion Forcing's training objective very versatile for many of the tasks we want to do, e.g. interpolation, and varying noise level sampling. However, we do believe that these schemes could worth more exploration if one is to scale up our method to a much bigger number of context frames.

## A.6 Causal Variant

In principle, one can implement DFoT and History Guidance with a causal transformer as well. For example, CausVid (Yin et al., 2024) has proved the effectiveness of Diffusion Forcing on fast causal video synthesis and doesn't conflict with History Guidance. However, we'd like to highlight that one can also use our non-causal DFoT to achieve causal sampling. Different from traditional transformer-based models, DFoT doesn't need to enforce an attention mask to achieve causality. Instead, at generation time, one can mask out the future with white noise to prevent any information from the future from leaking into the neural network. In fact, there might be use cases when one may want some low-frequency information from the future, and then one can fractionally mask out the future via noise as masking to achieve so. On the other hand, the motivation behind causal video diffusion models is often speed and real-time generation using KV caching. In that case, one either needs to train a causal DFoT directly or consult advanced techniques like attention sink (Xiao et al., 2024) to perform windowed attention effectively.

## A.7 Incorporating Other Conditioning

Throughout our discussions in the main paper, conditioning is history exclusively. What if one wants to integrate the Diffusion Forcing Transformer into a text-conditioned diffusion model? One claim of the DFoT is that it doesn't require architectural changes so one can fine-tune an existing model into a DFoT model. This is still the case here: if one already has a text-conditioned video diffusion model, presumably built to accept such conditioning via an adaptive layer norm, one simply take DFoT as an add on to their existing architecture to obtain a DFoT model that accepts both text and history as conditioning. DFoT's Figure 2 does not assert that one cannot use an external AdaLN layer with DFoT, but is rather saying no architectural changes is needed.

## A.8 Extended Temporal History Guidance

Temporal history guidance addresses the challenge of out-of-distribution (OOD) history by composing scores conditioned on different, shorter history subsequences, which are closer to being in-distribution. However, since the model receives the entire video sequence as input during sampling—including both the history and the noisy frames being generated—the OOD problem can arise throughout the entire video sequence, not just in the history portion. To mitigate this, we propose further decomposing the generation $\mathcal{G}$ into generation subsequences $\mathcal{G}_1, \mathcal{G}_2, \ldots, \mathcal{G}_J \subset \mathcal{G}$. In line with the original temporal history guidance, the history $\mathcal{H}$ is already decomposed into history subsequences $\mathcal{H}_1, \mathcal{H}_2, \ldots, \mathcal{H}_I \subset \mathcal{H}$. This allows us to compose scores conditioned on even shorter, and thus more in-distribution, subsequences in $\{\mathcal{H}_i\}_{i=1}^I \times \{\mathcal{G}_j\}_{j=1}^J$. Specifically, the composed score is given by:

$$\bigoplus_{j=1}^J \sum_{i=1}^I \nabla \log p_k(\mathbf{x}_{\mathcal{G}_j}^k | \mathbf{x}_{\mathcal{H}_i}) \tag{21}$$

where $\bigoplus$ denotes a frame-wise averaging operation. We refer to this method as *Extended Temporal History Guidance*, as it extends the concept of temporal history guidance by composing both history and generation subsequences. Empirically, we find this method to be more effective than the original temporal history guidance when the video sequence is clearly OOD (e.g., RealEstate10K OOD history experiment), and thus requires shorter subsequences to be in-distribution.

# Supplementary Visuals

Before delving into further details, we list extensive figures (Figures 8 to 14) that supplement the main paper's content. Detailed descriptions for these figures can be found in Appendix D.

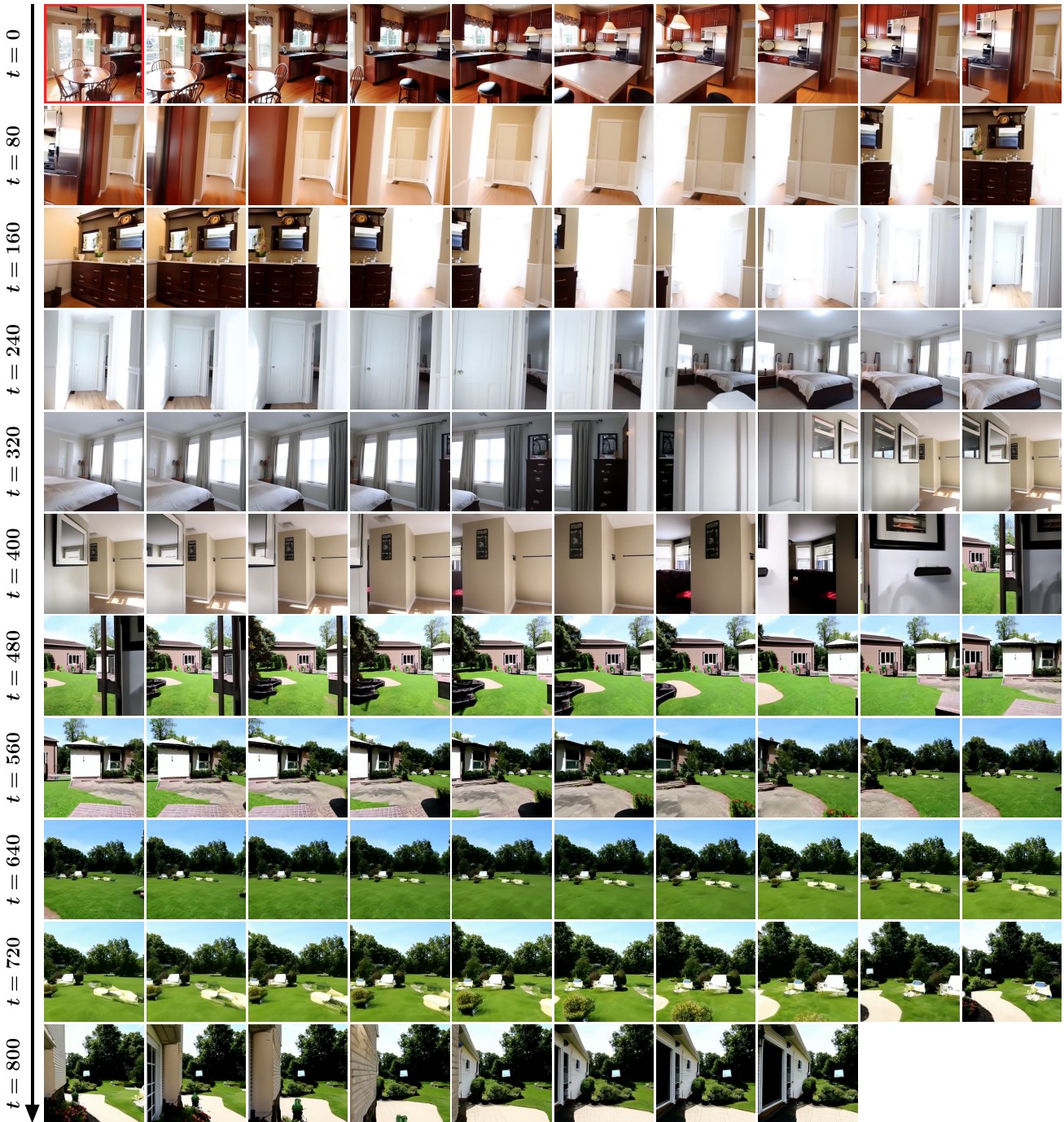

(a) **Long navigation video generated by DFoT with HG.** # frames = 862.

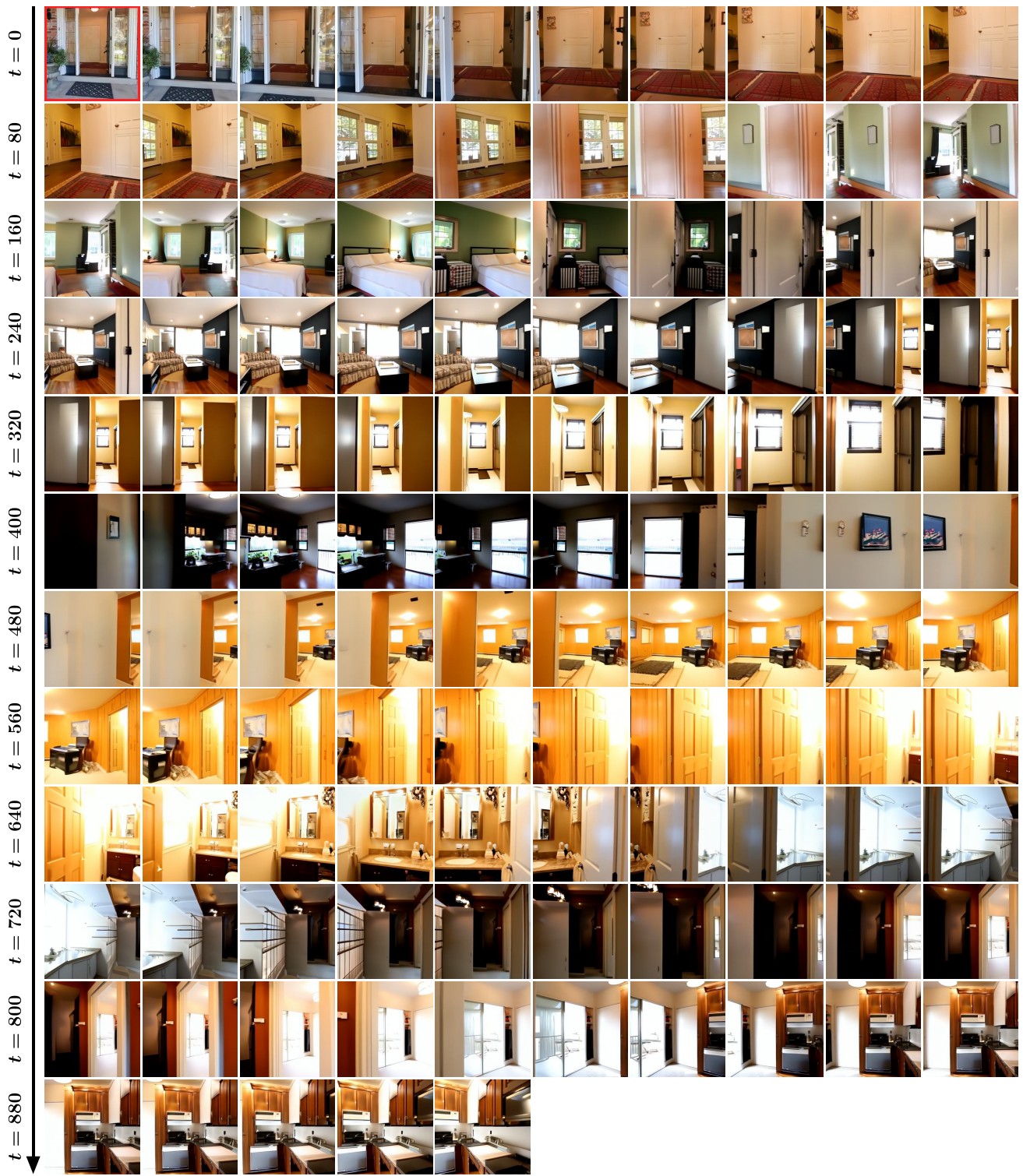

(b) **Long navigation video generated by DFoT with HG.** # frames = 917.

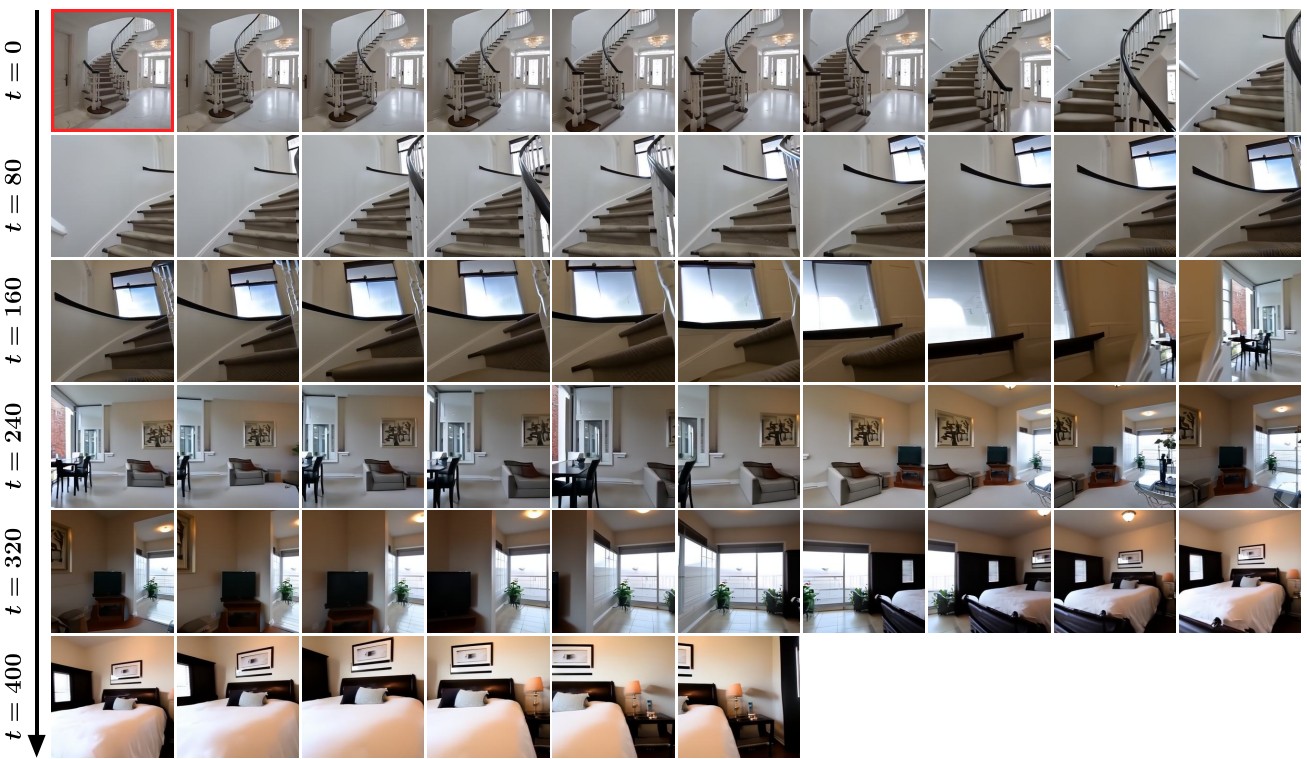

(c) **Long navigation video generated by DFoT with HG.** # frames = 442.

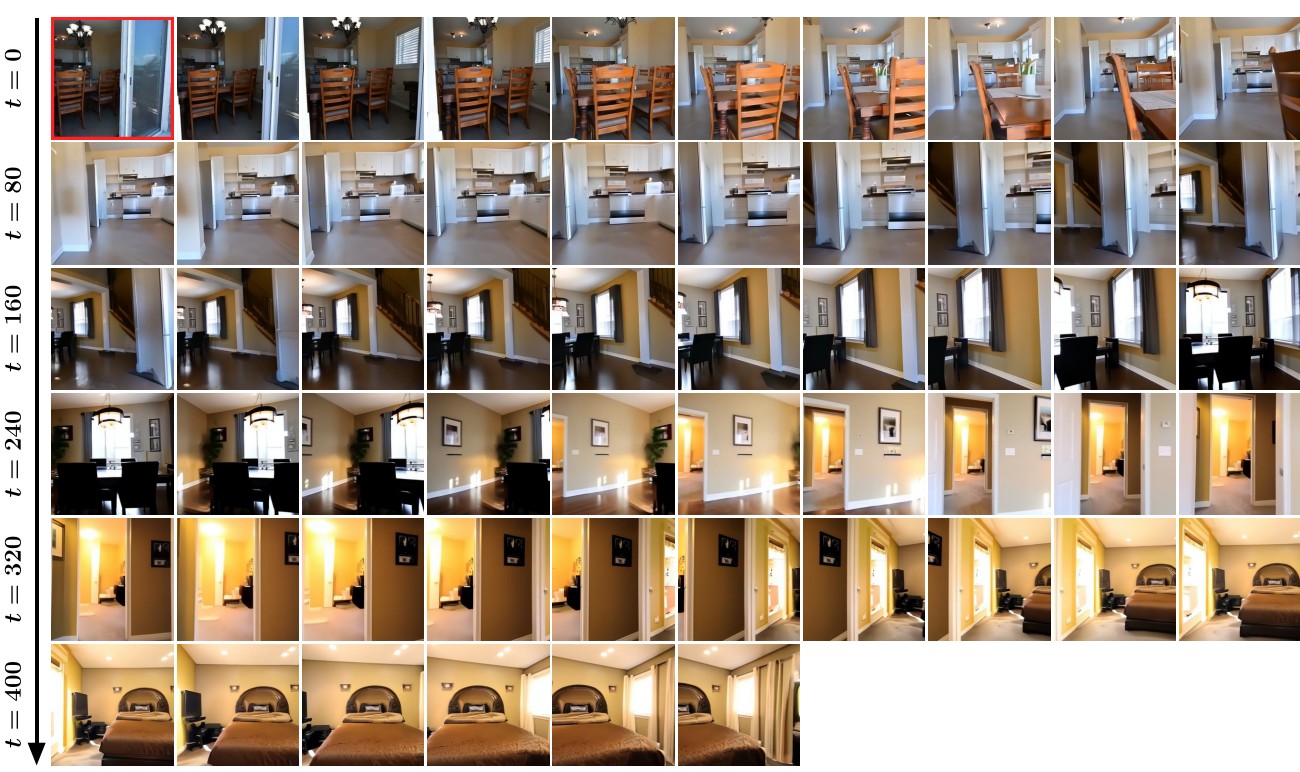

(d) **Long navigation video generated by DFoT with HG.** # frames = 442.

*Figure 8.* **Long navigation videos generated by DFoT with HG-v and HG-f, from a single history frame on RealEstate10K.** We subsample with a stride of 8 frames for visualization. The videos exhibit consistent transitions navigating while through diverse indoor and outdoor scenes, maintaining high stability over hundreds of frames. This is enabled by the improved sample quality and consistency from HG, along with DFoT's flexibility that allows both interpolation and extrapolation.

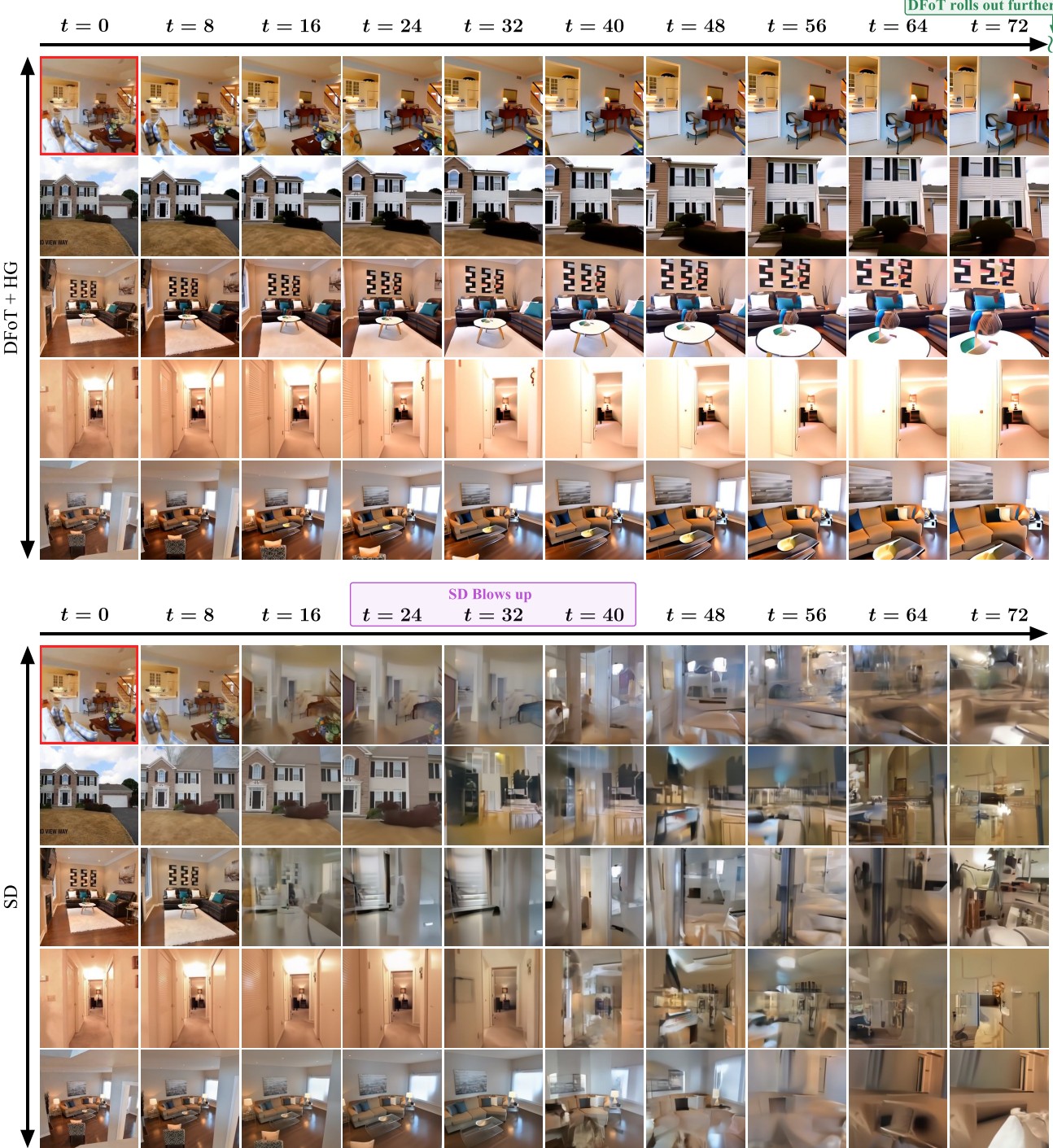

*Figure 9.* **Qualitative comparison of DFoT with HG vs. SD on long video generation.** Given a single history frame, we task both models to generate videos of moving straight ahead and visualize them with a stride of 8 frames. While SD quickly diverges after $t \approx 30$ frames, DFoT with HG maintains high stability until $t = 72$ and can roll out further.

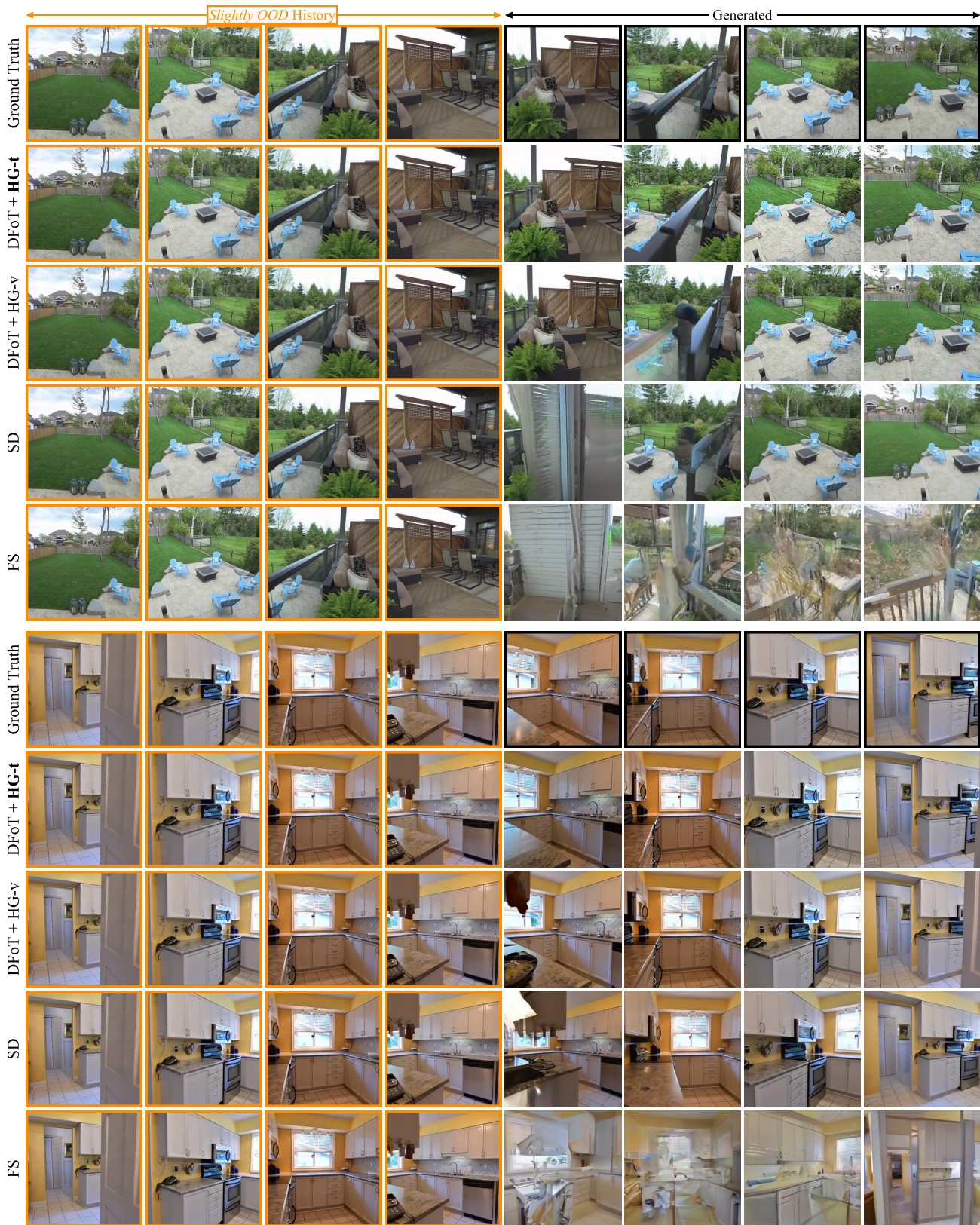

(a) Given slightly OOD history with rotation angles in $[120°, 130°]$, baselines and DFoT with HG-v generate inconsistent frames with artifacts. In contrast, DFoT with HG-t generates consistent videos that highly resemble the ground truth. This is the region where HG-t starts showing its generalization gap with other methods.

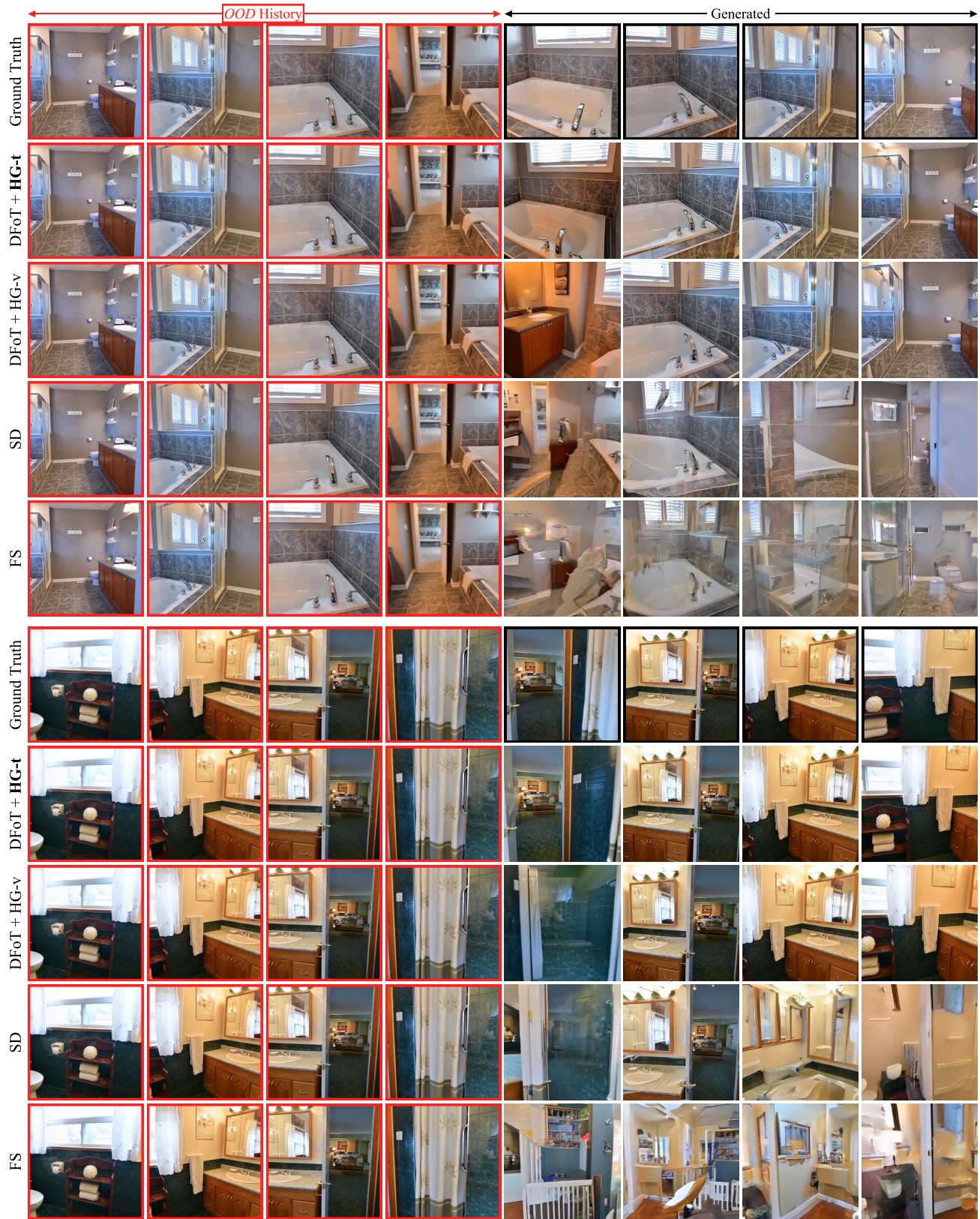

(b) Given *OOD* history, all baselines completely fail yet DFoT with HG-t still manages to generate high-quality, accurate videos.

*Figure 10.* **Qualitative results of testing robustness to out-of-distribution history on RealEstate10K.** We provide wide-angle, 4-frame history and task the models to generate the next 4 frames that interpolate between the history frames. As the angle increases, the history becomes more out-of-distribution, and thus we split the results into slightly OOD and OOD depending on the angle range.

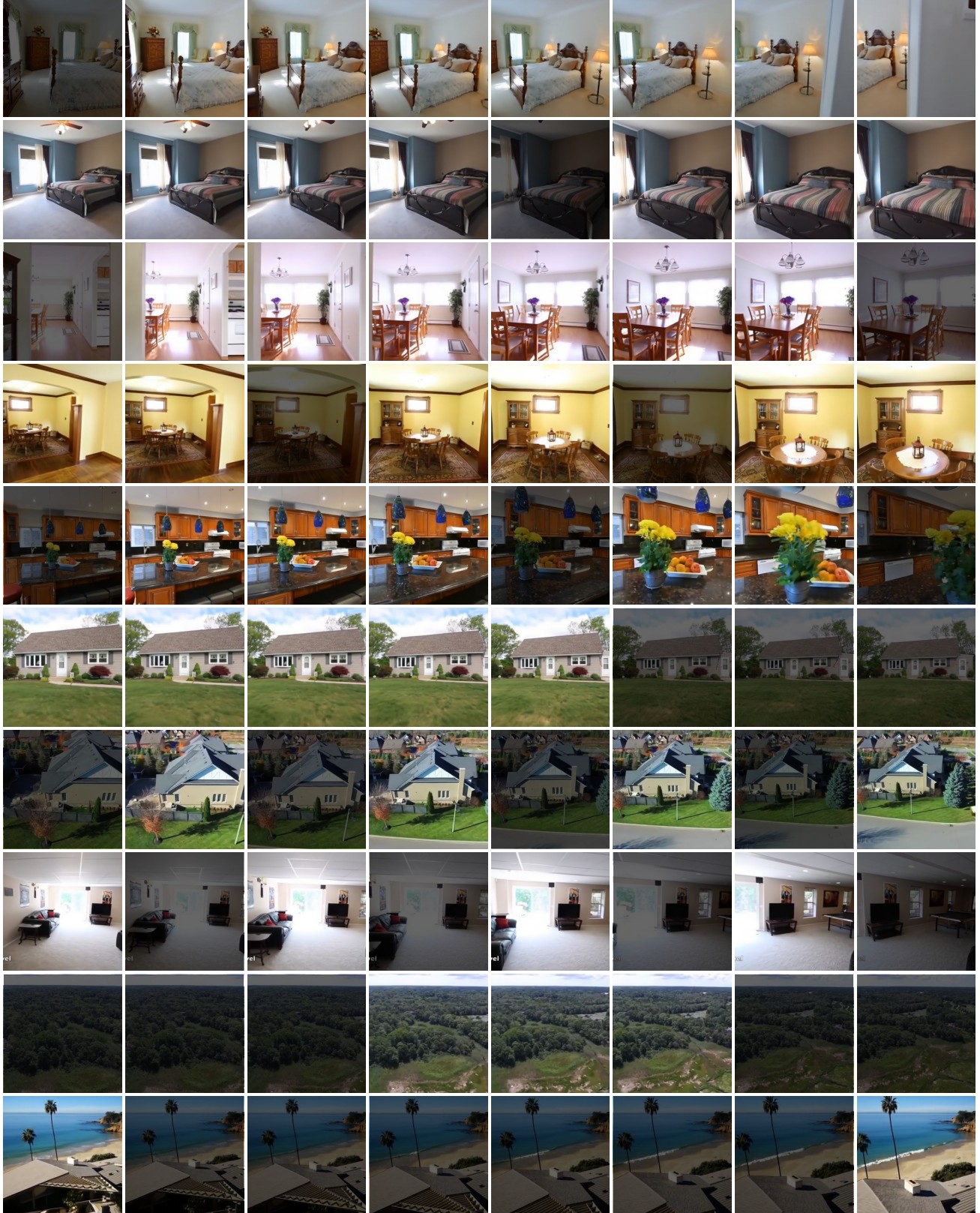

*Figure 11.* **An illustration of the empirical flexibility of DFoT**, showing ten samples from RealEstate10K, where a *single DFoT model* infills the missing frames given different history. DFoT successfully generates consistent samples across ten diverse tasks, each varying in the history length from 1 to 6 frames and at different timestamps.

*Extrapolation*

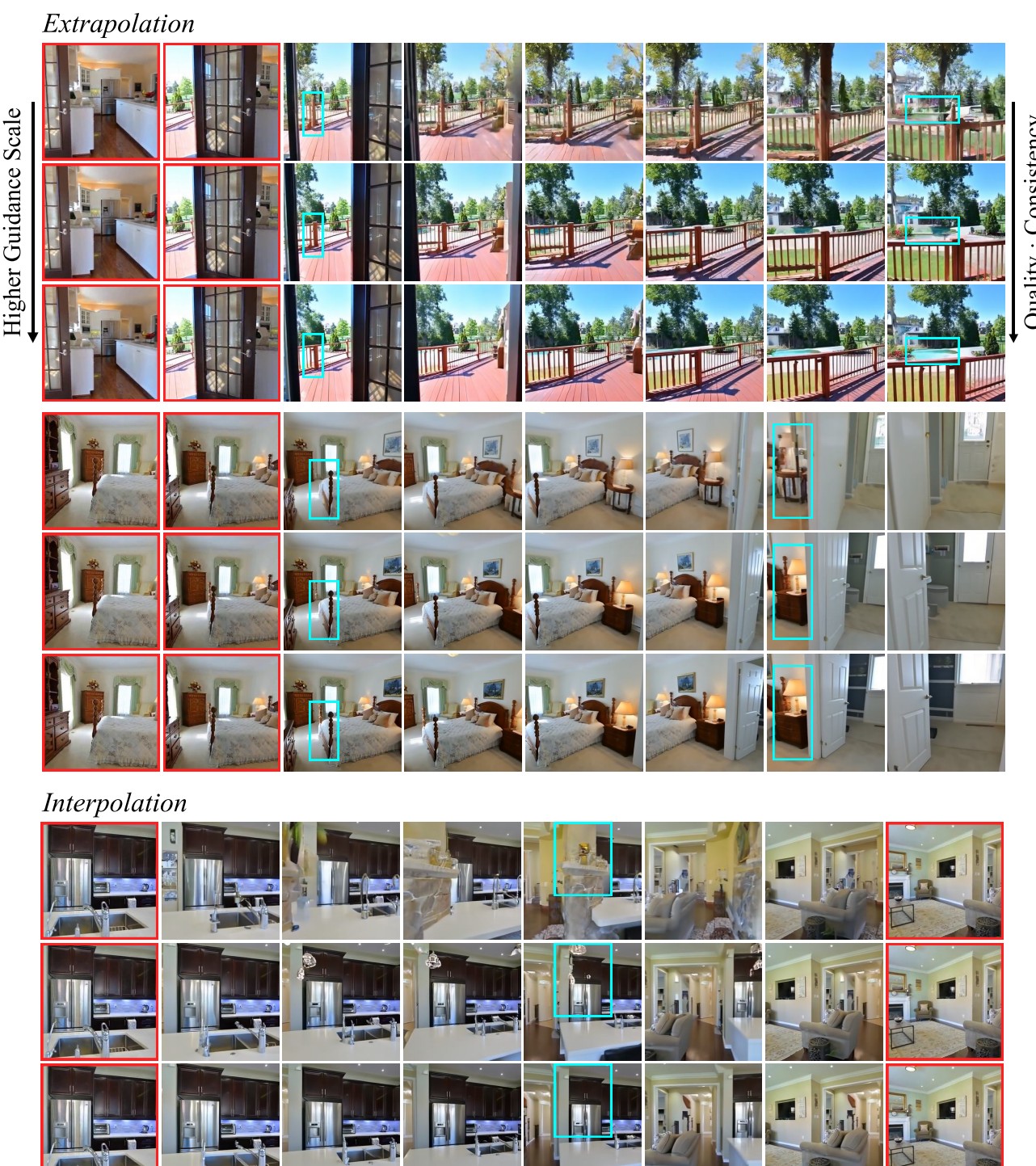

*Interpolation*

Figure 12. **Improved video generation quality with vanilla history guidance on RealEstate10K**, for both *extrapolation* and *interpolation* tasks. HG-v, with an increasing guidance scale, enhances fidelity and consistency while effectively removing artifacts. Videos are sampled conditioned on two history frames, with varying guidance scales $\omega = 1$ (top, *without HG-v*), 2 (middle), and 3 (bottom). *Zoom into the boxed regions to see notable differences.*

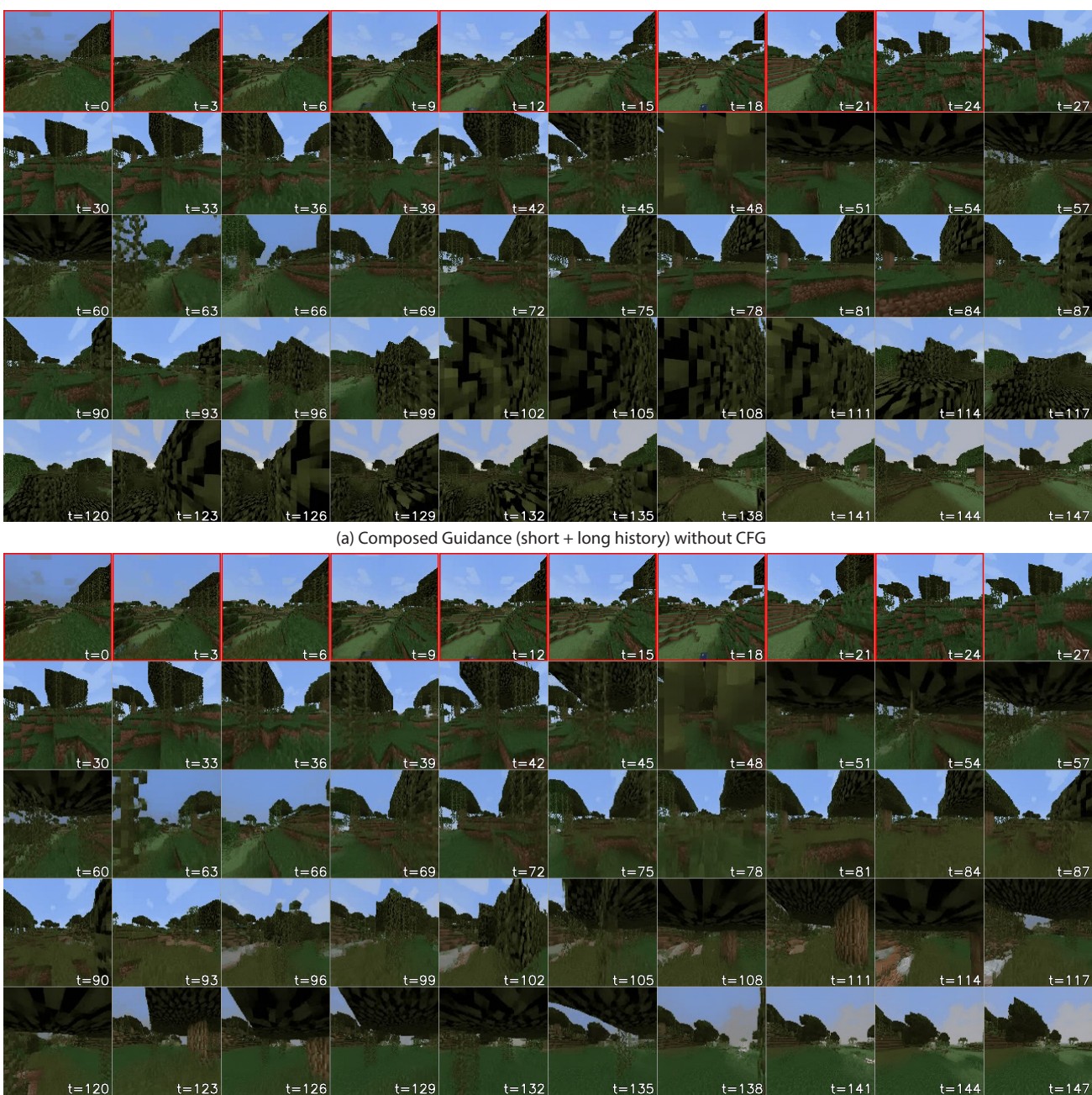

(a) Composed Guidance (short + long history) without CFG

(b) Conditional Generation (long-history only) without CFG

*Figure 13.* **Visualization of long context generation on Minecraft.** We visualize the generation up to the maximum length of the training set. Given 25 initial frames (red), DFoT with temporal history guidance (upper) can roll out stably without blowing up even without CFG. In contrast, one can clearly see that without temporal history guidance (lower), conditional generation easily becomes blurry in later frames. This is likely because the shorter-context model is less likely to fall out of distribution, using its generation power to compensate for the unconfident, blurry prediction from the longer-context model.

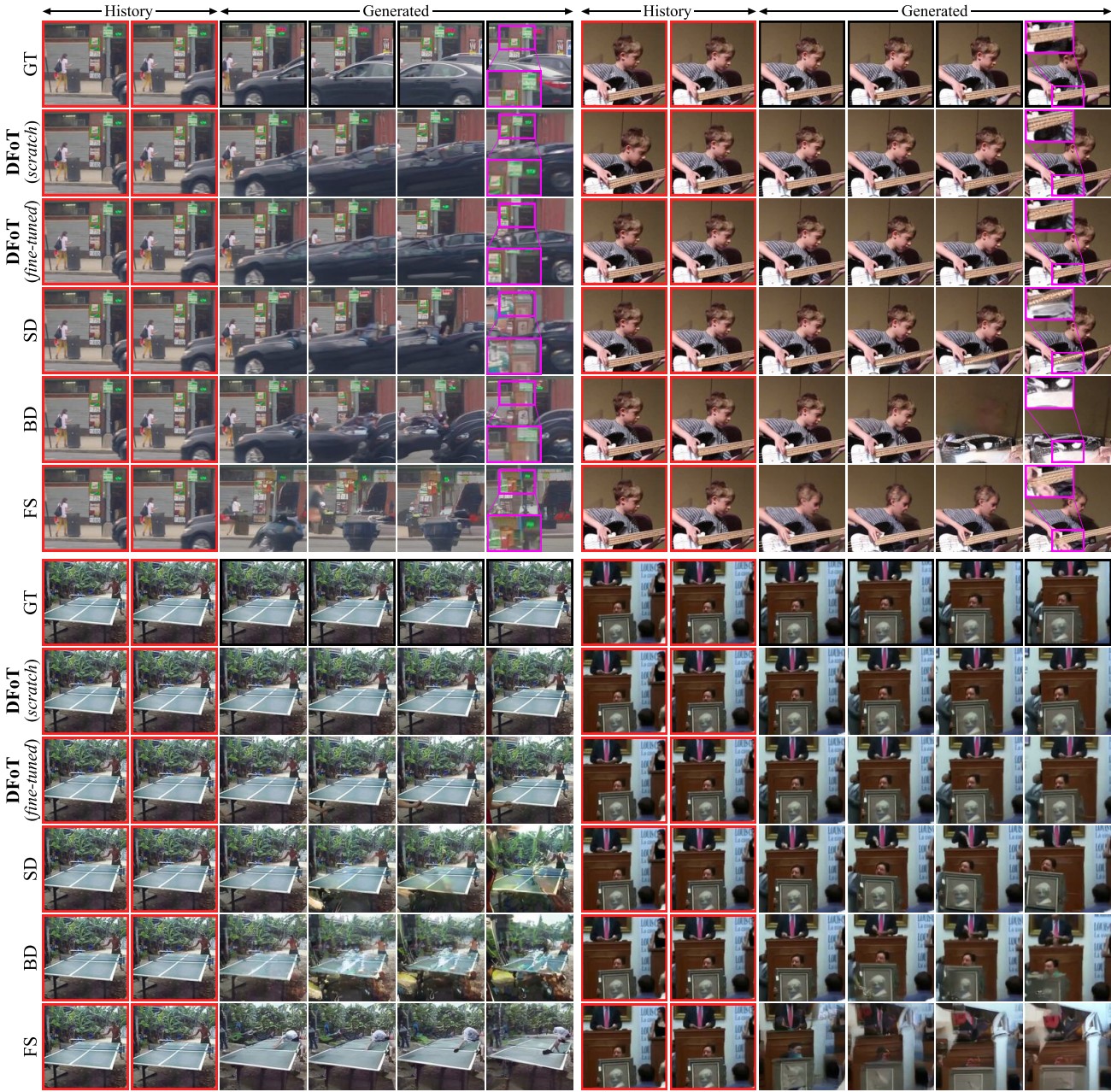

*Figure 14.* **Additional qualitative comparison on Kinetics-600**. We uniformly subsample 6 frames $\{0, 3, 6, 9, 12, 15\}$ from 16-frame videos, conditioned on 5-frame histories. Both DFoT variants, *scratch* and *fine-tuned*, consistently align with the history, generating high-quality samples that closely resemble the ground truth. In contrast, the baselines, typically ordered as SD > BD > FS, struggle to maintain consistency and often exhibit artifacts.

# B  Extended Related Work

## B.1  History-conditioned Guidance

In this section, we discuss how CFG is employed for guiding with history in video diffusion models. The most common case is in **Image-to-Video Diffusion Models** (Blattmann et al., 2023a; Xing et al., 2023; Yang et al., 2024), where the model uses the *first frame* for guidance. Typically, the conditioning frame is incorporated into the architecture by concatenating it channel-wise with each frame to be generated, and additionally, the CLIP (Radford et al., 2021) embedding of the conditioning frame is used for cross-attention.

Few **Conditional Video Diffusion Models** have pushed the boundary by guiding with *fixed set of few frames*. Specifically, VideoLDM (Blattmann et al., 2023b) uses the first $\{1, 2\}$ frames for guidance, W.A.L.T. (Gupta et al., 2024) guides with the first 2 latent tokens, i.e. $\{5\}$ frames, and 4DiM (Watson et al., 2025) guides with the first $\{1, 2, 8\}$ frames. Similarly, in **Multi-view Diffusion Models**, which is similar to video diffusion models but do not differentiate frame order, CAT3D (Gao et al., 2024) guides with the first $\{1, 3\}$ frames.

Architecturally, these models incorporate history frames in various ways. VideoLDM concatenates a binary mask, indicating whether each history frame is masked, along with all masked history frames, feeding them to every temporal layer using a learnable downsampling encoder. W.A.L.T. simplifies this by directly concatenating the history frames and binary mask to the noisy generation input, omitting the encoder. 4DiM and CAT3D process the entire sequence—both history and generation frames—as a single sequence, with a binary mask concatenated along the channel dimension to indicate whether each frame is masked.

In summary, guiding with history in video models has been explored to a limited extent. While these models differ in how they incorporate history frames into the architecture, they all process history frames separately from generated frames, except for 4DiM and CAT3D, leading to inflexibility of guidance. Additionally, these models are trained using CFG-style random dropout of history frames, which categorizes them as special cases of *Binary-Dropout Diffusion*, shown to be suboptimal. These limitations are highlighted in Section 3. In contrast, our work enables guiding with arbitrary, variable-length history frames without the need for binary-dropout training, facilitated by our modified training objective and architecture design.

# C  Experimental Details

Below, we provide additional details on datasets, architectures, training, evaluation metrics, and protocols for our experiments.

## C.1  Datasets

**Kinetics-600 (Kay et al., 2017)** is a widely used benchmark dataset for video generation, featuring 600 classes of approximately 400K action videos. In addition to its role as a standard benchmark, the task is history-conditioned video generation, making it ideal for evaluating our methods. Following prior works, we use a resolution of $128 \times 128$ pixels. Despite the large volume of videos and their low resolution, generating high-quality samples from the Kinetics-600 dataset is challenging even with large models due to the diversity and complexity of the content, and thus qualifies as our primary benchmark.

**RealEstate10K (Zhou et al., 2018)** is a dataset of home walkthrough videos, accompanied by camera pose annotations. While the dataset is predominantly used in novel view synthesis tasks, we utilize it for several reasons: 1) The camera poses allow for a more controlled evaluation of video models; for instance, we can easily switch between highly stochastic and deterministic tasks by altering the camera poses, 2) The dataset's nature enables the examination of the consistency of generated videos at a 3D level, and 3) The dataset's relatively smaller size compared to other text-conditioned video datasets makes it more computationally feasible to train our models, while still providing high-resolution videos. We use a resolution of $256 \times 256$ pixels.

**Minecraft (Yan et al., 2023)** is a dataset of Minecraft gameplay videos, where the player randomly navigates using 3 actions: forward, left, and right. The dataset consists of 200K videos, each with a length of 300 frames, each frame has a corresponding action label. The dataset is designed in a way that good FVD can only be achieved with a long context under action conditioned setting. Specifically, the dataset contains many trajectories where the player turns around and visits areas that it had visited before. While the original dataset is $128 \times 128$ pixels, we train and evaluate on an upsampled version of $256 \times 256$ pixels, to generate higher-quality samples.

**Fruit Swapping** is an imitation learning dataset associated with a fruit rearrangement task adopted from Diffusion

Forcing (Chen et al., 2024). The task involves a tabletop setup where an apple and an orange are randomly put in two of the three empty clots. A single-arm robot is tasked with swapping the two fruits' slots using the third, empty slot as shown in Figure 17. The task requires long-horizon memory since one must remember the initial configuration of the slots to determine the final, target configuration. While the three slots provide a discrete state, each slot has a diameter of 15 centimeters and the fruit can be anywhere in the slot as soon as half of its column resides inside the slot. The task is made even harder when an adversarial human deliberately perturbs the fruit within its slot during the task execution - if there are 10 possible locations within each slot, there would already be $10^3$ combinations of waypoints. This requires a robot policy to be reactive to the fruit locations rather than memorizing all possible combinations. The dataset contains 300 expert demonstrations of the entire swapping task collected by a model-based planner, during which no disturbance happens. The robot may move an apple from slot 1 to the center of slot 2, move the orange from slot 3 to the center of slot 1, and then move the apple from slot 2 to slot 3. Notably, it had never seen a situation where the apple changed its location from center to edge during the middle of the manipulation due to adversarial humans. In addition, the dataset features 300 additional demonstrations of re-grasping, which is a very short recovery behavior when it narrowly misses the fruit. In these re-grasping demonstrations, the robot arm only repositions to grab the missed object without moving it to another slot. Therefore, the dataset contains 300 demonstrations that involve moving fruits but no regrasping, and 300 demonstrations of regrasping but no moving fruit. The former has an average length of 540 frames and the later has an average length of around 50 frames.

### C.2 Implementation Details

We provide a summary of our implementation details in Table 2 and discuss them below.

**Pixel vs. Latent Diffusion.** In this work, we validate DFoT and HG using both pixel and latent diffusion models. For Kinetics-600 and Minecraft, we train a latent diffusion model to enhance computational efficiency. Specifically, for Minecraft, we train an ImageVAE (Kingma, 2013) from scratch, which compresses $256 \times 256$ images into $32 \times 32$ latents, following the approach of Stable Diffusion (Rombach et al., 2022). For Kinetics-600, we train a chunk-wise VideoVAE that compresses $\{1, 4\} \times 128 \times 128$ video chunks into $16 \times 16$ latents, to more aggressively reduce computational costs. This approach resembles CausalVideoVAE, commonly used in prior works (Yu et al., 2023b; Gupta et al., 2024), which compresses an entire $17 \times 128 \times 128$ video into $5 \times 16 \times 16$ latents via causal convolutions. However, we choose to compress every 4 frames separately to preserve DFoT's flexibility. Moreover, this ensures that consistency is influenced solely by the performance of the diffusion model, not the VAE. We implement the VideoVAE and training procedure following Open-Sora-Plan (Lin et al., 2024a). Lastly, for RealEstate10K, we train directly in pixel space, based on the observation that latent diffusion models struggle to correctly follow camera pose conditioning, leading to poor performance on this dataset. Architectures and training details differ significantly between pixel and latent diffusion models, as we discuss in the following sections.

**Architecture.** We employ the DiT (Peebles & Xie, 2023) and U-ViT (Hoogeboom et al., 2023; 2024) backbones for the latent and pixel diffusion models, respectively. Both are transformer-based architectures; however, the key difference is that DiT's transformer blocks operate at a single resolution, whereas U-ViT incorporates multiple resolutions, with transformer blocks residing at each resolution. Due to this difference, we observe that the U-ViT backbone scales better in the pixel space. For improved scalability and temporal consistency, instead of using factorized attention (Ho et al., 2022b), where attention is applied separately to spatial and temporal dimensions, we employ 3D attention that operates on all tokens simultaneously. In addition to this, we incorporate 3D RoPE (Su et al., 2023; Gervet et al., 2023) as relative positional encodings for the $T, H, W$ dimensions.

All conditioning inputs, including noise levels, actions, and camera poses, are injected into the model using an AdaLN layer, following (Peebles & Xie, 2023). For noise levels, since each frame retains independent noise levels in DFoT, an AdaLN layer is applied separately to each token, using the noise level of the corresponding frame. Minecraft actions are converted into one-hot vectors, which are then transformed into embeddings through an MLP layer and added to the noise level embeddings. For camera pose conditioning in RealEstate10K, we compute the relative camera pose with respect to the first frame. Following the methodologies of 3DiM (Watson et al., 2023) and 4DiM (Watson et al., 2025), this relative pose is then converted into ray origins and directions, which are then transformed into 180-dimensional positional embeddings, similar to Nerf (Mildenhall et al., 2021). Across the resolutions of U-ViT, the camera pose embeddings are spatially downsampled to match the resolution before being injected into the model.

**Diffusion.** We use a cosine noise schedule (Nichol & Dhariwal, 2021) for all of our diffusion models. For the RealEstate10K and Minecraft models, we shift the noise schedule to be significantly noisier (Hoogeboom et al., 2023) by a factor of

*Table 2.* **Implementation details for DFoT and baseline models.**

| | Kinetics-600 | RealEstate10K | Minecraft | Imitation Learning |
|---|---|---|---|---|
| *VAEs* | | | | |
| Input | $\{1,4\} \times 128 \times 128$ | | $1 \times 256 \times 256$ | |
|   Compression $(f_t, f_s)$ | $\{1,4\}, 8$ | | 1, 8 | |
|   Latent channels | 16 | | 4 | |
| Training steps | 600k | | 50k | |
|   Optimizer | Adam | - | Adam | - |
|   Batch size | 64 | | 96 | |
|   Learning rate | 1e-4 | | 4e-4 | |
|   EMA | 0.999 | | ✗ | |
| *VDMs* | | | | |
| Input | $17 \times 128 \times 128$ | $8 \times 256 \times 256$ | $50 \times 256 \times 256$ | $21 \times 32 \times 32$ |
|   Latent | $5 \times 16 \times 16$ | ✗ | $50 \times 32 \times 32$ | ✗ |
|   Frame skip | 1 | $10 \rightarrow \text{Max}$ | 2 | 15 |
| Backbone | DiT | U-ViT | DiT | Attention UNet |
|   Patch size | 1 | 2 | 2 | 1 |
|   Layer types | Transformer | $[\text{ResNet} \times 2, \text{Transformer} \times 2]$ | Transformer | Attention, Conv |
|   Layers | 28 | $[3, 3, 6, 20]$ | 12 | 8 |
|   Hidden size | 1152 | $[128, 256, 576, 1152]$ | 768 | 128 |
|   Heads | 16 | 9 | 12 | 4 |
| Training steps | 640k | 500k | 200k | 100k |
|   Warmup steps | 10k | 10k | 10k | 10k |
|   Optimizer | AdamW | AdamW | AdamW | AdamW |
|   Batch size | 192 | 96 | 96 | 64 |
|   Learning rate | 2e-4 | 5e-5 | 1e-4 | 5e-4 |
|   Weight decay | 0 | 1e-2 | 1e-3 | 1e-3 |
|   EMA | 0.9999 | 0.9999 | 0.9999 | ✗ |
| Diffusion type | Discrete | Continuous | Discrete | Discrete |
|   Noise schedule | Cosine | Shifted Cosine | Shifted Cosine | Cosine |
|   Noise schedule shift | ✗ | 0.125 | 0.125 | ✗ |
|   Parameterization | $\mathbf{v}$ | $\mathbf{v}$ | $\mathbf{v}$ | $\mathbf{x}_0$ |
|   Sampler | DDIM | DDIM | DDIM | DDIM |
|   Sampling steps | 50 | 50 | 50 | 50 |

0.125, which we find markedly enhances sample quality, especially for RealEstate10K. This finding aligns with prior works (Chen, 2023; Hoogeboom et al., 2023) that highlight the importance of adding sufficient noise during training, especially when dealing with highly redundant images, such as those with high resolution. Another important design choice is the parameterization of diffusion models. We employ the $\mathbf{v}$-parameterization (Salimans & Ho, 2022) for all models, which has been widely adopted in image and video diffusion models (Ho et al., 2022a; Lin et al., 2024b) due to its superior sample quality and quicker convergence, except for the robot model, where we use the $\mathbf{x}_0$-parameterization. Lastly, to expedite training, we use min-SNR loss reweighting (Hang et al., 2023) for Kinetics and robot learning, and sigmoid loss reweighting (Kingma & Gao, 2023; Hoogeboom et al., 2024) for RealEstate10K and Minecraft.

**Training.** We train models *for each dataset* and *for each model class (e.g., DFoT, SD, etc.)*, using the same pipeline within each dataset. We apply a *frame skip*, where training video clips are subsampled by a specific stride: a value of 1 for Kinetics-600, 2 for Minecraft, and 1 for Imitation Learning. For RealEstate10K, we use an increasing frame skip, starting from 10 and extending to the maximum frame skip possible within each video, to help the model learn various camera poses. Throughout all training, We employ the AdamW (Loshchilov, 2017) optimizer, with linear warmup and a constant learning rate. Additionally, we utilize fp16 precision for computational efficiency and clip gradients to a maximum norm of 1.0 to stabilize training. For robot imitation learning, we follow the setup in Diffusion Forcing (Chen et al., 2024) where we concatenate actions and the next observation together for diffusion, with the exception that we stack the next 15 actions

together for every video frame.

**Sampling.** For all experiments, we use the deterministic DDIM (Song et al., 2020) sampler with 50 steps. Sampling with history guidance, which requires multiple scores at every sampling step, is implemented by stacking the corresponding inputs across the batch dimension to compute the scores in parallel. These scores are then composed to obtain the final score for the DDIM update.

**Compute Resources.** We utilize 12 H100 GPUs for training all of our video diffusion models, with each model requiring approximately 5 days to train under our chosen batch size. One exception is the Robot model, which is trained on 4 RTX4090 GPUs for 4 hours. We note that most of the video models converge in validation metrics with a fraction of our reported total training steps. However, we chose to train them longer because the industry baselines on these datasets (Yu et al., 2023a; Ruhe et al., 2024) are trained for a great number of epochs that are even unmatched by our final training steps. There was no noticeable overfitting throughout the process.

## C.3 Evaluation Metrics.

**Fréchet Video Distance (FVD, Unterthiner et al. (2018)).** We employ FVD as the primary evaluation metric for video generation performance. Similar to FID (Heusel et al., 2017), FVD computes the Fréchet distance between the feature distributions of generated and real videos, with features extracted from a pre-trained I3D network (Carreira & Zisserman, 2017). Lower FVD scores indicate better video generation performance. Unlike image-wise metrics such as FID, FVD evaluates entire video sequences, capturing temporal consistency and dynamics in addition to quality and diversity, making it the most suitable metric for our video generation tasks. Moreover, FVD is computed for the entire video, including both history and generated frames, to assess the consistency between them.

**VBench (Huang et al., 2024).** We use VBench, an evaluation suite designed to assess video generation models in a comprehensive manner, when separate evaluation for different aspects of video generation is needed. Among 16 sub-metrics, we focus on 5 metrics to assess three aspects: 1) *Frame-wise Quality*, calculated as the average of *Aesthetic Quality* and *Imaging Quality*, assesses the visual quality of individual frames; 2) *(Temporal) Consistency*, derived as the average of *Subject Consistency* and *Background Consistency*, evaluates the short- and long-term consistency of generated videos; and 3) *Dynamic Degree* assesses the degree of dynamics, i.e., the amount of motion in the generated videos. All metrics are better when higher, evaluate the generated videos independently without comparison to the ground truth, and are computed by averaging over all generated videos.

**Learned Perceptual Image Patch Similarity (LPIPS, Zhang et al. (2018)).** We use LPIPS as an alternative metric for highly deterministic tasks, where video-wise metrics may not be as sensitive and accurate. LPIPS computes the perceptual similarity between the generated and corresponding ground truth frames, with lower scores indicating higher similarity. We compute LPIPS only for the generated frames, excluding the history frames, to evaluate whether the generated frames are visually similar to the ground truth frames.

## C.4 Details on Video Generation Benchmark (Section 6.2)

**Kinetics-600 Benchmark.** We closely follow the experimental setup of prior works (Ho et al., 2022b; Yu et al., 2023a;b; Ruhe et al., 2024). On the test split of the dataset, we evaluate the models on a video prediction task, where the model is conditioned on the first 5 history frames and asked to predict the next 11 frames. Since our models, utilizing VideoVAE, generate 3 future tokens corresponding to 12 frames, we drop the last frame to align with the prediction task. We report the FVD score computed on 50K generated 16-frame videos, using three different random seeds.

**Resource Comparison Against Industry-Level Literature Baselines.** In Table 1, we show that DFoT not only outperforms generic diffusion baselines trained with the same pipeline but also holds its ground against strong literature baselines, including Video Diffusion (Ho et al., 2022b), MAGVIT (Yu et al., 2023a), MAGVIT-v2 (Yu et al., 2023b), W.A.L.T (Gupta et al., 2024), and Rolling Diffusion (Ruhe et al., 2024). We have selected only the highest-performing baselines from the literature for comparison, omitting others for brevity.

A critical aspect of our evaluation is the comparison of computational resources. Our DFoT is trained with fewer resources compared to these industry-level baselines. Specifically, two primary factors affect the performance of diffusion models: network complexity and training batch size. Our DFoT model is a 673M parameter model with a DiT backbone, trained with a batch size of 196.

*(i) Network Complexity.* As Video Diffusion and Rolling Diffusion have different backbones from ours, we compare the number of parameters; they are billion-parameter models, each with 1.1B and 1.2B, significantly larger than our model.

For MAGVIT, MAGVIT-v2, and W.A.L.T, which are pure transformer models with similar backbones, we use Gflops as a measure of computational complexity, as suggested by (Peebles & Xie, 2023). Our model is of DiT/XL size, whereas the baselines are DiT/L size, making them slightly smaller. In terms of Gflops, our model has $\approx 1.5$ times more Gflops compared to these baselines.

*(ii) Batch Size.* Video Diffusion, MAGVIT, and MAGVIT-v2 are trained with a batch size of 256, while W.A.L.T and Rolling Diffusion are trained with a batch size of 512, which is significantly larger than ours.

When considering both network complexity and training batch size, MAGVIT and MAGVIT-v2 use comparable resources to our model, whereas Video Diffusion, W.A.L.T, and Rolling Diffusion require significantly more resources. Despite this resource disadvantage, DFoT proves to be highly competitive with these strong baselines. It is only slightly behind W.A.L.T, comparable to MAGVIT-v2, and outperforms the rest. This highlights the superior performance of DFoT as a base video diffusion model.

### C.5 Details on History Guidance Experiment (Section 6.3)

For the Kinetics-600 rollout experiment, the models generate the next 59 frames using sliding windows, given the first 5 history frames. The sliding windows are applied such that the model is always conditioned on the last 2 latent tokens and generates the next 3 latent tokens. As with the Kinetics-600 benchmark, we drop the last frame to align with the task. We assess the FVD and VBench scores on 1,024 generated 64-frame videos.

*History Guidance Scheme.* To investigate the effect of HG-v and HG-f, we vary guidance scales using an equally spaced set of $\omega \in \{1.0, 1.5, 2.0, 2.5, 3.0, 3.5, 4.0\}$ for both methods. For HG-f, we use a fixed fractional masking degree of $k_{\mathcal{H}} = 0.8$, which we find to generate videos with sufficient dynamics.

### C.6 Details on OOD History Experiment (Section 6.4, Task 1)

In **Task 1** of Section 6.4, we have shown that video diffusion models easily fail to generalize when the conditioning history is OOD, and temporal history guidance resolves this challenge, through a systematic study on RealEstate10K. Below, we detail the experiment.

**What makes a history OOD?** As shown in the training data distribution of Figure 7, we find that the rotation angle of the camera poses within a single training scene is typically small, rarely exceeding $100°$. Hence, a history with a wider rotation angle, such as $150°$, is considered OOD. Based on this observation, we assign the following tasks to the models: *"Given a 4-frame history, with varying rotation angles, generate 4 frames that interpolates between these frames."*

**Evaluation Based on Rotation Angles.** We categorize all scenes based on their rotation angles, into the bins of $[0°, 10°], [10°, 20°], \ldots, [170°, 180°]$. Based on the statistics of the training scenes, we conceptually classify the bins of $[0°, 10°], \ldots, [90°, 100°]$ as *in-distribution*, $[100°, 110°], \ldots, [130°, 140°]$ as *slightly OOD* ($< 500$ training scenes), and $[140°, 150°], \ldots$ as *OOD* ($< 100$ training scenes). We then randomly select 32 test scenes (or less if the bin contains fewer scenes) from each bin. For each scene, we select 4 equally spaced frames from the beginning and end of it as the history, and designate the target frames as those in between. We evaluate by computing the LPIPS between the generated and target frames, and report the average LPIPS score for each bin, as shown in Figure 7.

*History Guidance Scheme.* From a full history $\mathcal{H} = \{0, 1, 2, 3\}$, we compose scores conditioned on the following two history subsequences: $\mathcal{H}_1 = \{0, 1, 2\}$ and $\mathcal{H}_2 = \{1, 2, 3\}$, each with a guidance scale of $\omega_1 = \omega_2 = 2$. Additionally, we implement an extended version of temporal history guidance discussed in Appendix A.8, by also composing generation subsequences: $\mathcal{G}_1 = \{4, 5, 6\}$ and $\mathcal{G}_2 = \{5, 6, 7\}$ chosen from the full generation $\mathcal{G} = \{4, 5, 6, 7\}$. For the baseline using vanilla history guidance, we apply a guidance scale of $\omega = 2$ to the full history $\mathcal{H}$.

### C.7 Details on Long Context Generation (Section 6.4, Task 2).

We train a 50-frame DFoT model that can condition on history up to a length of 25 following the simplified objective Appendix A.5. Note that this is equivalent to 100 frames under the original video with a frameskip of 2, or one-third of the maximum video length. We sample an initial context of 25 from the dataset and use our trained model to auto-regressively diffuse the next 25 frames conditioned on the previous 25. We roll out 5 times, or 125 frames in total, converging the maximum video length in the dataset.

*History Guidance Scheme.* During sampling, we compose the scores from one long-context model and one short-context model, with context lengths of 25 and 4 respectively. Subtracting the unconditioned score doesn't play a significant role on this dataset so we proceed to compose the above two scores only, with a simple weighting of $50\%$ each.

### C.8 Details on Robot Imitation Learning (Section 6.4, Task 3).

**Baselines.** We compare against other diffusion-based imitation learning methods using our same architecture and implementation. First, we compare against a typical Markovian model, which diffuses the next few actions only based on current observation. Then, we use a variant of this Markovian model, which can see the previous two frames as a short history but still no long-term memory. Notice that these two short history lengths represent the current mainstream approaches (Chi et al., 2023). In addition, we have a third baseline trained to condition on the entire history so far, representing a family of decision-making as sequence generation methods. For the convenience of notation, we will refer to these baselines as Markov model, 2-frame model, and full-history model. All baselines are trained to diffuse actions and next observations jointly.

**The Need to Compose Subtrajectories.** As we mentioned in the dataset description, robot imitation learning is a sequence task that requires both long-term memory and local reactive behavior. While both are important to the final task's success, a short-context model will trivially fail most of the time since it won't remember which final state to proceed to. Therefore we focus on our experiment design on exploiting the failure mode of long-context models. One predominant failure mode is overfitting - since the imitation learning dataset is extremely small, a long-context model can attribute an action to any coincidental features. For example, all swapping trajectories in the dataset feature the behavior of putting the first fruit in the very center of the initially empty slot and coming back later to move it away from that center location. How should the model determine where it should pick up this fruit? There is little guarantee for it to determine correctly that it shall proceed to move its gripper right above that fruit versus just blindly going to the center. Whenever a human perturbs this fruit from the very center of the slot to the edge of the slot, an overfitted model will still move to the very center and proceed to grasp air, ignoring the actual location of that fruit. Therefore, theoretically, a full-history model would never be able to react to such perturbation, since it had never seen a trajectory with such perturbation and a successful trajectory would be out-of-distribution. Instead, it needs to mix in some behavior from a local reactive policy to perform the task, leveraging the fact that whenever a long history is out-of-distribution, you can always fall back to a shorter context model and imitate relevant sub-trajectories. Therefore, the only way to solve this task under the adversarial human is to stitch sub-trajectories together while keeping a long-term memory.

*History Guidance Scheme.* To achieve the aforementioned stitched behavior, we compose three diffusion models with a context of 1 frames, 4 frames, and full history. We assign the full-history model with a small weight of $0.2$, the 1 frame model, and the 4 frame model with a weight of $0.45$ each. Like Minecraft, we didn't find subtracting unconditioned score super important in this task so we omitted it. The frames here refer to the bundle of the next $15$ actions and the single future video frame after that as we mentioned earlier in implementation details.

### C.9 Details on Ultra Long Video Generation (Section 6.5).

We provide additional details on generating long navigation videos on RealEstate10K, incorporating all advanced techniques associated with DFoT and history guidance. The generation of long navigation videos is divided into two phases: (i) a rollout phase, where the model generates a long video using a sliding window approach, and (ii) an interpolation phase, where the generated frames are further interpolated to create a smooth video. The process is detailed below.

**(i) Rollout Phase.** During the rollout phase, starting with a *single image* randomly selected from the dataset, the model generates a long video using a sliding window, where it is conditioned on the last 4 frames to generate the next 4 frames. The first iteration is an exception, where the model is conditioned on the single image and generates the next 7 frames. Importantly, navigation cannot rely on the ground truth camera poses for two reasons: 1) videos in the dataset are relatively short (less than 300 frames), so we quickly exhaust available camera poses, and 2) the navigation task is highly stochastic, meaning the ground truth camera poses may not align with the generated frames (e.g., moving straight into a wall). To address this, we have developed a simple navigation UI, allowing a *user to navigate freely in the scene by providing inputs* after each sliding window iteration. Specifically, the user can specify the horizontal and vertical angles, relative to the current frame, for the desired navigation direction, as well as the movement distance. This input is converted into a sequence of camera poses, which are then used as conditioning input for the model to sample the next set of frames. This process is repeated until the desired video length is achieved.

**(ii) Interpolation Phase.** Next, in the interpolation phase, leveraging DFoT's flexibility which supports interpolation, we interpolate between the generated frames by a factor of 7. Specifically, using every pair of consecutive generated frames as history, we interpolate 6 frames between them. Camera poses for the interpolated frames, which should be given as input to the model, are computed by linearly interpolating the camera poses of the frames at both ends. More specifically, rotation matrices are interpolated using SLERP (Shoemake, 1985), and translation vectors are linearly interpolated.

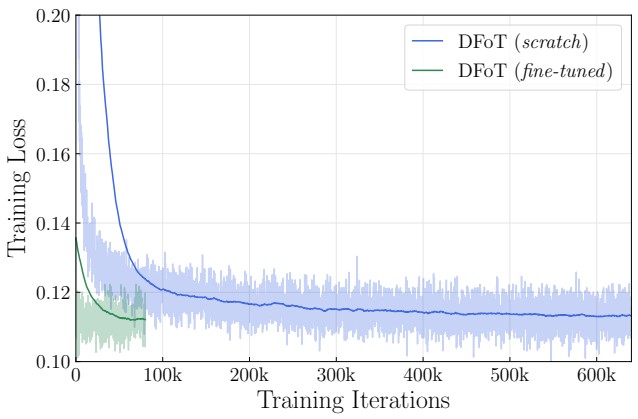 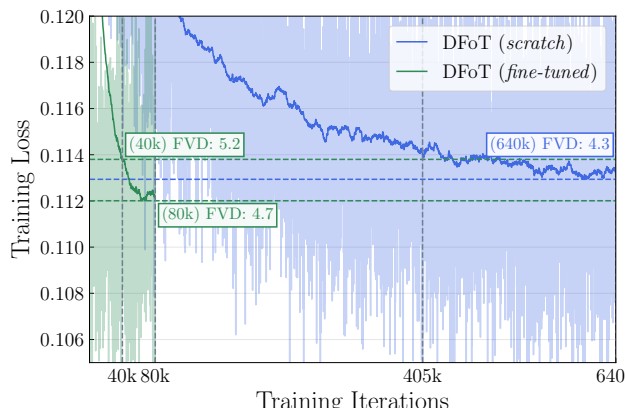

(a) A comprehensive view of the training loss curves. DFoT *(fine-tuned)* achieves a low training loss early in the iterations and converges significantly faster than DFoT *(scratch)*.

(b) A zoomed-in view of the training loss curves. Only after 80k iterations, DFoT *(fine-tuned)* displays a lower training loss than DFoT *(scratch)* trained for 640k iterations.

*Figure 15.* **Training loss curves for DFoT, trained from *scratch* and *fine-tuned* from the pre-trained FS model, on Kinetics-600.**

***History Guidance Scheme.*** Finally, we discuss how history guidance is utilized throughout the navigation task. During the sliding window rollout, the default HG scheme is HG-f, which we find to be extremely stable during long rollouts. Specifically, we apply HG-f with a guidance scale of $\omega = 4$ with a fractional masking degree of $k_\mathcal{H} = 0.4$, chosen to ensure optimal stability. Additionally, we switch to HG-v with a guidance scale of $\omega = 4$ for more challenging situations, such as when the model needs to "extrapolate" to new areas. This is because HG-v performs better in such challenging scenarios, although it is less stable than HG-f, and thus is used sparingly. This switch is triggered when the model is asked to change the direction by more than $30°$, or when the model is asked to move further than a certain distance. During the interpolation phase, we apply HG-v with a small guidance scale of $\omega = 1.5$, to ensure the interpolated video is smooth and consistent.

**Stabilization.** As an additional techinque, we also employ the stabilization technique proposed in Diffusion Forcing (Chen et al., 2024), where the previously generated frames are marked to be slightly noisy at a level of $k = 0.02$, to prevent error accumulation, thereby further stabilizing the long rollout.

## D   Additional Experimental Results

In this section, we present additional experimental results to (i) answer potential questions that may provide further insights into our proposed DFoT and HG, and (ii) further elaborate and provide additional samples for Section 6.

### D.1   Additional Results on Fine-tuning to DFoT

Below we provide detailed results on fine-tuning a pre-trained full-sequence (FS) model to DFoT, both from training and sampling perspectives.

**Training Dynamics.** We show the training loss curves of for two variants of DFoT, one trained from *scratch* for 640k iterations, and the other *fine-tuned* from the pre-trained FS model for 80k iterations, in Figure 15. We observe that the pre-trained model already provides a good initialization for DFoT, as the model starts with a low training loss and converges rapidly in the early iterations, in Figure 15a. Surprisingly, the fine-tuned model achieves a lower training loss than the model trained from scratch after only 80k iterations, as shown in Figure 15b. Moreover, after 40k iterations, the fine-tuned model exhibits a training loss comparable to the model trained from scratch for 405k iterations, which is ∼10x speedup. This highlights the superior efficiency and ease of training DFoT by fine-tuning from a pre-trained model. While this opens up the possibility of fine-tuning large foundational video diffusion models to DFoT with small computational cost, we leave this as future work.

**FVD Metric Evolution.** In contrast to the training loss, Figure 15b (or Table 1) shows that the fine-tuned model achieves a slightly higher FVD score than the model trained from scratch, although being highly competitive even after 40k iterations. We attribute this discrepancy to the use of EMA, which is commonly employed in diffusion models to enhance sample quality (Ho et al., 2020; Dhariwal & Nichol, 2021). By default, we use an EMA decay of 0.9999, and thus the model weights used for sampling are affected by the last tens of thousands of training iterations. Therefore, the fine-tuned model's superior

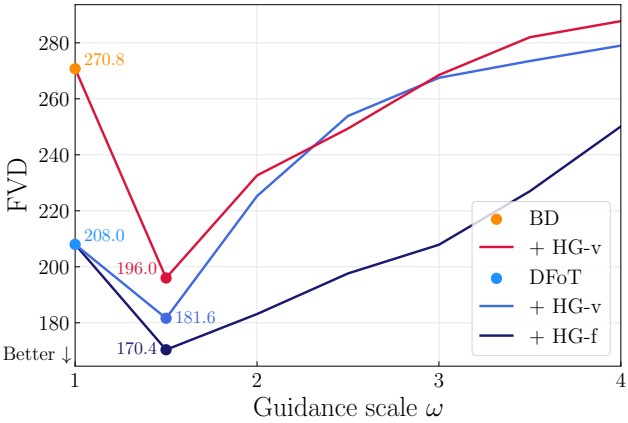

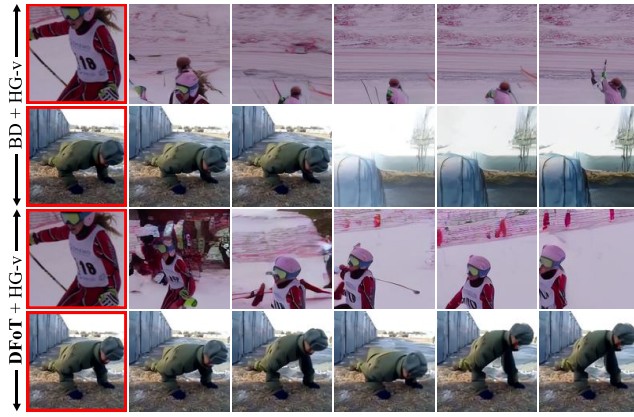

(a) FVD as a function of guidance scale $\omega$ for DFoT and BD using HG. Both with HG-v, DFoT yields better FVD-$\omega$ curves than BD and thus achieves a lower best FVD score. Applying HG-f, which is specific to DFoT, enlarges the performance gap.

(b) Qualitative comparison of DFoT and BD using HG-v with optimal guidance scales $\omega = 1.5$. While DFoT generates consistent, high-quality samples, BD struggles to remain consistent with the history frames and produces artifacts. Red box = history frames.

*Figure 16.* **History Guidance works better with DFoT than with Binary-Dropout Diffusion (BD).**

training loss does not immediately translate to a lower FVD score, but we expect it to outperform the model trained from scratch after an additional short training period. While one may consider simply fine-tuning the model without EMA to speed up, EMA is crucial for sample quality; for example, at 80k iterations, FVD without EMA is 7.3, significantly higher than the 4.7 with EMA. This suggests that choosing a smaller EMA decay that still guarantees sample quality, through sophisticated strategies such as post hoc EMA tuning (Karras et al., 2024), may be a promising direction for future work.

### D.2 Ablation Study on Binary-Dropout Diffusion with Vanilla History Guidance

While we have shown that Binary-Dropout Diffusion (BD) performs poorly as a base model (**Q2** of Section 6.2), BD still can implement vanilla history guidance due to its binary dropout training. As such, a natural question is: *How does BD perform with HG-v, compared to DFoT?* To answer this question, we repeat the Kinetics-600 rollout experiment in Section 6.3 using BD with HG-v, comparing against DFoT with HG. See Figure 16 for the results. We observe that DFoT consistently outperforms BD across all guidance scales except for $\omega = 2.5$, as shown in Figure 16a. Under their optimal guidance scales of $\omega = 1.5$, DFoT achieves a lower FVD score of 181.6 compared to BD's 196.0, and qualitatively, generates more consistent, high-quality samples, as shown in Figure 16b. When using HG-f, which is only applicable to DFoT, DFoT further outperforms BD, achieving an FVD score of 170.4. These results highlight that DFoT is a better base model for implementing history guidance, both in performance and in a variety of guidance methods that can be applied.

### D.3 Detailed Results on Long Context Generation (Section 6.4, Task 2)

We calculate the FVD on 1024 samples across all 125 generated frames. A simple conditional diffusion model with context full context achieves an FVD of 97.625 while our temporal guidance achieves an FVD of 79.19 (lower is better). We note that while traditionally FVD is a bad metric for videos with high intrinsic variance, it's well-suited for our benchmark since both action-conditioning and the dataset design constrain the possible variance. We visually observe that Diffusion Forcing Transformer's prediction aligns well with the ground truth semantically over the majority of the frames in a video, showing the variance is well-warranted. We visualize one randomly picked sample in Figure 13, showing that temporal guidance can maintain high-quality details far into the future even without CFG. In the meanwhile, the long-context model without temporal guidance can suffer from the high dimensional context, which makes it much more likely to see out-of-distribution frames in its history.

### D.4 Detailed Results on Long-horizon yet Reactive Imitation Learning (Section 6.4, Task 3)

We examine the success rate of robot imitation learning quantitatively by randomizing the environment 100 times before testing the temporal guidance model as well as its baselines. We found that the Markov baseline fails to perform the task completely as expected since it has trouble sticking to a specific plan - it would move away from fruit and then move back halfway since it has no memory. The 4-frame model suffers from the same issue and cannot finish the task. It does react well to perturbations on the object and picks up the fruit from time to time, showing short context indeed prevents

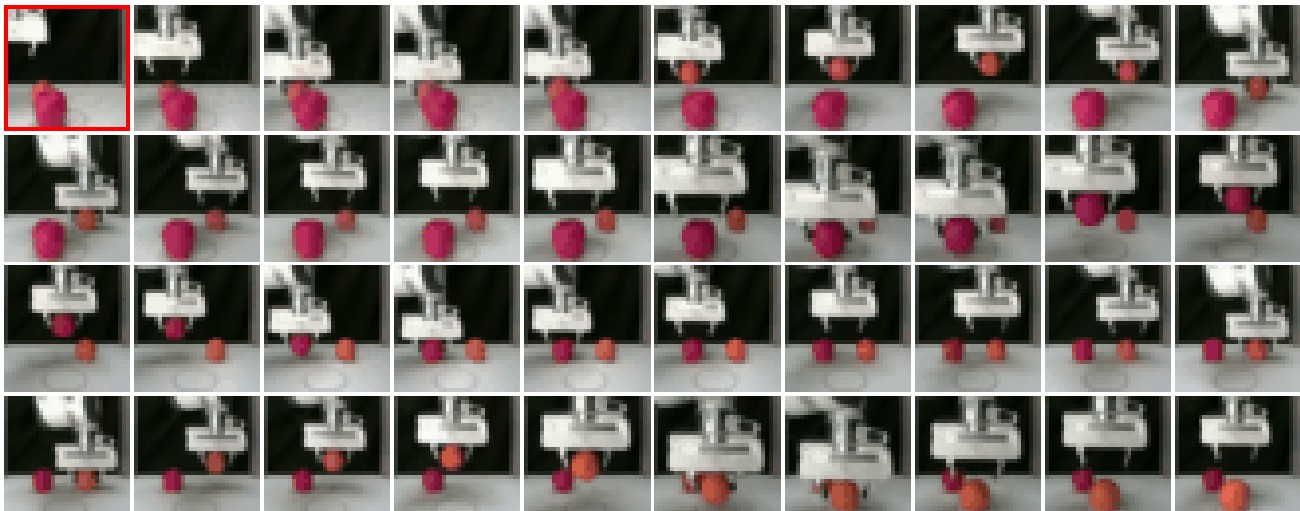

*Figure 17.* **Visualization of the fruit-swapping task through a DFoT generated video.** Two fruits are randomly put within two random slots. The robot is tasked with swapping its slots using the third slot and moving one fruit at a time. This task requires long-horizon memory because it needs to remember the initial location of the fruit for the task completion, but also react to different fruit locations within each slot, which is combinatorically impossible form the dataset.

overfitting from temporal locality. We found that the full-history model, with the maximum possible memory, performs well whenever there is no human perturbation. However, as soon as the adversarial human perturbs the fruit during the task execution, this policy often blindly goes to the very center of the third slot while the object is already moved to the edge of the slot. The policy will then proceed to close its gripper, holding nothing, and then move to the next slot, thinking it has something in its hand. There are occasional cases when this doesn't happen and the model actually reacts to the adversarial perturbation, although infrequently and only happens to the case then perturbation from the slot center isn't too big. Overall this shows that using a full-context model naively can make the model suffer from overfitting and one may want to manually emphasize the temporal locality prior. Finally, we tested DFoT composed guidance and found it to achieve a much higher success rate of 83%, showing that it's actually stitching the subtrajectories to make decisions, or at least simultaneously borrowing the memory from the full-context model while staying locally reactive using the short-context model. In addition, we attempted a few stronger perturbations such that the adversarial human will deliberately knock off the fruit from the robot's gripper when it's closing. We found that temporal guidance can even react to this by regrasping and eventually finishing the whole swapping task. However, even temporal guidance achieves only 28% to this strong perturbation since it's way too out-of-distribution and may require more data. Qualitatively, we visualize a generated robot trajectory with an unseen configuration in Figure 17.

### D.5 Additional Qualitative Results

We present additional qualitative results to supplement our main findings in Section 6. Please refer to Figures 8 to 12 and 14 for detailed visual comparisons, which are discussed below.

**DFoT vs. Baselines (Section 6.2, Q1).** We present additional qualitative comparisons of DFoT against baselines in Figure 14, as an extension to the qualitative results shown in Figure 4. Consistent with the quantitative findings in Table 1, DFoT produces more consistent and higher-quality samples compared to all baselines.

**Empirical Flexibility of DFoT (Section 6.2, Q3).** As evidence of the empirical flexibility of DFoT, we present additional qualitative results on RealEstate10K in Figure 11. Our DFoT model successfully generates consistent samples, given histories that vary both in length and timestamps. This highlights the effectiveness of our new training objective, which transforms DFoT into a flexible multi-task model, uniformly achieving high performance across diverse tasks.

**Improving Video Generation via History Guidance (Section 6.3).** In addition to the results shown in Figure 6a for Kinetics-600, we present further qualitative results on RealEstate10K in Figure 12, highlighting the effectiveness of vanilla history guidance in improving video generation. With increasing guidance scales, the generated samples exhibit significantly higher frame quality and consistency, likewise to the results on Kinetics-600. This behavior is consistent across different tasks—extrapolation and showcasing the broad applicability of history guidance in any history-conditioned video generation

task.

**Robustness to Out-of-Distribution (OOD) History (Section 6.4, Task 1).** We provide additional qualitative results for **Task 1** from Section 6.4, as illustrated in Figure 10. These results demonstrate that HG-t enables DFoT to *uniquely* remain robust to OOD history. Failure cases clearly observed in baselines show that typically, video diffusion models only perform well when the history is in-distribution. By composing in-distribution short history windows, HG-t can effectively approximate strictly OOD histories that were unseen during training.

### D.6  Detailed results on Ultra Long Video Generation (Section 6.5).

We present extended results from Section 6.5 below.

**DFoT vs. SD on Long Rollout.** To begin with, we highlight the significant challenges of generating long navigation videos using the RealEstate10K dataset. Specifically, we investigate the performance of SD, the most conventional and competitive baseline. To mitigate the stochastic nature of navigation that complicates comparisons, we evaluate DFoT with HG and SD on a simple navigation task of moving straight, which is almost deterministic. We avoid using interpolation—applicable only to DFoT —to ensure a fair comparison. The results, shown in Figure 9, indicate that SD struggles to maintain consistency with the history frame, failing around frame ∼30. We attribute this to SD's inferior quality and consistency, along with its inability to recover from small errors during generation. In contrast, DFoT with HG succeeds to stably roll out beyond frame 72. Alongside the qualitative comparison, we note that 4DiM (Watson et al., 2025), an SD model that, to our knowledge, produces the longest and highest-quality videos on RealEstate10K among the methods in the literature, generates videos with a maximum length of 32 frames, which is significantly shorter than our long navigation videos.

**More Samples.** We present four samples of long navigation videos generated by DFoT with HG in Figures 8a to 8d. These samples demonstrate the capability of DFoT with HG to stably generate extremely long videos. The generated videos are notably longer than those in the training dataset, which primarily cover a single room or small area, rather than multiple connected rooms or areas.

