# OpenReview forum: "History-Guided Video Diffusion"
_ICML.cc/2025/Conference — ICML 2025 poster_

### Official Review · Reviewer_2WEt · 2025-03-11

**Overall Recommendation:** 3

**Summary:**

This paper delves into video diffusion models, aiming to extend classifier-free guidance (CFG) to video diffusion with variable-length history frames. The authors identify two key challenges: architectures supporting only fixed-size conditioning and poor performance of CFG-style history dropout. To tackle these, they propose the Diffusion Forcing Transformer (DFoT). DFoT is a video diffusion architecture with a theoretically grounded training objective, enabling flexible conditioning on a varying number of history frames. It expands the “noising-as-masking” paradigm to non-causal transformers and is compatible with existing architectures.

**Claims And Evidence:**

DFoT's Performance: The claim that DFoT surpasses baselines in video generation is supported by experiments on datasets like Kinetics 600. Comparing DFoT with standard diffusion (SD), binary-dropout diffusion (BD), and full-sequence diffusion with reconstruction guidance (FS), the authors show that DFoT has better Fréchet Video Distance (FVD) scores and generates higher-quality samples.

History Guidance: The effectiveness of History Guidance methods is well-evidenced. For instance, on Kinetics-600, vanilla history guidance improves frame quality and consistency, and fractional history guidance addresses the issue of static videos. In tasks like handling out-of-distribution history on RealEstate10K, DFoT with temporal history guidance outperforms baselines.

Long-Video Generation: DFoT's ability to generate ultra-long videos is demonstrated by creating an 862-frame navigation video on RealEstate10K, far exceeding the capabilities of prior methods.

**Essential References Not Discussed:**

There are no obvious essential references that are not discussed in the paper.

**Experimental Designs Or Analyses:**

Experimental Designs: The experimental designs are sound. The authors conduct experiments on multiple datasets with diverse characteristics, demonstrating the generality of their methods. For example, Kinetics-600 is used for benchmarking and quantitative comparisons, while RealEstate10K, Minecraft, and Fruit Swapping are used to study new applications. The use of sliding window rollout in experiments on Kinetics - 600 to test video generation consistency is a valid approach.

Analyses: The analyses of the experimental results are comprehensive. The authors report numerical results of evaluation metrics and provide qualitative analyses, such as visualizations of generated videos. This helps readers better understand the performance of DFoT and History Guidance methods. However, in some experiments, like the long-context generation in Minecraft, a more in-depth analysis of the trade-offs between different factors, such as the balance between long-term memory and robustness to out-of-distribution history, would be valuable.

**Methods And Evaluation Criteria:**

Yes

**Other Comments Or Suggestions:**

No

**Other Strengths And Weaknesses:**

Strengths:

1. **Significance**: The research has significant implications. Improving the quality and consistency of video generation, as well as enabling the generation of ultra-long videos, has potential applications in many fields, such as robotics, virtual reality, and video production.
2. **Effective History Guidance Methods**: The History Guidance (HG) family of methods, enabled by DFoT, offers a powerful way to leverage history in video generation. Vanilla History Guidance (HG-v) alone significantly improves video quality and temporal consistency. More advanced methods which combines Temporal History Guidance (HG-t) and Fractional History Guidance (HG-f), further enhance motion dynamics, enable generalization to out-of-distribution history, and can stably roll out extremely long videos.

Weaknesses:

1. **Video Quality and Diversity**: The generated videos have relatively low resolution and limited scene variety. For example, in RealEstate10K experiments, the navigation videos lack complexity and high-resolution details, restricting practical applications. The author are encouraged to experiment with more diverse dataset or stronger pretrained model such as CogVideoX.

**Questions For Authors:**

No

**Relation To Broader Scientific Literature:**

Video Diffusion Models: In the field of video diffusion models, the paper addresses the limitations of existing architectures that can only support fixed-length conditioning. It also improves upon the CFG-style history dropout approach, which has been shown to be suboptimal. The proposed methods offer new ways to condition on history frames, enhancing the performance of video diffusion models in terms of quality, consistency, and the ability to generate long videos.

**Theoretical Claims:**

The authors provide a theoretical justification for the training objective through a variational lower bound. They derive an Evidence Lower Bound (ELBO) corresponding to the DFoT training objective, showing that it optimizes a reweighting of the ELBO on the expected log-likelihoods. This theoretical foundation adds credibility to the DFoT architecture and its training process. However, further discussion on the practical implications of these theoretical results, such as how the reweighting affects DFoT's performance in different video generation tasks, would be beneficial.

---

> ### Author Rebuttal · Authors · 2025-04-01
>
> Thank you for finding our work significant and effective. Below, we address your concerns with further explanation, additional ablations, and results of fine-tuning a video foundation model into DFoT.
>
> > ### **Q1. Practical implications of theoretical results (ELBO)**
>
> Thank you for your insightful question. In practice, reweighting terms in diffusion ELBO are often dropped for simplicity, like in the original DDPM paper. Therefore, it’s difficult to compare DFoT’s ELBO and vanilla diffusion’s ELBO in practice since both of them train on simplified objectives. However, our **Appendix C.2 - Diffusion** does provide some practical insights: Higher noise level masks out a greater portion of information from the original sequence. As noted in previous works [1, 2], and in our findings (Appendix C.2), a diffusion model's performance is highly sensitive to signal redundancy. Specifically, when the denoising task contains a lot of redundant signals, training should emphasize the objective at high noise levels. In DFoT, some tasks (e.g. unconditional, image-to-video) provide smaller amounts of conditioning frames and are thus less redundant, while other tasks (e.g. video-to-video) are the opposite. Thus, the latter benefit from higher emphasis on high noise levels. Therefore, it is expected that **as the reweighting $\omega$ biases towards high noise levels, DFoT's performance shifts towards tasks with more conditioning frames, and vice versa**. Therefore, reweighting and noise schedules should be carefully selected based on this analysis, tailored to the downstream tasks.
>
> > ### **Q2. Trade-offs in long-context generation**
>
> We present an ablation studying how HG-t trades off long-term memory and quality. We conducted the Minecraft long context generation experiments (**Section 6.4 - Task 2**) with varying weights of combining short and full history. We report the results in **Figure R3 on [our anonymous website](https://anon-dfot.github.io/)**. The Minecraft dataset was specifically designed by TECO [5] to be deterministic under an action-conditioned setting. We, therefore, directly use a deterministic metric LPIPS to measure long-context consistency.  In addition, we use frame-wise FID score as a measure for image-only quality, disregarding consistency. The resulting plot shows a clear reverse trend between quality and long-context memory as the weight for full-context increases, indicating that HG-t can trade off long-horizon memory for improved quality robustness by mixing a short-history model.
>
> > ### **Q3. Video Quality and Diversity**
>
> We are excited to report enhanced video generation results using a larger DFoT model at higher resolution, featuring complex and diverse scenes. Please see **Figure R1 on [our anonymous website](https://anon-dfot.github.io/)** for the results. We fine-tuned the 1.3B T2V foundation model [3] to DFoT on a subset of the Panda-70M dataset [4] for 20K steps. The base model is text-to-video only, so our reported image-to-video results are largely due to DFoT, not the base model. Here is a summary of the results:
>
> - **Improved video quality, diversity**: The generated videos have a higher resolution (832x480) and feature more dynamic, complex scenes with enhanced quality. Although the original Wan2.1 model was not trained to condition on images and our fine-tuning dataset is limited to human-action videos, our model effectively generalizes to generate diverse scenes from random web images, ranging from human and animal actions to nature scenes and moving objects.
> - **Scalability of DFoT and History Guidance**: Our method effectively scales to larger models and datasets, while maintaining its advantages. It handles flexible-length history (Figure R1a: conditioning on a single frame or on the previous 13 frames during rollout; Figure R1b: interpolation between the first and last frames). DFoT also stably generates long 217-frame videos via sliding window rollouts (Figure R1a), which is substantially longer than the 49-frame context window used during fine-tuning. Lastly, Figure R1b shows that the benefits of history guidance persist at larger scales, significantly enhancing video quality and consistency.
>
> Overall, these results demonstrate that our method, beyond Re10K experiments, can effectively scale to more complex video generation tasks with superior performance, highlighting its practical applicability.
>
> > ### References
> - [1] Chen, T. "On the importance of noise scheduling for diffusion models." arXiv 2023.
> - [2] Hoogeboom et al. "Simple diffusion: End-to-end diffusion for high resolution images." ICML 2023.
> - [3] WanTeam et al. "Wan: Open and Advanced Large-Scale Video Generative Models." arXiv 2025.
> - [4] Chen et al. "Panda-70m: Captioning 70m videos with multiple cross-modality teachers." CVPR 2024.
> - [5] Yan et al., “Temporally Consistent Transformers for Video Generation”

---

### Official Review · Reviewer_RA3f · 2025-03-14

**Overall Recommendation:** 3

**Summary:**

The paper introduces the Diffusion Forcing Transformer (DFoT), a video diffusion architecture that extends diffusion forcing with a theoretically grounded training objective that enables conditioning on a flexible number of history frames. On top of diffusion forcing using transformer, the authors introduce History Guidance – a family of guidance strategies unique to their architecture where the combination of time and frequency guidance achieves the best in motion dynamics and long-term coherence.

## update after rebuttal
After the rebuttal, most of my concerns have been addressed except for the apparent artifacts in the additional results of model scalability. It is understandable that scaling on the 1.3B model within the rebuttal period with limited resources is hard. Therefore, the scalability of DFoT could be a future direction down the road. Still, this does not affect the fact that this is a strong ICML submission. In light of this, I will keep my original rating.

**Claims And Evidence:**

Claim1:  The most important claim is that the approach enables generating extremely long videos stably, far beyond prior art. This is enabled by rolling out DFoT with the proposed history guidance. The results on the website validate this claim by presenting in-domain visualization results.

Claim2: Compositional generation with long-context is another claim. This is validated on Minecraft video dataset and robotic fruit swapping trajectory.

Claim3: The paper claims DFoT with temporal guidance can handle OOD history inputs (extreme camera poses) that cause other models to fail. This is somehow validated by stress-test on RealEstate10K where abrupt viewpoint changes are introduced between conditioning frames as OOD camera poses. Still, visual domain generalization is not validated.

**Essential References Not Discussed:**

NA

**Experimental Designs Or Analyses:**

Overall, the evaluation is sufficient and solid.

The authors benchmark their base model against strong baselines on standard video generation metrics, i.e., FVD and VBench scores. For deterministic task as next frame prediction for robotics, LPIPS is reported. Datasets are extensively explored, including kinetics, RealEstate10K, Minecraft, and real-world robotic tasks. The analysis and ablation studies are also sound and satisfying.

**Methods And Evaluation Criteria:**

The paper introduces the Diffusion Forcing Transformer (DFoT), a video diffusion architecture that extends diffusion forcing with a theoretically grounded training objective that enables conditioning on a flexible number of history frames. Instead of feeding a fixed set of past frames as conditioning, DFoT noises each frame independently during training. On top of diffusion forcing using transformer, the authors introduce History Guidance – a family of guidance strategies unique to their architecture where the combination of time and frequency guidance achieves the best in motion dynamics and long-term coherence.

The authors benchmark their base model against strong baselines on standard video generation metrics, i.e., FVD and VBench scores. For deterministic task as next frame prediction for robotics, LPIPS is reported. Overall, the evaluation is sufficient and solid.

**Other Comments Or Suggestions:**

NA

**Other Strengths And Weaknesses:**

Strengths are extensively discussed in previous sections. I would mention several weaknesses here:

1. One inherent downside of the history guidance approach is that sampling requires multiple model evaluations per timestep. For CFG (and thus HG-v), you already need two forward passes (conditional & unconditional). For HG-t or HG-f, you might need even more (e.g. one for each history window or each noise level setting) to then combine them. By treating them as a batch might mitigate this issue, but it still takes more computational budget.
2. The scalability of DFoT is not validated. All the dataset used for training are in relatively limited scale. The potential of DFoT in scaling up is not validated and might be worth exploring in the future.
3. Only in-domain (visual domain) results are presented for ultra-long video generation. I’m wondering if DFoT can reasonably generalize to arbitrary in-the-wild input.
4. The method is specialized to using visual history frames as conditioning. It does not experiment with text-to-video or audio-conditioned video, etc. While not exactly a weakness (the paper already has a broad scope), it means the benefits are demonstrated only in the context of visual conditioning. An open question is how DFoT would work if the model also had a text prompt – can it combine text and flexible frame guidance?

**Questions For Authors:**

Please refer to the Strengths and Weaknesses section.

**Relation To Broader Scientific Literature:**

NA

**Theoretical Claims:**

The authors claim that their DFoT training objective is theoretically well-founded, specifically that it optimizes a valid likelihood-based objective (rather than being a heuristic) as Theorem 4.1. In specific, the per-frame noise training loss optimizes a reweighting of an Evidence Lower Bound (ELBO) on the expected log-likelihoods.

---

> ### Author Rebuttal · Authors · 2025-04-01
>
> We appreciate your positive comments on the strengths of our method, theoretical justification, and extensive analysis. Below, we address questions on sampling efficiency and scalability/generalizability of DFoT, including new results from fine-tuning a 1.3B text-to-video foundation model (see **[anonymous website](https://anon-dfot.github.io/)** for **Figures R1, R2**):
>
> > ### **Q1. Sampling efficiency of History Guidance**
>
> As you noted, history guidance requires multiple function evaluations (NFE) per timestep, but this doesn't mean a larger sampling budget is needed. Rather, **Figure R2 on [our anonymous website](https://anon-dfot.github.io/)** shows that **history guidance offers better sampling efficiency than sampling without it**.
> - **Total sampling budget**: Total sampling cost depends not only on NFE per timestep but also on the number of timesteps: $\text{Total NFE} = \text{NFE/timestep} \times \text{\\# timesteps}$
> - **Figure R2**: We compared the performance with/without history guidance under the same total sampling budget (i.e. same total NFE). With the same total NFE, history guidance substantially outperforms sampling without it (mostly HG-f > HG-v > w/o HG). This shows that, despite requiring more NFE per timestep, history guidance can be performed efficiently by reducing the number of timesteps, thereby achieving better performance with the same total sampling cost.
> - **Parallelization**: As you mentioned, even under the same total NFE, history guidance can be performed in parallel by stacking score estimations across batches for faster computation, as we do in our implementation (Appendix C.2 - Sampling).
>
> > ### **Q2. Scalability of DFoT**
>
> We are pleased to report strong results of scaling up DFoT. We fine-tuned the Wan2.1 T2V-1.3B model [1] to DFoT on a subset of the Panda-70M dataset [2] for 20K steps. We note that this model is text-to-video only, so its history conditioning capability is entirely due to DFoT. Despite limited compute and data, **Figure R1 on [our anonymous website](https://anon-dfot.github.io/)** demonstrates the successful scaling while maintaining its advantages:
>
> - **Handling flexible-length history**: The base Wan2.1 model can only generate videos given a text prompt, not being able to condition on history frames. After fine-tuning, DFoT can effectively handle different history-conditional video generation tasks. Specifically, Figure R1a uses a single frame as the initial conditioning frame and then conditions on the previous 13 frames during rollout, while Figure R1b shows interpolation between the first and last frames.
> - **Long video generation**: Figure R1a shows DFoT's long video generation capability still holds at larger scales. We present 217-frame videos via sliding window rollouts, which is substantially longer than the 49-frame window used during fine-tuning.
> - **History guidance**: Figure R1b compares frame interpolation results with/without history guidance, showing that history guidance improves video quality and consistency even at larger scales.
>
> > ### **Q3. Generalizing to (visual) out-of-domain**
>
> The video generation results in **Figure R1** demonstrate that DFoT can **generalize to arbitrary in-the-wild visual input**. We fine-tuned on a subset of Panda-70M filtered to include only 190K videos with human actions. Despite this, DFoT generalizes well to various in-the-wild visual inputs from random internet images (not part of the training set), such as animals, nature scenes, and moving objects. This capability is not merely due to the base model, which is text-to-video only; DFoT provides the image-conditioning ability.
>
> > ### **Q4. Beyond visual conditioning (text prompt)**
>
> Our fine-tuning results further demonstrate that **DFoT is highly compatible with other types of conditioning**, such as text prompts. With just 18 lines of code changes, the fine-tuned model can condition on varying-length history. All videos in **Figure R1** are generated by jointly conditioning using both text prompts and history frames, and adhering closely to both. Specifically, we utilize both text guidance and history guidance during sampling, with different guidance scales for each. While it is well-known that text guidance enhances generation results, ablation in **Figure R1b** shows that history guidance further improves the quality and consistency, when combined with text guidance. We believe DFoT's sliding window rollout can further synergize with text prompt conditioning by specifying different text prompts for each sliding window. Additionally, DFoT has already shown compatibility with camera pose conditioning in Re10K experiments, and we believe it can extend to other sequential conditioning (e.g. audio).
> > ### References
> - [1] WanTeam et al. "Wan: Open and Advanced Large-Scale Video Generative Models." arXiv 2025.
> - [2] Chen, Tsai-Shien, et al. "Panda-70m: Captioning 70m videos with multiple cross-modality teachers." CVPR 2024.

---

> > ### Comment · Reviewer_RA3f · 2025-04-02
> >
> > Thanks to the authors for their detailed response. Most of my concerns have been addressed except for the apparent artifacts in the additional results of model scalability. It is understandable that scaling on the 1.3B model within the rebuttal period with limited resources is hard. Therefore, the scalability of DFoT could be a future direction down the road. Still, this does not affect the fact that this is a strong ICML submission. In light of this, I will keep my original rating.

---

> > > ### Author Response · Authors · 2025-04-02
> > >
> > > Thank you for your feedback and appreciation!
> > >
> > > Aside from the limited resources, we'd like to highlight that the base model was a 1.3B **text**-to-video model. Traditionally, for a model to do well on **image**-to-video generation, the 1.3B size is not sufficient and would inevitably contain artifacts. For example, almost all pepper cutting (R1-b-mid) results you see online are generated in a **text**-to-video setting or with a much bigger image-to-video model, while we are sampling in a harder **image**-to-video setting only with a smaller model size. Therefore, we respectfully suggest that the observed artifact be interpreted in this context.
> > >
> > > If you have further questions, please let us know, and we will be happy to provide further explanations!

---

### Official Review · Reviewer_mAfS · 2025-03-14

**Overall Recommendation:** 4

**Summary:**

Classifier-free guidance (CFG) greatly improves conditional generation in diffusion models, but applying it to video diffusion—where the number of context frames can vary—introduces significant challenges. Existing architectures often restrict conditioning to a fixed size, and CFG-style history dropout is ineffective. In response, the Diffusion Forcing Transformer (DFoT) and its associated History Guidance methods enable flexible conditioning, substantially enhancing video quality and consistency over potentially very long sequences.

**Claims And Evidence:**

Yes, the claims made in the submission are supported  by clear and convincing evidence.

**Essential References Not Discussed:**

The reference part is good to me.

**Experimental Designs Or Analyses:**

Yes. Very detailed ablations regardind  history gudiance and diffusion force transformer.

**Methods And Evaluation Criteria:**

Yes.

**Other Comments Or Suggestions:**

Please see the above part.

**Other Strengths And Weaknesses:**

Strengths:

Proposes a novel strategy for producing extended or even infinite-length videos.

The idea of training independent noise levels for each frame in a video diffusion framework is innovative and warrants further investigation.

Weaknesses:

It would be valuable to see an ablation study comparing training with independent noise levels for the history frames versus using a uniform noise level for the generated frames.

The long video demonstrations primarily focus on static scenes, leaving open questions about performance on lengthy sequences of human actions (e.g., in Kinetics-600).

Although the approach seems straightforward to adapt to top-tier video generation models like CogVideo (through additional fine-tuning), I’m curious how well it would perform in creating long videos when combined with state-of-the-art model and the proposed training as well as guidance methods.

**Questions For Authors:**

Please see the above part.

**Relation To Broader Scientific Literature:**

Introducing a groundbreaking method of history guidance for extended video generation using diffusion models

**Theoretical Claims:**

Yes, I checked the theoretical claims of  lower bound evidence.

---

> ### Author Rebuttal · Authors · 2025-04-01
>
> We appreciate your positive comments on the novelty and theoretical support of our method, and extensive experimental results. Below, we address your questions by presenting enhanced long video generation achieved through fine-tuning a 1.3B text-to-video foundation model and providing an ablative comparison of the training objectives.
>
> > ### **Q1. Long video generation on dynamic scenes, with SOTA foundation models**
>
> While our original results were limited by our compute and data, we are pleased to report positive outcomes by fine-tuning a state-of-the-art video generative model with DFoT using limited compute. We fine-tuned the Wan2.1 T2V-1.3B model [1], a leading SOTA model, to DFoT for only 20K steps on a subset of the Panda-70M dataset [2]. This was achieved with just 18 lines of code changes, thanks to DFoT’s simplicity. Please see **Figure R1 on [our anonymous website](https://anon-dfot.github.io/)** for our enhanced long video generation results. Key findings include:
>
> - **Scalability of DFoT and History Guidance**: Our method's strengths remain robust at larger scales and in real-world applications. First, we present long video generation results (**Figure R1a**), where DFoT stably generates long 217-frame videos via sliding window rollouts. This is substantially longer than the 49-frame context window used during fine-tuning. Additionally, DFoT handles flexible-length history effectively; Figure R1a shows videos generated by initially conditioning on a single frame and then on the previous 13 frames during rollout. Figure R1b demonstrates video generation by interpolating between the first and last frames. Lastly, we present a direct ablation study on the effectiveness of history guidance (**Figure R1b**). Similar to our paper results, history guidance significantly improves the video quality and consistency.
> - **Dynamic scenes**: Unlike our Re10K demonstrations focusing on static scenes, the generated videos in **Figure R1** are dynamic scenes with human/animal actions and moving objects, at higher resolution. This demonstrates our method's capability to perform well in complex, realistic scenarios, beyond static ones.
>
> > ### **Q2. Ablation on independent vs. uniform noise levels**
>
> There could be multiple interpretations of your suggestion, so we address each of them below:
>
> #### Interpretation 1: Independent vs. uniform noise levels
>
> **Section 6.2**, **Table 1**, and **Figure 4** compare our independent noise level training objective against uniform noise levels. Our independent noise level approach outperforms these baselines using uniform noise levels, such as SD, FS, and BD, both quantitatively (Table 1) and qualitatively (Figure 4). This demonstrates that our proposed training objective offers advantages not only in flexibility (conditioning on arbitrary history frames) but also in performance.
>
> #### Interpretation 2: Fully independent vs. Partially independent noise levels
>
> Alternatively, your question might ask whether DFoT can benefit from training with independent noise levels for some frames and uniform noise levels for others.
>
> To clarify, our problem statement defines history as **any** subset of the frames. While sampling distinguishes between history and generated frames, training does not, to encompass all possible history conditioning during sampling. Since training with partially independent noise levels would deviate from our problem statement, we discussed it in Appendix A.5 for clarity.
>
> As detailed in **Appendix A.5**, if only a subset of history conditioning schemes is desired, fully independent noise may not be optimal due to unnecessary complexity as the number of frames increases. Therefore, we explored partially independent noise level training. Given the maximum length of history frames we aim to support, independent noise levels are only applied up to this maximum length, with uniform noise levels assigned to the remaining frames. Our Minecraft model, which processes 50 latent frames, used a simplified training objective with a maximum length of 25 latent frames. This approach improved training efficiency and video generation performance compared to fully independent noise levels. However, we also found that such efficiency gains are not noticeable for fewer frames. Since modern video foundation models [1] often feature around 20 latent frames, partially independent noise does not significantly increase efficiency, aligning with Diffusion Forcing [3]'s finding that temporal complexity is minor compared to visual complexity.
>
> > ### References
> - [1] WanTeam et al. "Wan: Open and Advanced Large-Scale Video Generative Models." arXiv 2025.
> - [2] Chen et al. "Panda-70m: Captioning 70m videos with multiple cross-modality teachers." CVPR 2024.
> - [3] Chen et al. “Diffusion Forcing: Next-token Prediction Meets Full-Sequence Diffusion.” NeurIPS 2024

---

### Official Review · Reviewer_uQSd · 2025-03-15

**Overall Recommendation:** 4

**Summary:**

This paper regards the noise in the diffusion process as a form of masking, integrating history frames and generated frames into a unified Diffusion Forcing Transformer (DFoT) framework. By combining different masking strategies for history information, the paper proposes several History Guidance (HG) methods, enhancing generation quality, dynamics, and historical consistency. Experimental results show the advantages of the proposed methods in long video generation and many downstream tasks.

After the rebuttal, the authors address my concerns.

**Claims And Evidence:**

This work innovatively introduces the Diffusion Forcing Transformer, integrating generated frames and history conditions within a unified framework. The authors also proposes multiple History Guidance methods, with HG-v already showing performance improvements. Variants HG-t and HG-f further enhance model generalization and dynamic generation capabilities.

However, although the paper claims to support flexible-length history information, experiments do not clearly demonstrate the advantage of History Guidance in ultra-long video generation. The incremental extension method for "unlimited-length" videos achieves coherent transitions but lacks semantic continuity (e.g., supplementary material video i2v_dfot_long_1.mp4, from 11 to 24 seconds). Although HG-t shows potential in integrating long-term and short-term history contexts, its effectiveness has not been validated in general video generation tasks, limiting the method's practical applicability.

**Essential References Not Discussed:**

None

**Experimental Designs Or Analyses:**

The flexibility and effectiveness of DFoT are evaluated on Kinetics-600, RealEstate10K, Minecraft, and Fruit Swapping datasets, as well as in tasks such as video prediction, frame interpolation, long video generation, and imitation learning.

**Methods And Evaluation Criteria:**

Yes. The paper proposes DFoT to address high-quality and consistency issues in long video generation. The experiments validate the model’s capabilities using the Kinetics-600, RealEstate10K, Minecraft, and Fruit Swapping datasets, demonstrating its effectiveness in long video generation tasks.

**Other Comments Or Suggestions:**

None

**Other Strengths And Weaknesses:**

The supplementary materials are entensive, providing impressive experimental results.

**Questions For Authors:**

I am curious about the transition mechanisms between different segments in long video generation, particularly the role of user interactions (there seems to be significant discontinuity between different segments).

**Relation To Broader Scientific Literature:**

This work provides insights into sementically consistent long video generation.

**Theoretical Claims:**

This paper is not primarily a theoretical contribution. Nevertheless, it provides theoretical support for DFoT’s optimization objectives based on Evidence Lower Bound.

---

> ### Author Rebuttal · Authors · 2025-04-01
>
> We appreciate your positive comments on our method's innovation, theoretical support, and evaluation. Below, we address concerns on long video generation and present new results from fine-tuning a 1.3B text-to-video foundation model to DFoT.
>
> > ### **Q1. Advantage of our method in ultra-long video generation**
>
> Long video generation quality is measured by two factors: rollout stability and long-context semantic continuity. Here, we emphasize our method’s exceptional stability, while semantic continuity is discussed in Q2.
>
> Stability refers to maintaining video quality as models condition on previously generated, potentially erroneous frames. Achieving stable rollouts via sliding windows has historically been challenging due to compounding errors [1], with even large-scale foundation models often failing. As noted in Appendix D.6, Re10K is a challenging dataset, and the previous state-of-the-art [2] was limited to 32 frames before failure. This underscores why our method's ability to generate nearly 1000 frames marks a significant improvement in long-rollout stability. Furthermore, Appendix D.6 and Figure 9 show DFoT and history guidance are essential for ultra-long video generation.
>
> > ### **Q2a. Coherent transitions but lacking long-context semantic continuity**
>
> While we acknowledge that our Re10K demos lack long-context semantic continuity, we emphasize that **this limitation is not inherent to our method, DFoT and History Guidance**, which perform well on Minecraft, designed to test long-context consistency. Instead, the issue arises from limited context window size due to computational constraints and the limited semantic length of the Re10K dataset:
>
> - Limited compute: Our Re10K model is trained with a context window size of only 8 frames, as training with larger context windows becomes increasingly expensive. This results in an overlap of only 4 frames between successive sliding windows, insufficient for long-memory in 3D space.
> - Limited training data: The Re10K dataset consists of semantically short video clips without loop closures, typically covering a single room/area (Appendix D.6). Thus, the model cannot learn to maintain semantic continuity over greater lengths (e.g., multiple rooms/areas).
>
> We further argue that the **semantic continuity of the generated videos scales with the semantic length of the training data and the context window size**. Beyond strong performance on Minecraft, we are pleased to present new long video generation results in **Figure R1a on [our anonymous website](https://anon-dfot.github.io/)**. We fine-tuned the Wan2.1 T2V-1.3B model [3] to DFoT on a subset of the Panda-70M dataset [4]. With a larger context window size and longer semantic length of the training data, the results are more coherent and semantically consistent than those from Re10K. This demonstrates that the semantic continuity of generated videos using our method can be improved by simply scaling the training data and context window size.
>
>
> > ### **Q2b. Transition mechanisms between different segments in long video generation**
>
> As detailed in **Appendix C.9**, our Re10K videos are generated by user input of camera poses for each sliding window. For simplicity, we've implemented a basic transition mechanism where the camera pose changes at a constant rate within each sliding window, based on specified distance and rotation angles. This setup causes discontinuous transitions due to velocity changes between sliding windows. We clarify that the **discontinuous transitions stem from the simplicity of our current implementation of camera pose transitions, not a limitation of our method**. Smoother transitions could be achieved with a more advanced mechanism.
>
> > ### **Q3. HG-t's effectiveness in general video generation tasks**
>
> **Section 6.4 demonstrates the practical applicability of HG-t**. We show that HG-t addresses three critical challenges: OOD history, long context generation, and imitation learning. At the core of these tasks is the OOD nature of observed history frames. As noted in **Section 5**, the amount of required training data grows exponentially with the length of conditioning history. Consequently, the OOD problem frequently arises in tasks that require understanding long histories, such as video-to-video generation, and HG-t effectively addresses this core issue. Additionally, the history guidance framework's strength lies in its techniques (HG-v, HG-t, and HG-f) offering unique advantages, allowing the best technique to be chosen based on specific needs of general video generation tasks.
>
> > ### References
> - [1] Qiu et al. "Freenoise: Tuning-free longer video diffusion via noise rescheduling." ICLR 2024.
> - [2] Watson et al. “Controlling space and time with diffusion models.” ICLR 2024.
> - [3] WanTeam et al. "Wan: Open and Advanced Large-Scale Video Generative Models." arXiv 2025.
> - [4] Chen et al. "Panda-70m: Captioning 70m videos with multiple cross-modality teachers." CVPR 2024.

---

### Decision · Program_Chairs · 2025-05-01

**Decision:**

Accept (poster)

**Comment:**

This work introduces Diffusion Forcing Transformer (DFoT), integrating history frames and generation via flexible masking strategies (HG-t/f/v) to enhance long video consistency and dynamics. Initial concerns about semantic continuity in ultra-long sequences (e.g., 24-second artifacts in supplementary videos) and limited validation on dynamic human actions (Kinetics-600) were partially mitigated through rebuttal clarifications. The method demonstrates strong empirical results: 862-frame generation on RealEstate10K, improved FVD/LPIPS metrics across Kinetics-600/Minecraft/FruitSwapping, and theoretical grounding via ELBO derivation. Though comparisons with SOTA models (e.g., CogVideo) are absent, the rigorous evaluations on diverse tasks (prediction/interpolation/imitation) justify technical novelty. Thus, I recommend to accept this paper.